# MME-REALWORLD: COULD YOUR MULTIMODAL LLM CHALLENGE HIGH-RESOLUTION REAL-WORLD SCENARIOS THAT ARE DIFFICULT FOR HUMANS?

**Yi-Fan Zhang**[1,5]   **Huanyu Zhang**[1,5]   **Haochen Tian**[1,5]   **Chaoyou Fu**[2,†]
**Shuangqing Zhang**[2]   **Junfei Wu**[3]   **Feng Li**[3]   **Kun Wang**[4,5]   **Qingsong Wen**[6,†]
**Zhang Zhang**[1,5,†]   **Liang Wang**[1,5]   **Rong Jin**[7]   **Tieniu Tan**[1,2,5]
[1]CASIA, [2]NJU, [3]HKUST, [4]NTU, [5]UCAS, [6]Squirrel AI Learning, [7]Meta AI
[†] Corresponding Author

## ABSTRACT

Comprehensive evaluation of Multimodal Large Language Models (MLLMs) has recently garnered widespread attention in the research community. However, we observe that existing benchmarks present several common barriers that make it difficult to measure the significant challenges that models face in the real world, including: 1) small data scale leads to a large performance variance; 2) reliance on model-based annotations results in restricted data quality; 3) insufficient task difficulty, especially caused by the limited image resolution. To tackle these issues, we introduce MME-RealWorld. Specifically, we collect more than 300 K images from public datasets and the Internet, filtering 13,366 high-quality images for annotation. This involves the efforts of professional 25 annotators and 7 experts in MLLMs, contributing to 29,429 question-answer pairs that cover 43 subtasks across 5 real-world scenarios, extremely challenging even for humans. As far as we know, **MME-RealWorld is the largest manually annotated benchmark to date, featuring the highest resolution and a targeted focus on real-world applications**. We further conduct a thorough evaluation involving 29 prominent MLLMs, such as GPT-4o, Gemini 1.5 Pro, and Claude 3.5 Sonnet. Our results show that even the most advanced models struggle with our benchmarks, where none of them reach 60% accuracy. The challenges of perceiving high-resolution images and understanding complex real-world scenarios remain urgent issues to be addressed.

## 1 INTRODUCTION

In recent years, we have witnessed a significant flourish of Multimodal Large Language Models (MLLMs) (Dai et al., 2024; Liu et al., 2023b; Zhang et al., 2024; Fu et al., 2025). A primary objective behind designing MLLMs has been to develop general intelligent agents capable of comprehensively perceiving human queries and environmental situations through the integration of various multimodal sensory data. Consequently, a plethora of comprehensive evaluation benchmarks have emerged to rigorously assess model capabilities. However, some common concerns also arise:

- **Data Scale.** Many existing benchmarks contain fewer than 10K Question-Answer (QA) pairs (Fu et al., 2024b), such as MME (Fu et al., 2023a), MMbench (Liu et al., 2023c), MMStar (Chen et al., 2024), MM-Vet (Yu et al., 2024), TorchStone (Bai et al., 2023b), and BLINK (Fu et al., 2024c). The limited number of QA can lead to evaluation fluctuations.

- **Annotation Quality.** While some benchmarks, such as MMT-Bench (Ying et al., 2024) and SEED-Bench (Li et al., 2024b), are relatively larger in scale, their annotations are generated by LLMs or MLLMs. This annotation process is inherently limited by the performance of the used models. In our benchmark, for example, the best-performing model, InternVL-2, merely achieves 50% accuracy. Consequently, relying on models would inevitably introduce significant noise, compromising the quality of the annotations.

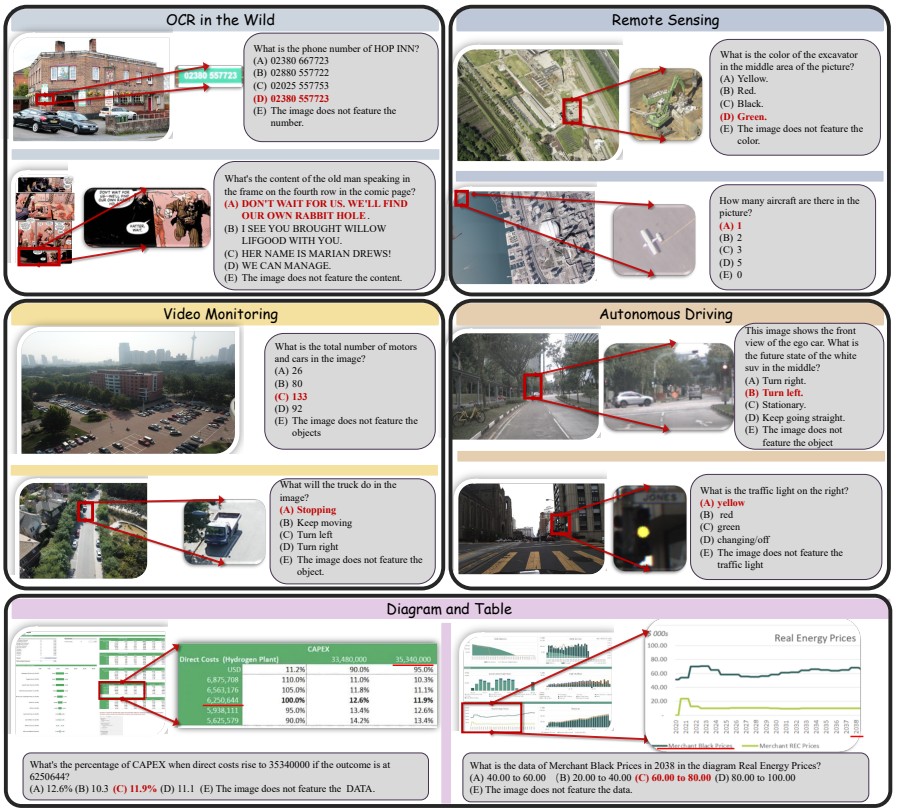

Figure 1: **Diagram of MME-RealWorld.** Our benchmark contains 5 real-world domains, covering 43 perception and reasoning subtasks. Each QA pair offers 5 options. We highlight and magnify the image parts relevant to the question in a red box for better visibility.

- **Task Difficulty.** To date, the top performance of some benchmarks has reached the accuracy of 80%-90% (Mathew et al., 2021; Masry et al., 2022; Singh et al., 2019; Liu et al., 2023c; Li et al., 2023c), and the performance margin between advanced MLLMs is narrow. This makes it challenging to verify the benefits or improvements of advanced models and to distinguish which one is significantly better.

In light of these concerns, we propose a new benchmark named MME-RealWorld. We first pay attention to a series of well-motivated families of datasets, considering images from sources such as autonomous driving, remote sensing, video surveillance, newspapers, street views, and financial charts. These scenarios are difficult even for humans, where we hope that MLLMs can really help. Considering these topics, we collect a total of 13,366 high-resolution images from more than 300K public and internet sources. These images have an average resolution of 2,000×1,500, containing rich image details. 25 professional annotators and 7 experts in MLLMs are participated to annotate and check the data quality, and meanwhile ensuring that all questions are challenging for MLLMs. Note that most questions are even hard for humans, requiring multiple annotators to answer and double-check the results. As shown in Fig. 2(a), MME-RealWorld finally contains 29,429 annotations for 43 sub-class tasks, where each one has at least 100 questions. 29 advanced MLLMs are evaluated on our benchmark, along with detailed analysis. We conclude the main advantages of MME-RealWorld compared to existing counterparts as follows:

- **Data Scale.** With the efforts of a total of 32 volunteers, we have manually annotated 29,429 QA pairs focused on real-world scenarios, making this the largest fully human-annotated benchmark known to date.

- **Data Quality.** 1) Resolution: Many image details, such as a scoreboard in a sports event, carry critical information. These details can only be properly interpreted with high-resolution images, which are essential for providing meaningful assistance to humans. To

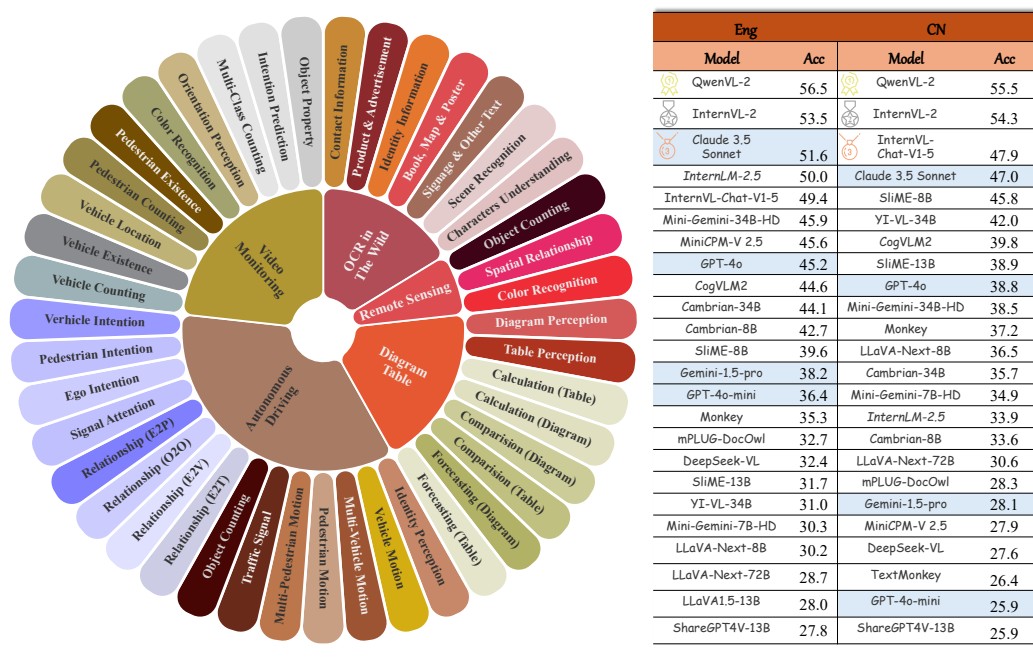

| Eng | | CN | |
|---|---|---|---|
| **Model** | **Acc** | **Model** | **Acc** |
| QwenVL-2 | 56.5 | QwenVL-2 | 55.5 |
| InternVL-2 | 53.5 | InternVL-2 | 54.3 |
| Claude 3.5 Sonnet | 51.6 | InternVL-Chat-V1-5 | 47.9 |
| InternLM-2.5 | 50.0 | Claude 3.5 Sonnet | 47.0 |
| InternVL-Chat-V1-5 | 49.4 | SliME-8B | 45.8 |
| Mini-Gemini-34B-HD | 45.9 | YI-VL-34B | 42.0 |
| MiniCPM-V 2.5 | 45.6 | CogVLM2 | 39.8 |
| GPT-4o | 45.2 | SliME-13B | 38.9 |
| CogVLM2 | 44.6 | GPT-4o | 38.8 |
| Cambrian-34B | 44.1 | Mini-Gemini-34B-HD | 38.5 |
| Cambrian-8B | 42.7 | Monkey | 37.2 |
| SliME-8B | 39.6 | LLaVA-Next-8B | 36.5 |
| Gemini-1.5-pro | 38.2 | Cambrian-34B | 35.7 |
| GPT-4o-mini | 36.4 | Mini-Gemini-7B-HD | 34.9 |
| Monkey | 35.3 | InternLM-2.5 | 33.9 |
| mPLUG-DocOwl | 32.7 | Cambrian-8B | 33.6 |
| DeepSeek-VL | 32.4 | LLaVA-Next-72B | 30.6 |
| SliME-13B | 31.7 | mPLUG-DocOwl | 28.3 |
| YI-VL-34B | 31.0 | Gemini-1.5-pro | 28.1 |
| Mini-Gemini-7B-HD | 30.3 | MiniCPM-V 2.5 | 27.9 |
| LLaVA-Next-8B | 30.2 | DeepSeek-VL | 27.6 |
| LLaVA-Next-72B | 28.7 | TextMonkey | 26.4 |
| LLaVA1.5-13B | 28.0 | GPT-4o-mini | 25.9 |
| ShareGPT4V-13B | 27.8 | ShareGPT4V-13B | 25.9 |

(a) **Real-World Tasks**      (b) **Leaderboard**

Figure 2: **Task Categories** (left). Our benchmark spans 5 key domains and 43 subtasks highly related to real-world scenarios, including 13, 366 high-resolution images and 29, 429 annotations. **Model Performance** (right). Average accuracies of advanced MLLMs are shown across both the English and Chinese splits of the dataset.

the best of our knowledge, MME-RealWorld features the highest average image resolution among existing competitors. 2) Annotation: All annotations are manually completed, with a professional team cross-checking the results to ensure data quality.

- **Task Difficulty and Real-World Utility.** The performance of different MLLMs is shown in Fig. 2(b), in which we can see that even the most advanced models have not surpassed 60% accuracy. Additionally, as illustrated in Fig. 1, many real-world tasks are significantly more difficult than those in traditional benchmarks. For example, in video monitoring, a model needs to count the presence of 133 vehicles, or in remote sensing, it must identify and count small objects on a map with an average resolution exceeding 5000×5000.

- **MME-RealWorld-CN.** Existing Chinese benchmark (Liu et al., 2023c) is usually translated from its English version. This has two limitations: 1) Question-image mismatch. The image may relate to an English scenario, which is not intuitively connected to a Chinese question. 2) Translation mismatch (Tang et al., 2024). The machine translation is not always precise and perfect enough. We collect additional images that focus on Chinese scenarios, asking Chinese volunteers for annotation. This results in 5, 917 QA pairs.

## 2 MME-REALWORLD

In this section, we outline the data collection process, question annotation procedure, and provide a statistical overview of each domain and subtask in MME-RealWorld and its Chinese version. We visualize different tasks from the 5 image domains in Fig. 1. Detailed information on data sources, evaluation tasks, and visualized results can be found in Sec. B.

### 2.1 INSTRUCTION AND CRITERION

For each question, we manually construct four options, with one being the correct answer and the other three being the texts appearing in the image or options similar to the correct one. This greatly

Table 1: **Prompt setting of MME-RealWorld**.

| |
|---|
| `[Image] [Question]` The choices are listed below:
(A) `[Choice A]`
(B) `[Choice B]`
(C) `[Choice C]`
(D) `[Choice D]`
(E) `[Choice E]`
Select the best answer to the above multiple-choice question based on the image. Respond with only the letter (A, B, C, D, or E) of the correct option.
The best answer is: |

enhances the difficulty, forcing the model to deeply understand the details of the image. We also provide an additional choice E, which allows the model to reject for answering because there is no right answer. We try to use the model's default prompt for multiple-choice questions, but if the model does not have the default prompt, we use a common prompt as shown in Tab. 1.

**Evaluation Metric.** We first apply a rule-based filter to the answers generated by MLLM, aligning them with the given answer options and checking for correctness against the ground truth. Let the dataset be denoted as $\mathcal{D} = \{\mathcal{D}_d = \{\mathcal{T}_t\}_{t=1}^{T_d}\}_{d=1}^{D}$, where each domain $\mathcal{D}_d$ consists of $T_d$ subtasks. For each subtask, we calculate the accuracy across all annotations. For each domain, we compute two metrics: 1) **Average Accuracy (Avg).** the weighted average accuracy across all subtasks, given by $\sum_{t=1}^{T_d} \text{Avg}(\mathcal{T}_t) \times |\mathcal{T}_t|/|\mathcal{D}_d|$, where $|\cdot|$ is the instance number contained in one set, and 2) **Class-based Average Accuracy (Avg-C).** the unweighted average accuracy across subtasks, given by $\sum_{t=1}^{T_d} \text{Avg}(\mathcal{T}_t)/T_d$. Similarly, for the entire dataset, we report the overall Average Accuracy across all samples, and the class-based average accuracy across domains.

## 2.2 DATA COLLECTION AND ANNOTATION

**Optical Character Recognition in the Wild (OCR).** It is specifically designed to evaluate the model's ability to perceive and understand textual information in the real-world. We manually selecte $3,293$ images with complex scenes and recognizable text information from $150,259$ images in existing high-resolution datasets as our image sources. These images span various categories such as street scenes, shops, posters, books, and competitions. The volunteers are worked for annotation, each with at least a foundational understanding of multimodal models, to independently generate questions and answers. These annotations are subsequently reviewed and further refined by another volunteers. Based on the image annotations, we categorize these $3,297$ images into 5 perception tasks, totaling $5,740$ QA pairs: contact information and addresses, identity information, products and advertisements, signage and other text, as well as natural text recognition in elevation maps and books. Additionally, there are two reasoning tasks with $500$ QA pairs: 1) scene understanding of the entire image, which requires the model to locate and comprehend important text such as competition results, and 2) character understanding, focusing on comics or posters where the model needs to analyze relationships and personalities based on dialogue or presentation.

**Remote Sensing (RS).** The images have a wide range of applications in real-world scenarios. Some images possess extremely high quality, with individual image sizes reaching up to 139MB and containing very rich details, which makes it difficult even for humans to perceive specific objects. We manually select $1,298$ high-resolution images from over $70,000$ public remote sensing images, ensuring that each image is of high quality, with sufficient resolution and rich detail. One professional researcher is involved in annotating the data, and another researcher checks and improves the annotations, resulting in $3,738$ QA pairs. There are 3 perception tasks: object counting, color recognition, and spatial relationship understanding.

**Diagram and Table (DT).** Although there are already some datasets related to table and chart understanding, they mostly feature simple scenes. We focus on highly complex chart data, such as financial reports, which contain extensive numerical information and mathematical content, presenting new challenges for MLLMs. We filter $2,570$ images from the internet, with annotations performed by two volunteer and reviewed by another one. We categorize these annotations into 4 tasks based on the question format: 1) Diagram and Table Perception ($5,433$ QA pairs): involve locating specific values of elements within the diagrams and tables; 2) Diagram Reasoning ($250$ QA pairs): include tasks such as identifying the maximum and minimum values in a chart, performing

simple calculations, and predicting trends; and 3) table Reasoning (250 QA pairs): focus on simple calculations related to specific elements, understanding mathematical concepts like maximum and minimum values, and locating corresponding elements.

**Autonomous Driving (AD).** It demands extensive general knowledge and embodied understanding capability. We emphasize challenging driving scenarios that involve distant perceptions and intricate interactions among dynamic traffic agents. Specifically, we manually select a subset of $2,715$ images from over $40,000$ front-view images captured by onboard cameras in open-source datasets. These images cover a diverse range of weather conditions, geographic locations, and traffic scenarios. Besides, a volunteer carefully annotates each image, and the other one conducts a thorough review, resulting in $3,660$ QA pairs for perception tasks and $1,334$ QA pairs for reasoning tasks. The perception tasks include objects identification, object attribute identification, and object counting for traffic elements such as vehicle, pedestrian, and signals. The latter is categorized into 3 main tasks: 1) Intention Prediction: focus on predicting driving intention of a designated traffic agent in the short-term future. 2) Interaction Relation Understanding: involve reasoning about ego vehicle's reaction to other traffic elements, and the interactions between these elements. 3) Driver Attention Understanding: require reasoning about the traffic signal that the driver should pay attention to.

**Monitoring (MO).** The images are from various application scenarios for public safety, e.g., streets, shopping malls, and expressway intersections. We focus on complex high-resolution monitoring images that include many real-world challenges, like scale variations and out-of-view, as possible which could test whether the model handles them robustly in practice. Specifically, $1,601$ high-resolution images are manually selected from over $10,000$ public dataset images, which are captured from a broad range of cameras, viewpoints, scene complexities, and environmental factors across day and night. In terms of annotations, two volunteers manually annotate each image carefully, and multi-stage careful inspections and modifications are performed by another one. When these refined image annotations are completed, $1,601$ images are categorized into 3 main perception tasks, totaling $2,196$ QA pairs, including object counting and location, and attribute recognition. Furthermore, 3 reasoning tasks are well-designed with $498$ QA pairs: 1) calculate the sum of different objects, which requires the model to perceive various objects and calculate their total number accurately; 2) intention reasoning, focusing on reasoning the next route and turn of the specific object; 3) attribute reasoning, focusing on reasoning the specific materials and functions of the given objects.

Fig. 15 shows the distribution of tasks across various domains.

## 2.3 MME-RealWorld-CN

The traditional general VQA approach (Liu et al., 2023c) uses a translation engine to extend QA pairs from English to Chinese. However, it may face visual-textual misalignment problems (Tang et al., 2024), failing to address complexities related to nuanced meaning, contextual distortion, language bias, and question-type diversity. Additionally, asking questions in Chinese about images containing only English texts is not intuitive for benchmarking Chinese VQA capabilities. By contrast, we follow the steps below to construct a high-quality Chinese benchmark:

- **Selection.** For video monitoring, autonomous driving, and remote sensing, many images do not contain English information. Therefore, we select a subset of the aforementioned question pairs, double-checking to ensure they do not contain any English information.

- **Translation.** Translate the questions and answers by four professional researchers, all of whom are familiar with both English and Chinese.

- **Collection.** For diagrams and tables, since the original images often contain English information (e.g., legends/captions), we collect additional 300 tables and 301 diagrams from the Internet, where the contents are in Chinese. This data is further annotated by one volunteer, resulting in $301 \times 4$ QA pairs, where the task type is the same as diagram and table in MME-RealWorld. Similarly, for OCR in the wild, we also collect additional 939 images for all the subtasks.

In total, MME-RealWorld-CN has $1,889$ additional images and total $5,917$ QA pairs, which is a smaller version of MME-RealWorld, but it retains similar task types, image quality, and task difficulty. The examples can be seen in Fig. 12.

## 2.4 QUALITY CONTROL AND ANALYSIS

During the annotation process, we impose the following requirements on annotators: 1. We ensure that all questions can be answered based on the image (except for specially constructed questions where the correct option is "E"), meaning that humans can always find the answers within the image. This approach prevents forcing annotators to provide answers based on low-quality images or images containing vague information. 2. The area of the object being questioned in each image must not exceed $1/10$ of the total image area. This ensures that the object is not overly prominent, preventing humans from easily identifying the answer at first glance. 3. Each annotation is cross-checked by at least two professional multimodal researchers to ensure accuracy and prevent annotation errors caused by human bias.

Table 2: **Comparison of benchmarks.** MME-RealWorld is the largest fully human-annotated dataset, featuring the highest average resolution and the most challenging tasks.

| Benchmark | # QA-Pair | Fully Human Annotation | CN | Average Resolution | LLaVA-1.5-7B Performance |
|---|---|---|---|---|---|
| VizWiz | 8000 | × | × | 1224×1224 | 50.0 |
| RealWorldQA | 765 | × | × | 1536×863 | - |
| MMStar | 1500 | × | × | 512×375 | 30.3 |
| ScienceQA | 21000 | × | × | 378×249 | 71.6 |
| ChartQA | 32719 | × | × | 840×535 | - |
| MM-Vet | 218 | × | × | 1200×675 | 31.1 |
| Seed-Bench | 19242 | × | × | 1024×931 | 66.1 |
| SEED-Bench-2-Plus | 2300 | × | × | 1128×846 | 36.8 |
| MMT-Bench | 32325 | × | × | 2365×377 | 49.5 |
| MathVista | 735 | × | × | 539×446 | 26.1 |
| TouchStone | 908 | × | × | 897×803 | - |
| VisIT-Bench | 1159 | × | × | 765×1024 | - |
| BLINK | 3807 | × | × | 620×1024 | 37.1 |
| CV-Bench | 2638 | × | × | 1024×768 | - |
| TextVQA | 5734 | ✓ | × | 985×768 | 58.2 |
| MME | 2374 | ✓ | × | 1161×840 | 76.0 |
| MMBench | 3217 | ✓ | ✓ | 512×270 | 64.3 |
| MME-RealWorld | 29429 | ✓ | ✓ | 2000x1500 | 24.9 |

The comparison of benchmarks is shown in Tab. 2. The maximal resolution of MME-RealWorld is $42,177,408$ pixels, with dimensions of $5304 \times 7952$. The average resolution is $3,007,695$ pixels, equivalent to an image size of approximately $2000 \times 1500$. This resolution is significantly higher than that of existing benchmarks. For instance, the highest benchmark, MME, has an average resolution of $975,240$ pixels, corresponding to an image size of about $1161 \times 840$. The exceptional image quality and our strict, fully human annotation process make our tasks the most challenging among all benchmarks. This is evidenced by the baseline model LLaVA-1.5-7B achieving an accuracy of just 24.9%, significantly lower than on other benchmarks. Although some benchmarks may approach our level of difficulty, this is primarily due to the inherent complexity of their tasks. For instance, MathVista focuses on pure mathematical problems, and MM-Vet involves multi-step reasoning—both of which are naturally challenging and result in lower baseline performance. However, the majority of our tasks are centered on real-world perception problems. This means that, current MLLMs still struggle to effectively address human-level perceptual challenges.

## 3 EXPERIMENTS

We evaluate a total of 24 open-source MLLMs, including Qwen-VL-Chat (Bai et al., 2023a), LLaVA, LLaVA-Next (Li et al., 2024a), TextMonkey (Liu et al., 2024b), mPLUG-DocOwl 1.5 (Hu et al., 2024a), ShareGPT4V (Chen et al., 2023b), MiniGPT-v2 (Chen et al., 2023a), Monkey (Li et al., 2023f), OtterHD (Li et al., 2023a), Cambrian-1 (Tong et al., 2024), Mini-Gemini-HD (Li et al., 2024d), MiniCPM-V 2.5 (Hu et al., 2024b), DeepSeek-VL (Lu et al., 2024a), YI-VL-34B[1], SliME (Zhang et al., 2024), CogVLM2[2], InternLM-XComposer2.5 (Zhang et al., 2023), InternVL-Chat V1-5, InternVL-2 (Chen et al., 2023c), and Qwen2-VL[3], as well as 4 close-source MLLMs, including, GPT-4o[4], GPT-4o-mini, Gemini 1.5 pro (Team et al., 2023), and Claude 3.5 Sonnet[5].

### 3.1 RESULTS ON MME-REALWORLD

#### 3.1.1 PERCEPTION

Tab. 3 presents the perception capabilities of different models across 5 domains. Overall, Qwen2-VL demonstrates the strongest perception abilities, outperforming other closed-source models. However, the performance varies across different tasks, with some key observations as follows:

---

[1] https://huggingface.co/01-ai/Yi-VL-34B
[2] https://github.com/THUDM/CogVLM2
[3] https://github.com/QwenLM/Qwen2-VL
[4] https://openai.com/index/hello-gpt-4o/
[5] https://www.anthropic.com/news/claude-3-5-sonnet

Table 3: **Experimental results on the perception tasks.** Models are ranked according to their average performance. Rows corresponding to proprietary models are highlighted in gray for distinction. "OCR", "RS", "DT", "MO", and "AD" each indicate a specific task domain: Optical Character Recognition in the Wild, Remote Sensing, Diagram and Table, Monitoring, and Autonomous Driving, respectively. "Avg" and "Avg-C" indicate the weighted average accuracy and the unweighted average accuracy across subtasks in each domain.

| Method | LLM | Perception | | | | | | |
|---|---|---|---|---|---|---|---|---|
| **Task Split** 
 **# QA pairs** | | OCR 
 5740 | RS 
 3738 | DT 
 5433 | MO 
 2196 | AD 
 3660 | Avg 
 20767 | Avg-C 
 20767 |
| Qwen2-VL | Qwen2-7B | 81.38 | 44.81 | 70.18 | 37.30 | 34.62 | 58.96 | 53.66 |
| InternVL-2 | InternLM2.5-7B-Chat | 73.92 | 39.35 | 62.80 | 53.19 | 35.46 | 55.82 | 52.94 |
| Claude 3.5 Sonnet | - | 72.47 | 25.74 | 67.44 | 32.19 | 40.77 | 52.90 | 47.72 |
| InternLM-XComposer2.5 | InternLM2-7B | 69.25 | 36.12 | 63.92 | 39.48 | 33.63 | 52.47 | 48.48 |
| InternVL-Chat-V1.5 | InternLM2-Chat-20B | 71.51 | 33.55 | 55.83 | 51.16 | 31.42 | 51.36 | 48.69 |
| Mini-Gemini-34B-HD | Nous-Hermes-2-Yi-34B | 69.55 | 40.40 | 44.36 | 39.61 | 32.70 | 48.05 | 45.32 |
| MiniCPM-V 2.5 | Llama3-8B | 66.79 | 27.69 | 52.81 | 38.70 | 34.15 | 47.37 | 44.03 |
| Cambrian-1-34B | Nous-Hermes-2-Yi-34B | 66.45 | 38.63 | 40.44 | 45.98 | 33.61 | 46.68 | 45.02 |
| GPT-4o | - | 77.69 | 28.92 | 46.68 | 33.93 | 22.43 | 46.43 | 41.93 |
| CogVLM2-llama3-Chat | Llama3-8B | 69.97 | 28.76 | 47.51 | 33.74 | 30.22 | 45.84 | 42.04 |
| Cambrian-1-8B | Llama3-8B-Instruct | 58.68 | 40.05 | 32.73 | 47.68 | 38.52 | 43.82 | 43.53 |
| SliME-8B | Llama3-8B | 53.45 | 42.27 | 29.34 | 40.62 | 33.66 | 40.29 | 39.87 |
| Gemini-1.5-pro | - | 67.62 | 13.99 | 39.90 | 31.11 | 26.64 | 39.63 | 35.85 |
| GPT-4o-mini | - | 62.51 | 6.69 | 44.23 | 26.50 | 24.18 | 37.12 | 32.82 |
| Monkey | Qwen-7B | 54.63 | 24.99 | 32.51 | 28.01 | 29.67 | 36.30 | 33.96 |
| mPLUG-DocOwl 1.5 | Llama-7B | 51.15 | 23.71 | 29.34 | 24.97 | 28.28 | 33.71 | 31.49 |
| DeepSeek-VL | DeepSeek-LLM-7b-base | 49.55 | 25.49 | 23.38 | 26.97 | 33.39 | 33.14 | 31.76 |
| SliME-13B | Vicuna-13B | 50.58 | 25.82 | 20.93 | 24.73 | 27.16 | 31.50 | 29.84 |
| Mini-Gemini-7B-HD | Vicuna-7B-v1.5 | 42.02 | 31.30 | 22.31 | 34.15 | 24.81 | 31.07 | 30.92 |
| YI-VL-34B | Yi-34B-Chat | 44.95 | 31.62 | 15.99 | 34.85 | 28.31 | 30.97 | 31.14 |
| LLaVA-Next | Llama3-8B | 47.94 | 25.42 | 26.63 | 19.46 | 18.66 | 30.14 | 27.62 |
| LLaVA-Next | Qwen-72B | 37.07 | 29.13 | 27.68 | 29.37 | 17.98 | 29.01 | 28.25 |
| LLaVA1.5-13B | Vicuna-13B | 44.10 | 23.27 | 20.17 | 20.45 | 26.12 | 28.42 | 26.82 |
| ShareGPT4V-13B | Vicuna-13B | 44.55 | 23.06 | 20.17 | 19.26 | 26.12 | 28.38 | 26.63 |
| MiniGPT-v2 | Llama 2-7B-Chat | 39.02 | 23.33 | 20.41 | 19.26 | 25.96 | 26.94 | 25.60 |
| ShareGPT4V-7B | Vicuna-7B | 39.39 | 22.10 | 20.08 | 19.13 | 26.04 | 26.73 | 25.35 |
| LLaVA1.5-7B | Vicuna-7B | 38.69 | 22.12 | 20.08 | 19.13 | 26.04 | 26.54 | 25.21 |
| Qwen-VL-Chat | Qwen-7B | 32.37 | 15.14 | 15.59 | 22.13 | 15.08 | 20.75 | 20.06 |
| TextMonkey | Qwen-7B | 37.30 | 11.69 | 5.93 | 16.14 | 14.26 | 18.18 | 17.06 |

- GPT-4o performs well in real-world OCR tasks, achieving 77% accuracy, which is second only to Qwen2-VL. However, its performance significantly drops in more challenging tasks, lagging behind other top-ranked models. This trend is also observed in other closed-source models, such as Gemini-1.5-Pro and GPT-4o-mini, which perform well in OCR tasks but struggle significantly in other real-world tasks. There are three possible reasons: 1) Close-source models often have limitations on the maximum image size and resolution when uploading local images. For example, Claude 3.5 Sonnet has a maximum resolution limit of 8K and a maximum image quality of 5MB, while GPT-4o and Gemini-pro allow up to 20MB. This restricts the input of some high-quality images, as we have to compress the images for upload. 2) Close-source models tend to be more conservative. We observe that the proportion of responses, where closed-source models output "E" indicating that the object in question is not present in the image, is high. This suggests that these models may adopt a conservative response strategy to avoid hallucinations or to provide safer answers. 3) Closed-source models sometimes refuse to answer certain questions. Due to different input/output filtering strategies, some samples are considered to involve privacy or harmful content and are therefore not answered.

- Models allowing higher resolution input, such as Mini-Gemini-HD and SliME, demonstrate a significant advantage over models directly using vision encoders like CLIP, such as ShareGPT4V and LLaVA1.5. At the same model size, these models consistently improve across different subtasks. This highlights the critical importance of high-resolution image processing for addressing complex real-world tasks.

- There are also notable trends across different domains. Remote sensing tasks involve processing extremely large images, demanding a deeper comprehension of image details. Models that focus on high-resolution input, such as Cambrian-1, Mini-Gemini-HD, and

Table 4: **Experimental results on the reasoning tasks.** Models are ranked according to their average performance. Rows corresponding to proprietary models are highlighted in gray for distinction. "OCR", "RS", "DT", "MO", and "AD" each indicate a specific task domain: Optical Character Recognition in the Wild, Remote Sensing, Diagram and Table, Monitoring, and Autonomous Driving, respectively. "Avg" and "Avg-C" indicate the weighted average accuracy and the unweighted average accuracy across subtasks in each domain.

| Method | LLM | Reasoning | | | | | |
|---|---|---|---|---|---|---|---|
| **Task Split** | | **OCR** | **DT** | **MO** | **AD** | **Avg** | **Avg-C** |
| **# QA pairs** | | 500 | 500 | 498 | 1334 | 2832 | 2832 |
| Claude 3.5 Sonnet | - | 61.90 | 61.20 | 41.79 | 31.92 | 44.12 | 49.20 |
| Qwen2-VL | Qwen2-7B | 63.40 | 48.60 | 33.13 | 31.47 | 40.39 | 44.15 |
| InternVL-2 | InternLM2.5-7B-Chat | 57.40 | 39.00 | 43.57 | 29.84 | 38.74 | 42.45 |
| GPT-4o | - | 61.40 | 44.80 | 36.51 | 26.41 | 37.61 | 42.28 |
| CogVLM2-llama3-Chat | Llama3-8B | 54.00 | 32.80 | 41.16 | 31.18 | 37.25 | 39.79 |
| InternVL-Chat-V1-5 | InternLM2-Chat-20B | 56.80 | 35.40 | 37.35 | 28.94 | 36.48 | 39.62 |
| Cambrian-1-8B | Llama3-8B-Instruct | 53.20 | 27.40 | 42.37 | 30.73 | 36.16 | 38.43 |
| SliME-8B | Llama3-8B | 53.20 | 29.40 | 36.14 | 31.55 | 35.80 | 37.57 |
| MiniCPM-V 2.5 | Llama3-8B | 44.00 | 31.80 | 36.95 | 31.03 | 34.50 | 35.95 |
| SliME-13B | Vicuna-13B | 41.00 | 39.00 | 33.13 | 30.80 | 34.46 | 35.98 |
| InternLM-XComposer2.5 | InternLM2-7B | 53.40 | 41.00 | 17.67 | 29.99 | 33.90 | 35.52 |
| GPT-4o-mini | - | 47.00 | 39.80 | 25.81 | 26.79 | 32.48 | 34.85 |
| YI-VL-34B | Yi-34B-Chat | 42.40 | 26.00 | 31.33 | 31.55 | 32.45 | 32.82 |
| LLaVA-Next | Llama3-8B | 55.20 | 23.40 | 21.08 | 30.73 | 32.06 | 32.60 |
| Mini-Gemini-34B-HD | Nous-Hermes-2-Yi-34B | 59.20 | 39.20 | 20.48 | 22.84 | 31.73 | 35.43 |
| Gemini-1.5-pro | - | 52.70 | 33.20 | 28.33 | 19.29 | 29.19 | 33.36 |
| Monkey | Qwen-7B | 27.20 | 20.80 | 27.31 | 33.04 | 28.84 | 27.09 |
| DeepSeek-VL | DeepSeek-LLM-7b-base | 45.20 | 23.80 | 16.67 | 27.31 | 27.98 | 28.25 |
| LLaVA-Next | Qwen-72B | 17.20 | 34.20 | 27.31 | 29.69 | 27.86 | 27.10 |
| Cambrian-1-34B | Nous-Hermes-2-Yi-34B | 55.00 | 36.00 | 19.48 | 16.07 | 27.06 | 31.64 |
| mPLUG-DocOwl 1.5 | Llama-7B | 42.60 | 19.80 | 20.48 | 26.04 | 26.88 | 27.23 |
| Mini-Gemini-7B-HD | Vicuna-7B-v1.5 | 35.40 | 24.60 | 25.90 | 23.29 | 26.12 | 27.30 |
| LLaVA1.5-13B | Vicuna-13B | 30.20 | 20.80 | 27.51 | 24.78 | 25.51 | 25.82 |
| ShareGPT4V-13B | Vicuna-13B | 26.00 | 20.80 | 27.31 | 24.55 | 24.63 | 24.67 |
| LLaVA1.5-7B | Vicuna-7B | 26.00 | 20.60 | 25.90 | 24.18 | 24.17 | 24.17 |
| ShareGPT4V-7B | Vicuna-7B | 24.15 | 20.60 | 26.10 | 24.18 | 23.88 | 23.76 |
| MiniGPT-v2 | Llama 2-7B-Chat | 30.00 | 20.40 | 16.87 | 23.66 | 23.01 | 22.73 |
| Qwen-VL-Chat | Qwen-7B | 28.60 | 13.60 | 16.47 | 24.63 | 21.95 | 20.83 |
| TextMonkey | Qwen-7B | 30.40 | 2.20 | 4.42 | 20.01 | 15.96 | 14.26 |

SliME, outperform other models in these tasks. Additionally, models trained on large amounts of chart data exhibit improved perception capabilities for complex charts. For instance, SliME and LLaVA1.5 have limited and relatively simple chart data in their training sets, resulting in inferior performance in this category compared to more recent models.

### 3.1.2 REASONING

Experimental results on the reasoning tasks are shown in Tab. 4. In terms of reasoning ability, Claude 3.5 Sonnet distinguishes itself as the top performer across most domains, particularly outpacing the second-place Qwen2-VL by 12.6% in chart-related tasks. The closed-source model GPT-4o also performs well, trailing slightly behind the third-place InternVL-2 but even outperforming it in several domains. Most open-source models perform poorly, with traditional baseline methods such as LLaVA1.5 and Qwen-VL-Chat yielding results close to random guessing. Furthermore, reasoning tasks are more challenging than perception tasks. Even the top-ranked model fails to achieve an average accuracy above 45%, with class-based accuracy not exceeding 50%. This indicates that current models still have a significant gap to bridge to reach human-level reasoning capabilities.

### 3.2 RESULTS ON MME-REALWORLD-CN

Results of perception tasks and reasoning tasks are presented in Tab. 5 and Tab. 12, respectively. The models show different performances compared to the MME-RealWorld English version.

1) Qwen2-VL and InternVL-2 significantly outperform existing models in both perception and reasoning tasks in the Chinese version. The performance of these two models even surpasses their

Table 5: **Experimental results on the perception tasks of MME-RealWorld-CN.** Models are ranked according to their average performance. Rows corresponding to proprietary models are highlighted in gray for distinction. "OCR", "RS", "DT", "MO", and "AD" each indicate a specific task domain: Optical Character Recognition in the Wild, Remote Sensing, Diagram and Table, Monitoring, and Autonomous Driving, respectively. "Avg" and "Avg-C" indicate the weighted average accuracy and the unweighted average accuracy across subtasks in each domain.

| Method | LLM | Perception | | | | | | |
|---|---|---|---|---|---|---|---|---|
| **Task Split** | | OCR | RS | DT | MO | AD | Avg | Avg-C |
| **# QA pairs** | | 1908 | 300 | 602 | 500 | 700 | 4010 | 4010 |
| InternVL-2.5 | InternLM2.5-7B-Chat | 75.79 | 49.33 | 80.40 | 29.20 | 35.86 | 61.72 | 54.12 |
| Qwen2-VL | Qwen-7B | 70.28 | 38.33 | 89.20 | 29.40 | 36.86 | 59.80 | 52.81 |
| InternVL-2 | InternLM2.5-7B-Chat | 69.92 | 41.33 | 71.63 | 39.3 | 34.14 | 57.97 | 51.26 |
| InternVL-Chat-V1-5 | InternLM2-Chat-20B | 60.59 | 32.00 | 60.12 | 32.40 | 32.14 | 49.90 | 43.45 |
| Claude 3.5 Sonnet | - | 54.44 | 32.67 | 74.09 | 25.00 | 32.43 | 48.25 | 43.73 |
| SliME-8B | Llama3-8B | 53.93 | 41.33 | 58.25 | 29.20 | 31.29 | 46.60 | 42.80 |
| GPT-4o | - | 55.90 | 23.67 | 54.86 | 25.20 | 21.14 | 43.44 | 36.15 |
| YI-VL-34B | Yi-34B-Chat | 51.41 | 34.33 | 49.52 | 25.20 | 27.71 | 42.45 | 37.63 |
| SliME-13B | Vicuna-13B | 50.63 | 17.33 | 48.49 | 17.80 | 33.23 | 40.69 | 33.50 |
| Cambrian-1-34B | Nous-Hermes-2-Yi-34B | 48.11 | 33.79 | 44.34 | 27.60 | 26.43 | 40.13 | 36.05 |
| CogVLM2-llama3-Chat | Llama3-8B | 46.12 | 22.00 | 39.48 | 24.80 | 34.14 | 38.57 | 33.31 |
| Mini-Gemini-34B-HD | Nous-Hermes-2-Yi-34B | 41.82 | 38.28 | 40.60 | 27.80 | 34.29 | 38.31 | 36.56 |
| LLaVA-Next | Llama3-8B | 40.62 | 31.67 | 37.49 | 35.40 | 27.29 | 36.50 | 34.49 |
| Gemini-1.5-pro | - | 48.32 | 12.33 | 39.78 | 25.20 | 17.57 | 36.10 | 28.64 |
| Monkey | Qwen-7B | 40.46 | 26.55 | 41.12 | 19.20 | 35.86 | 36.07 | 32.64 |
| InternLM-XComposer2.5 | InternLM2-7B | 39.26 | 38.33 | 38.88 | 19.40 | 33.57 | 35.66 | 33.89 |
| Mini-Gemini-7B-HD | Vicuna-7B-v1.5 | 39.66 | 17.24 | 39.29 | 16.80 | 28.29 | 33.09 | 28.26 |
| Cambrian-1-8B | Llama3-8B-Instruct | 32.71 | 35.86 | 30.28 | 27.60 | 35.57 | 32.44 | 32.40 |
| mPLUG-DocOwl 1.5 | LLama-7B | 33.33 | 18.62 | 31.83 | 25.60 | 28.43 | 30.19 | 27.56 |
| LLaVA-Next | Qwen-72B | 32.76 | 23.67 | 28.69 | 34.60 | 23.14 | 30.02 | 28.57 |
| MiniCPM-V 2.5 | Llama3-8B | 33.23 | 16.67 | 31.67 | 20.40 | 26.00 | 28.89 | 25.59 |
| DeepSeek-VL | DeepSeek-LLM-7b-base | 27.10 | 25.44 | 26.02 | 21.60 | 35.71 | 27.63 | 27.17 |
| TextMonkey | Qwen-7B | 31.24 | 11.38 | 30.76 | 19.60 | 26.71 | 27.44 | 23.94 |
| GPT-4o-mini | - | 29.56 | 7.33 | 31.79 | 22.00 | 24.00 | 26.32 | 22.94 |
| Qwen-VL-Chat | Qwen-7B | 27.36 | 15.00 | 27.89 | 24.29 | 27.36 | 26.13 | 24.38 |
| ShareGPT4V-13B | Vicuna-13B | 27.94 | 17.59 | 27.57 | 16.80 | 28.14 | 25.75 | 23.61 |
| LLaVA1.5-13B | Vicuna-13B | 27.52 | 17.33 | 26.25 | 17.00 | 28.66 | 25.45 | 23.35 |
| MiniGPT-v2 | Llama 2-7B-Chat | 26.78 | 19.31 | 27.05 | 14.40 | 29.43 | 25.18 | 23.39 |
| ShareGPT4V-7B | Vicuna-7B | 26.73 | 17.24 | 25.75 | 16.60 | 28.14 | 24.86 | 22.89 |
| LLaVA1.5-7B | Vicuna-7B | 26.36 | 16.67 | 25.75 | 16.60 | 28.14 | 24.64 | 22.70 |

performance on the English version of MME-RealWorld, indicating that they jave been specifically optimized for Chinese data.

2) There is a substantial difference in how models handle Chinese and English data, with some models performing much worse in Chinese scenarios, particularly in reasoning tasks. For instance, GPT-4o and GPT-4o-mini show a performance drop of nearly 10%. However, some models seem to excel in Chinese-related tasks. Notably, models based on `Llama3-8B` generally achieve strong results in both Chinese perception and reasoning tasks, such as SliME and CogVLM2. This suggests that `Llama3-8B` may be an effective LLM backbone for Chinese tasks.

### 3.3 FINE-GRAINED ANALYSIS AND FINDINGS

**Existing Models Still Lacking in Image Detail Perception.** Fig. 3 displays the frequency with which various models choose "E" as their answer. We compare 4 close-source models with the top-level open-source model, InternVL-2. During our annotation process, the frequency of "E" answers does not exceed 5% of the overall data, meaning it represents only a small portion of the total QA pairs. However, nearly all models show a much higher frequency of "E" outputs than the actual number of "E" instances present in our benchmark. This indicates that most models' visual perception modules fail to identify the objects in the images corresponding to our questions.

**Computation Efficiency.** There is a significant disparity in computation efficiency among different models when processing high-resolution images. For example, using models similar to LLMs (e.g., Vicuna-13B), the computational requirements for handling images exceeding 1024×1024 resolution are as follows: LLaVA1.5 requires 16.37 TFLOPS, SliME requires 40.82 TFLOPS, while LLaVA-

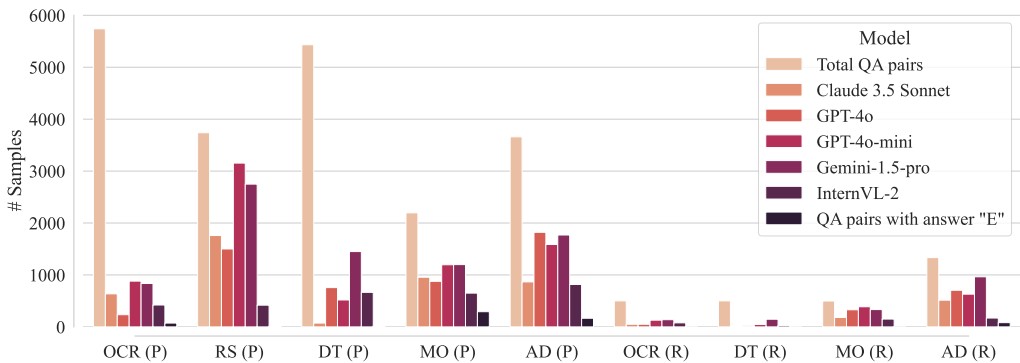

Figure 3: **Frequency of outputting answer "E" for different models** across various domains. The notation in parentheses indicates the task type: P for perception and R for reasoning. The total QA pairs and those with answer "E" are also presented for comparison.

Next and Mini-Gemini-HD require 78.37 and 87.59 TFLOPS, respectively. LLaVA-Next and SliME employ dynamic chunking and encoding of images, while Mini-Gemini-HD uses a higher-resolution vision encoder and significantly increases the number of vision tokens, resulting in a computation cost approximately 5 times that of LLaVA1.5. Additionally, existing methods have inherent limitations in handling high-resolution images. For example, Mini-Gemini-HD resizes images larger than 672×672 to this size, causing a loss of more details. Moreover, we observe interesting phenomena in closed-source models regarding image resolution. For instance, GPT-4o-mini uses over 10,000 tokens for some large images, which is about 10 times more than other closed-source models, although its performance does not significantly surpass other models. Overall, we currently lack methods that can efficiently handle higher resolution images with lower computational overhead.

**Appendix Overview: Supplementary Analyses and Detailed Results.** Due to the space constraints of this paper, we are unable to include some detailed discussions of several important analyses, which we summarize here. First, we explore simple tricks to mitigate the challenges posed by high-resolution images, specifically by segmenting the images into smaller sub-images and querying each one individually. The responses are then aggregated using a majority vote. While this method seems intuitive, it leads to a significant decline in performance, mainly due to the loss of positional context and the risk of splitting fine-grained objects across sub-images. Additionally, we conduct an analysis of the distribution of incorrect choices across various models, revealing distinct response patterns. Larger models show a tendency to select safer options, while smaller models often lean towards the first choice. Finally, we observe that the instruction-following capabilities of some current open-source models are still insufficient, indicating substantial room for improvement. These analyses, along with the performance results for each subtask, are presented in the Appendix Sec. D - Sec. G due to space constraints in the main text.

## 4 CONCLUSION

In this paper, we have introduced MME-RealWorld, a comprehensive benchmark designed to address key limitations in existing evaluations of MLLMs, such as data scale, annotation quality, and task difficulty. As the largest purely human-annotated dataset with the highest resolution to date, MME-RealWorld benefits from the participation of 32 annotators, ensuring high data quality and minimal individual bias. Most QA pairs focus on real-world scenarios, such as autonomous driving and video surveillance, which have significant applicability. Furthermore, we propose MME-RealWorld-CN, a benchmark specifically focused on Chinese scenarios, ensuring that all images and questions are relevant to Chinese contexts. Our evaluation of a wide range of models reveals significant performance gaps, highlighting the current models' shortcomings in complex image perception and underscoring, and the need for further advancements.

ACKNOWLEDGMENTS

This work was supported by National Key R&D Program of China (2022ZD0117901) and National Natural Science Foundation of China (Grant No. 62236010, 62373355)

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

# MME-RealWorld

# ————Appendix————

CONTENTS

## A  RELATED WORK

**Multimodal Benchmark.** With the development of MLLMs, a number of benchmarks have been built. For instance, MME (Fu et al., 2023a) constructs a comprehensive evaluation benchmark that includes a total of 14 perception and cognition tasks. All QA pairs in MME are manually designed to avoid data leakage, and the binary choice format makes it easy to quantify. MMBench (Liu et al., 2023c) contains over $3,000$ multiple-choice questions covering 20 different ability dimensions, such as object localization and social reasoning. It introduces GPT-4-based choice matching to address the MLLM's lack of instruction-following capability and a novel circular evaluation strategy to improve the evaluation robustness. Seed-Bench (Li et al., 2023c) is similar to MME and MMBench but consists of $19,000$ multiple-choice questions. The larger sample size allows it to cover more ability aspects and achieve more robust results. SEED-Bench-2 (Li et al., 2023b) expands the dataset size to $24,371$ QA pairs, encompassing 27 evaluation dimensions and further supporting the evaluation of image generation. MMT-Bench (Ying et al., 2024) scales up the dataset even further, including $31,325$ QA pairs from various scenarios such as autonomous driving and embodied AI. It encompasses evaluations of model capabilities such as visual recognition, localization, reasoning, and planning. Additionally, other benchmarks focus on real-world usage scenarios (Fu et al., 2024c; Lu et al., 2024b; Bitton et al., 2023) and reasoning capabilities (Yu et al., 2024; Bai et al., 2023b; Han et al., 2023). There are also some benchmarks that focus on safety or MLLM reward model evaluation (Zhang et al., 2025) and mathematical problems (Yan et al., 2024). However, there are widespread issues, such as data scale, annotation quality, and task difficulty, in these benchmarks, making it hard to assess the challenges that MLLMs face in the real world.

**MLLMs.** This field has undergone significant evolution (Yin et al., 2023; Fu et al., 2023b), initially rooted in BERT-based language decoders and later incorporating advancements in LLMs. MLLMs exhibit enhanced capabilities and performance, particularly through end-to-end training techniques, by leveraging advanced LLMs such as GPTs (OpenAI., 2023; Brown et al., 2020), LLaMA (Touvron et al., 2023a;b), Alpaca (Taori et al., 2023), PaLM (Chowdhery et al., 2023; Anil et al., 2023), BLOOM (Muennighoff et al., 2022), Mistral (Jiang et al., 2023), and Vicuna (Chiang et al., 2023). Recent model developments, including Flamingo (Awadalla et al., 2023), PaLI (Laurençon et al., 2024), PaLM-E (Driess et al., 2023), BLIP-2 (Li et al., 2023e), InstructBLIP (Dai et al., 2024), Otter (Li et al., 2023a), MiniGPT-4 (Zhu et al., 2023), mPLUG-Owl (Ye et al., 2023), LLaVA (Liu et al., 2023b), Qwen-VL (Bai et al., 2023a),, and VITA series Fu et al. (2025; 2024a); Shen et al. (2025) bring unique perspectives to challenges such as scaling pre-training, enhancing instruction-following capabilities, and overcoming alignment issues. However, the performance of these models in the face of real scenarios has often not been revealed.

**High-resolution MLLMs.** Empirical studies have shown that employing higher resolution is an effective solution for many tasks (Bai et al., 2023a; Liu et al., 2023a; Li et al., 2023f; McKinzie et al., 2024). Approaches like LLaVA-Next (Liu et al., 2024a) segment high-resolution images into multiple patches, encoding each one independently before concatenating all local patch tokens with the original global image tokens, albeit at an escalated computational cost. Other models, such as Monkey (Li et al., 2023f) and LLaVA-UHD (Xu et al., 2024), also split images into patches but subsequently compress them to avoid redundant tokens. Mini-Genimi (Li et al., 2024d) comprises twin encoders, one for high-resolution images and the other for low-resolution visual embedding. They work in an attention mechanism, where the low-resolution encoder generates visual queries, and the high-resolution counterpart provides candidate keys and values for reference. Conv-LlaVA (Ge et al., 2024) employs ConvNeXt instead of ViT as the vision encoder. Cambrian (Tong et al., 2024) uses a set of learnable latent queries that interact with multiple vision features via cross-attention layers. Additionally, SliME (Zhang et al., 2024) stresses the importance of global features, compressing the local image patches twice but preserving all the global context. Although many of these models focus on improving resolution, none have been tested in a rigorous high-resolution benchmark, often providing only intuitive examples that lack informativeness and convincing results. Our proposed benchmark offers a rigorous evaluation to test the capabilities in understanding high-resolution images.

# B  DATA COLLECTION AND TASK SPLIT

## B.1  OCR IN THE WILD

**Data Characteristics.** The data is from real-world street scenes and high-resolution images of product advertisements. Text is dense or difficult to detect and requires careful observation to be identified.

### B.1.1  DATA SOURCES AND ANNOTATION PROCESS

**Data Sources.** We manually select images with complex scenes and recognizable text information from existing high-resolution datasets for our test images. The open-source datasets used include DIV2K and Flickr2K (Agustsson & Timofte, 2017), which offer paired high-resolution RGB images and their corresponding downscaled low-resolution RGB images by a factor of two. In our approach, we exclusively utilize high-resolution images, selecting and preserving images with complex scenes and contexts. Additionally, we include the LIU4K (Liu et al., 2019) dataset, which contains $2,000$ images with resolutions of at least 3K, most ranging from 4K to 6K. This dataset provides abundant materials for testing and evaluating performance on $4K/8K$ display devices, featuring diverse and complex low-level signal distributions and backgrounds. We also incorporate two large-scale Ultra-High-Definition datasets, UHD4K and UHD8K (Zhang et al., 2021), which collectively contain $23,000$ images. These datasets cater to various low-level image enhancement tasks, including image super-resolution (SR), image deraining (Derain), low-light image enhancement (LLIE), and image reflection removal (IRR). Finally, we use HQ-50K (Yang et al., 2023a), a large-scale, high-quality image restoration dataset containing $50,000$ high-quality images. HQ-50K stands out for its large scale, high resolution, varying compression rates, rich texture details, and semantic diversity.

**Annotation.** 20 volunteers annotate the question and answer pairs. 3 experts are tasked with checking and correcting the annotations to ensure quality.

### B.1.2  EVALUATION DIMENSIONS AND BENCHMARK STATISTICS

The evaluation of models in real-world complex scenes involves their ability to recognize and understand text, enabling us to ascertain their capacity to comprehend and process textual information within visual content, thereby enhancing the overall practicality and reliability of intelligent systems. Specifically, Optical Character Recognition (OCR) in complex contexts comprises five perception tasks and two reasoning tasks. For perception tasks,

1. **Contact information and addresses (Fig. 4(a)).** Recognizing telephone numbers, names of countries/cities/streets, and buildings (469 images and 577 QA pairs).

2. **Product and Advertisement Perception (Fig. 4(b)).** Identifying product names/prices or advertisements of shops or brands (803 images and $1,588$ QA pairs).

3. **Identity Information Perception (Fig. 4(c)).** Recognizing license numbers or ID cards of cars/humans (852 QA pairs).

4. **Other kind of Small Text on Signals or Indicators Perception (Fig. 4(d)).** Recognizing small text on indicators, signals, and similar objects (626 images and $1,198$ QA pairs).

5. **Book, Map and Poster Perception (Fig. 4(e)).** Recognizing dialogues/information on posters and specific locations involving a country/region on maps (785 images and $1,555$ QA pairs).

Additionally, our two reasoning tasks include:

1. **Scene Recognition (Fig. 5(a)).** Understanding the meaning of scenes in images, such as predicting the outcome of a game based on the scoreboard or what might happen in the future based on the scene, inferring the time by looking at a clock, or calculating object prices (250 images and 250 QA pairs).

2. **Characters Understanding (Fig. 5(b)).** Understanding the pertinent characteristics of characters in a poster or comic, including their relationships, emotions, intentions, or quantities (250 images and 250 QA pairs).

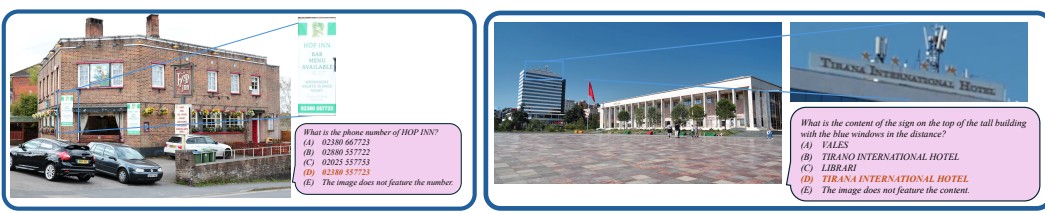

(a) Phone and Address Perception.    (b) Product and Advertisement.

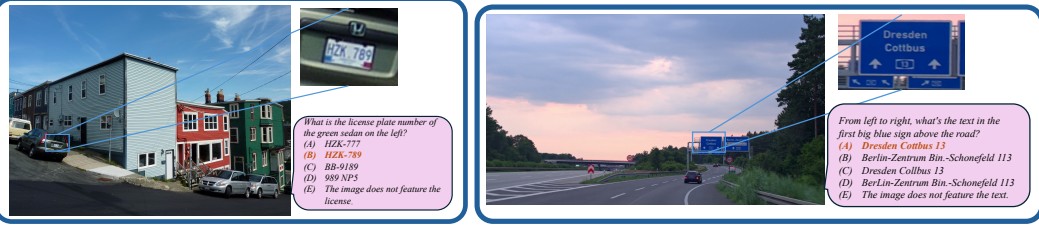

(c) Human/Car License.    (d) Other kind of small Text on Signals or Indicators.

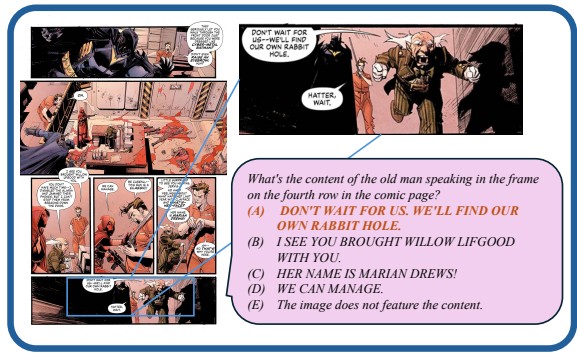

(e) Book, Map and Poster.

Figure 4: **Data Examples for Perception Tasks in OCR in the Wild**

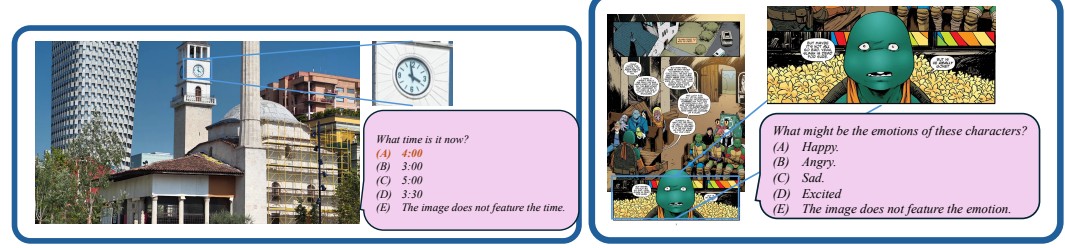

(a) Scene Recognition.    (b) Characters Understanding.

Figure 5: **Data Examples for Reasoning Tasks in OCR in the Wild**

Note that although we have 3,293 unique images, some tasks use overlapping image sets, so the total number of images listed in all the tasks is not exactly 3,293.

## B.2 DIAGRAM AND TABLE

**Data Characteristics.** Diagrams and tables with rich content present significant challenges for rapid localization and analysis, even for human researchers. These tasks demand a high level of perceptual capability.

### B.2.1 DATA SOURCES AND ANNOTATION PROCESS

Although there are existing datasets for evaluating diagrams and tables, such as ChartQA (Masry et al., 2022) and some open-source scientific chart data like Arxiv QA (Li et al., 2024c), we observe that these datasets often have relatively low image resolutions and limited content richness. Consequently, they are relatively easy for humans to interpret quickly, which does not align with the design goals of our benchmark. To address this, we source complex diagram data from the internet, such as detailed financial reports with large charts. Analyzing these large charts poses significant perceptual challenges, even for humans, and thus better aligns with the objectives of our benchmark.

**Annotation.** 20 volunteers are involved to generate question and answer pairs for the perception task. Additionally, one expert researcher is responsible for generating reasoning annotations. To ensure high-quality annotations, three experts are assigned to review and correct the annotations.

### B.2.2 EVALUATION DIMENSIONS AND BENCHMARK STATISTICS

The ability of multimodal models to perceive and understand diagram and table data has long been a focus of research. In our Diagram and Table domain, we have elevated the difficulty level to a point where even humans find it challenging to solve easily. We have collected $2,570$ images and $5,933$ annotations, categorizing the annotations into the following four types:

1. **Table Perception (Fig. 6(a)).** Identifying specific elements within a table by using the given table name, horizontal axis coordinates, and related location information to determine the value of elements in specific positions ($4,018$ QA pairs).

2. **Diagram Perception (Fig. 6(b)).** Identifying specific elements within a diagram by using the provided legend or title, along with specific location information, to determine the value of elements or the intervals they belong to ($1,415$ QA pairs).

Additionally, our two reasoning tasks include:

1. **Table Reasoning (Fig. 6(c)).** This involves tasks that go beyond simple perception, such as comparing the values of two elements in specific positions within a table, filtering the table based on given conditions, or determining the maximum and minimum values ($174$ QA pairs).

2. **Diagram Reasoning (Fig. 6(d)).** Similar to table reasoning, but reasoning with diagrams involves distinguishing specific colors in the legend and assessing the height of curves or bars ($326$ QA pairs).

## B.3 REMOTE SENSING

**Data Characteristics.** From real remote sensing data, some images have extremely high quality, with individual image sizes reaching up to 139MB and containing rich details.

### B.3.1 DATA SOURCES AND ANNOTATION PROCESS

We select high-resolution images from public remote sensing datasets with rich information. For example, the FAIR1M dataset (Sun et al., 2021) focuses on fine-grained object recognition and detection using high-resolution ($0.3 - 0.8$m) RGB images from Gaogen (GF) satellites extracted via Google Earth. It contains $15,000$ images annotated with rotated bounding boxes across 5 main categories (ships, vehicles, airplanes, courts, and roads), further divided into 37 sub-categories. The Potsdam dataset[6] dataset includes 38 patches of true orthophotos (TOP) extracted from larger mosaics. VGoogle (Hou et al., 2019), VBing (Hou et al., 2019), and VArcGIS (Hou et al., 2019) datasets, derived from Google Earth, Bing World Imagery, and ArcGIS World Imagery respectively,

---

[6]https://paperswithcode.com/dataset/isprs-potsdam

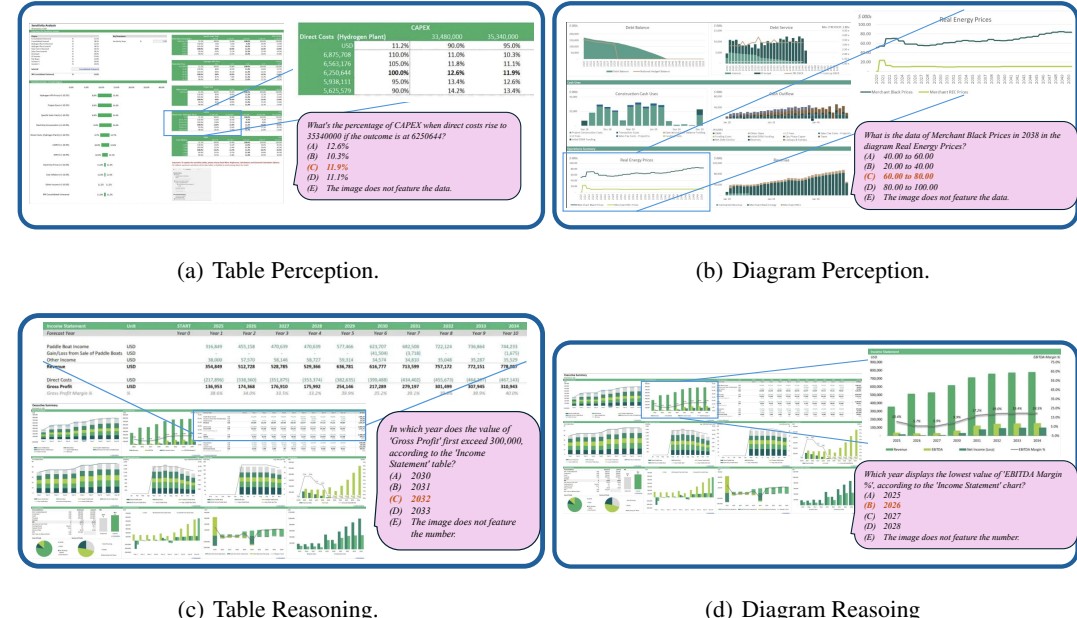

(a) Table Perception.

(b) Diagram Perception.

(c) Table Reasoning.

(d) Diagram Reasoing

Figure 6: **Data Examples for Diagram and Table Tasks**

each feature 38 classes with a total of approximately $59,000$ images per dataset. Each class contains at least $1,500$ images, with spatial resolutions ranging from $0.07$ to $38.22$ meters.

**Annotation.** For all the questions in this subsection, 20 volunteers manually create the questions and answers, while another expert reviews the quality of the questions to ensure they meet the required standards.

### B.3.2 EVALUATION DIMENSIONS AND BENCHMARK STATISTICS

Remote sensing images have a wide range of applications in real-world scenarios. During the construction of our dataset, we observe that many tasks are challenging for humans. For example, counting the number of airplanes in Fig. 7(a) requires careful observation and counting by human annotators. Automating this process with multimodal large models would be highly valuable for remote sensing applications. We select a total of $1,298$ high-quality images and design three specific tasks tailored for remote sensing images:

1. **Object Counting (Fig. 7(a)).** Task involves counting specific objects such as airplanes, ships, or buildings within a given image ($1,255$ QA pairs).

2. **Color Recognition (Fig. 7(b)).** Task involves identifying and describing the colors of specific objects in the image ($1,226$ QA pairs).

3.**Spatial Relationship Understanding (Fig. 7(c)).** Understanding both the absolute spatial relationships and relative spatial relationships between objects in the images ($1,257$ QA pairs).

### B.4 AUTONOMOUS DRIVING

**Data Characteristics.** The front-view driving datas are recorded using onboard cameras with various sensor configurations. The images encompass diverse weather conditions (e.g., sunny, night, rainy, etc.), geographic locations (e.g., US, SG, CN), and complex traffic scenarios (e.g., urban, highway, etc.).

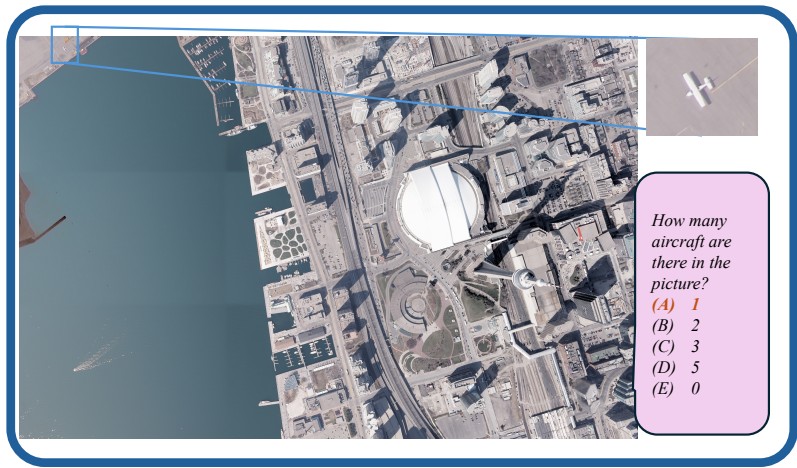

(a) Object Counting.

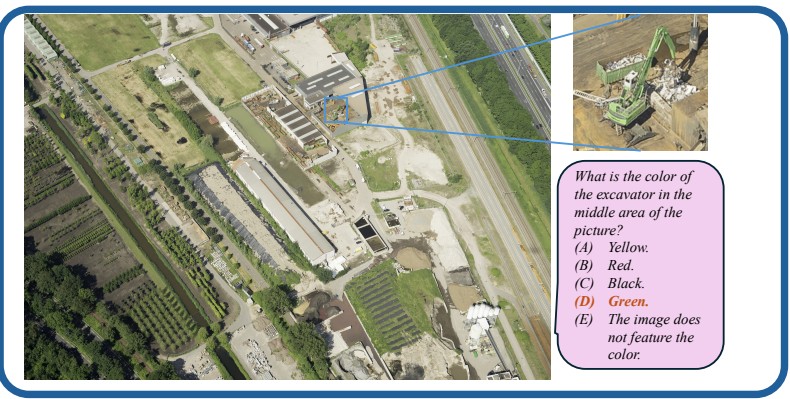

(b) Color Recognition.

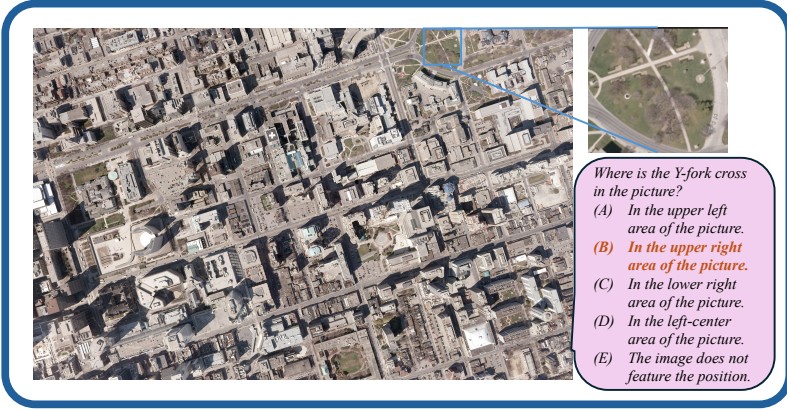

(c) Spatial Relationship Understanding.

Figure 7: **Data Examples for Perception Tasks in Remote Sensing**

### B.4.1 DATA SOURCES AND ANNOTATION PROCESS

**Data Sources.** We select high-quality images from large open-source driving datasets, each with distinct advantages. The Rank2Tell dataset (Sachdeva et al., 2024) ranks the importance level of surrounding objects for driving safety. Additionally, it provides dense annotations of semantic, spatial,

and relational attributes with bounding boxes for approximately $2,600$ frames captured at intersections, and it stitches images from three cameras to deliver a wide field of view (FOV). To enhance the reliability of autonomous driving systems, the CODA dataset (Li et al., 2022) collects $1,500$ driving scenes, each containing object-level corner cases, and labels more than 30 novel categories (e.g., garbage bag, concrete block, etc.). It focuses on evaluating performance of perception systems in detecting out-of-distribution (OOD) objects compared to common traffic elements. The nuScenes dataset (Caesar et al., 2020), one of the most popular real-world autonomous driving datasets, provides abundant 3D perception annotations with a semantic map and CAN bus expansion (Li et al., 2023d). Based on nuScenes (Caesar et al., 2020), DriveLM-nuScenes (Sima et al., 2023) links approximately $4,800$ key frames with driving behaviors and motions by formulating 3P reasoning (perception, prediction, planning) as a series of rich question-answer pairs in a directed graph.

**Annotation.** For all the questions in this subsection, a professional researcher manually generates the questions and answers based on the source datasets' labels, achieving their non-ambiguity, challenge and complexity. Another expert reviews the quality of the questions to ensure they meet the required standards.

### B.4.2 EVALUATION DIMENSIONS AND BENCHMARK STATISTICS

Vision-centric autonomous driving is one of the most significant applications of artificial intelligence. However, unresolved issues remain, including both object-level and task-level corner cases, as well as safe-critical and human-like planning (Yang et al., 2023b). MLLMs with general knowledge and the ability of driving scenarios embodied understanding (Gao et al., 2024; Zhou et al., 2024) are seen as a promising solution to achieve Level 4 autonomous driving. Specifically, we have designed three main perception tasks and three main reasoning tasks, which are further subdivided into a total of fifteen sub-tasks. It is worth noting that, as traditional detection tasks in autonomous driving have largely been addressed by modern perception models, our focus is shifting towards perception challenges involving small or distant objects, specifically those that occupy less than $1/100$ of the total image area. Meanwhile, LLMs must possess extensive driving expertise and even a deep understanding of 3D spatial concepts in order to effectively address the complex reasoning challenges. For perception tasks:

1. **Object Identification (Fig. 8(a)).** Describing the main traffic elements in front of the ego car including their categories and corresponding quantities ($1,101$ images and $1,101$ QA pairs).

2. **Object Attribute Identification.** Task involves identifying the attribute of a specific object according to its appearance and location (a total of 454 images and 523 QA pairs), and describing the attributes of all objects within a specific category (a total of $1,167$ images and $1,315$ QA pairs) in traffic scenarios. In terms of sub-tasks, the former includes the visual attribute of a traffic signal (Fig. 8(e), 157 images and 201 QA pairs) and the motion attribute of a pedestrian (Fig. 8(f), 152 images and 164 QA pairs) or a vehicle (145 images and 158 QA pairs), and the latter includes the motion attributes of multiple pedestrians (Fig. 8(c), 493 images and 492 QA pairs) or vehicles (Fig. 8(d), 674 images and 823 QA pairs).

3. **Object Counting (Fig. 8(b)).** Counting special traffic elements in the given image, such as cars, trucks, traffic signals, etc., especially some novel objects compared to traditional autonomous driving tasks such as garbage bags, dogs, concrete blocks, etc. (647 images and 720 QA pairs).

Furthermore, reasoning tasks are as follows:

1. **Intention Prediction.** Task involves predicting the intention of a designated traffic agent in the given image (a total of 582 images and 614 QA pairs). In terms of sub-tasks, it contains fine-grained behavior prediction of the ego vehicle (Fig. 9(a), 304 images and 304 QA pairs) and future intention of a pedestrian (95 images and 103 QA pairs) or a vehicle (Fig. 9(b), 183 images and 207 QA pairs).

2. **Interaction Relation Understanding.** Task involves reasoning the interaction relation between two specific traffic elements (a total of 444 images and 513 QA pairs). In terms of sub-tasks, it contains the ego vehicle's reaction to a specific object (Fig. 9(e)), which is further categorized into three categories: pedestrian (102 images and 106 QA pairs), vehicle (95 images and 101 QA pairs), and traffic signal (81 images and 105 QA pairs). Additionally, another sub-task is predicting the interactions between the aforementioned objects, excluding the ego vehicle (Fig. 9(d), 166 images and 201 QA pairs).

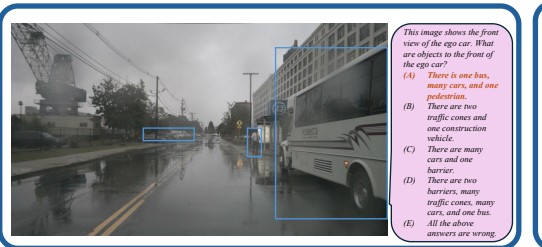
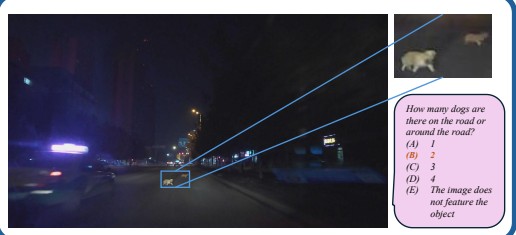

(a) Object Identification.

(b) Object Counting.

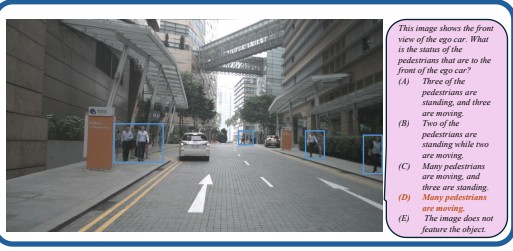
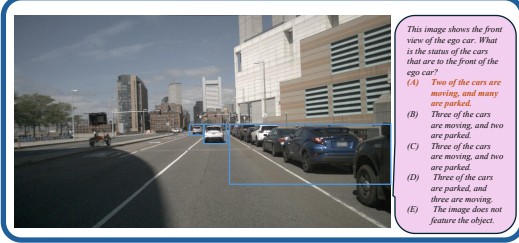

(c) Motion Attribute Identification of Multiple Pedestri-(d) Motion Attribute Identification of Multiple Vehicles.
ans.

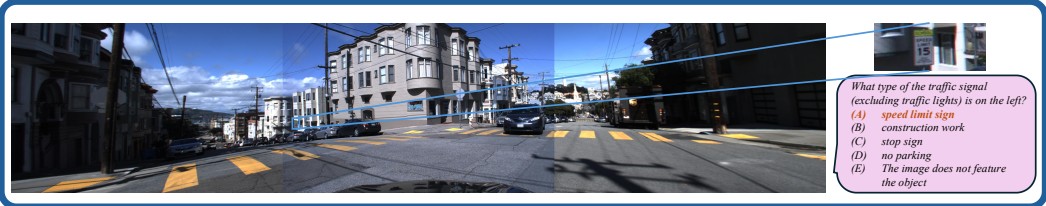

(e) Visual Attribute Identification of a Specific Traffic Signal

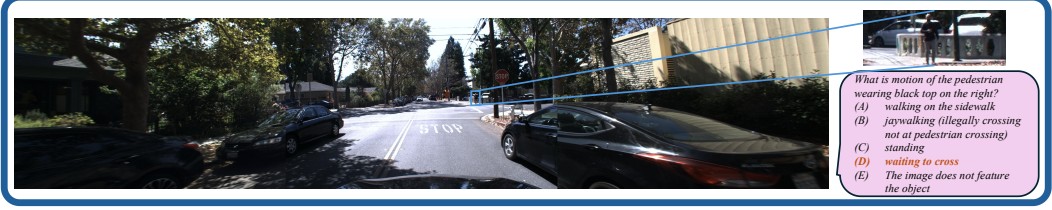

(f) Motion Attribute Identification of a Specific Pedestrian

Figure 8: **Data Examples for Perception Tasks in Autonomous Driving**

3. **Driver Attention Understanding (Fig. 9(c)).** Reasoning the traffic signal that the driver should pay attention to in the given front view image, such as yellow light, speed limit sign, no parking sign, etc. (217 images and 217 QA pairs).

### B.5 MONITORING

#### B.5.1 DATA SOURCES AND ANNOTATION PROCESS

**Data Characteristics.** Monitoring images are captured from different cameras (e.g., drone-equipped cameras, fixed surveillance cameras, infrared cameras), viewpoints (arbitrary and fixed viewpoints), scene complexities (e,g., streets, shopping malls, intersections, campus, etc.), and environmental factors (day and night).

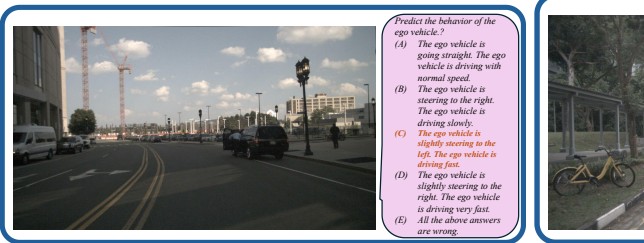 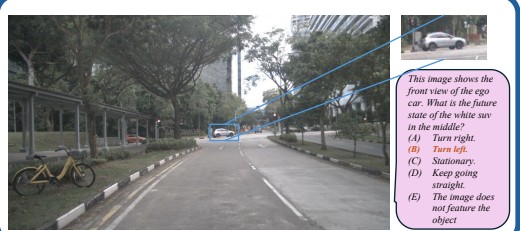

(a) Intention Prediction of Ego Vehicle.       (b) Intention Prediction of a Specific Vehicle.

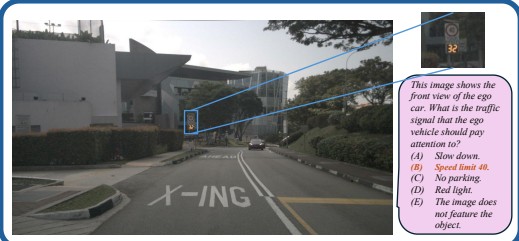 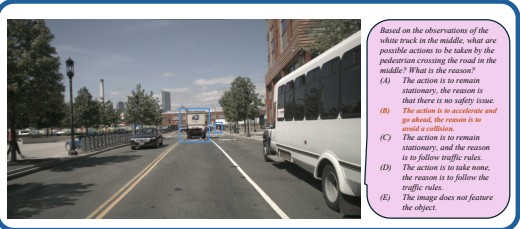

(c) Driver Attention Understanding       (d) Interaction Relation between Traffic Elements

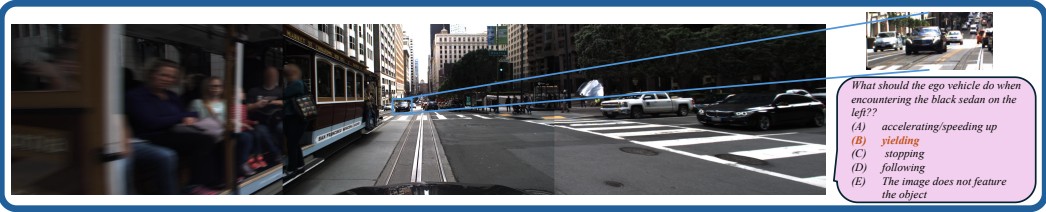

(e) Interaction Relation between Ego Vehicle and Traffic Elements

Figure 9: **Data Examples for Reasoning Tasks in Autonomous Driving**

We select high-resolution images from public monitoring image datasets with many real-world challenges. For example, the VisDrone dataset (Zhu et al., 2021) brings several challenges, e.g., viewpoint variations, scale variations, and out-of-view, etc. Additionally, its dataset contains 263 video clips with 179, 264 frames and 10, 209 static images, which are captured via various drone-equipped cameras across various categories (e.g., pedestrian, people, bicycle, car, van, truck, tricycle, awningtricycle, bus, and motor), density (e.g., sparse and crowded scenes) and environments (e.g., urban and rural regions). Additionally, The second dataset [7], is collected in diverse environments (e.g., street, mall, elevator, etc.) for crowd density prediction task, features 3, 000 images with only person category, captured by fixed surveillance cameras. This dataset is highly diverse from the camera viewpoints (low altitude, high altitude, fisheye, etc.), scale size, and scene complexities. The LLVIP dataset (Jia et al., 2021), which is a visible-infrared paired dataset for low-light vision, contains 30, 976 images taken in binocular cameras, and contains a large number of pedestrians. We only select manually infrared images from it to test the model's robustness on different modals.

**Annotation.** For all the questions in this subsection, two professional researchers manually create the questions and answers, and another expert reviews the quality of the questions to ensure they meet the required standards.

---

[7]This dataset is publicly accessible through the AI Studio website at https://aistudio.baidu.com/datasetdetail/28831.

### B.5.2 Evaluation Dimensions and Benchmark Statistics

Monitoring images are widely applied in real-world scenarios to increase public safety. Analyzing the monitoring images accurately with MLLMs would be highly valuable for public safety and crowd management. Specifically, we have designed three main tasks for monitoring images:

1. **Object Counting (Fig. 10(a)).** Task involves counting specific objects such as pedestrians, cars, or trucks in the given monitoring images (a total of $1,600$ images and $1,600$ QA pairs). Noted that, when the count of a specific object is equal to zero, this object counting task can be transformed well into the object existence task (Fig. 10(b), for judging whether a specific object exists in the given images. Thus, the object existence task can be regarded as a special case of the counting task. Additionally, we categorize this task into two sub-tasks for vehicle counting (608 images and 608 QA pairs) and person counting (992 images and 992 QA pairs), respectively.

2. **Object Location (Fig. 10(c)).** Task involves judging the location of the specific vehicles, like cars, or trucks in the given monitoring images (a total of 136 images and 136 QA pairs).

3. **Attribute Recognition.** Task involves identifying and describing the attributes of specific objects, e.g., *color recognition (Fig. 10(d))* and *orientation perception (Fig. 10(e))*, in the monitoring images (a total of 460 images and 460 QA pairs). Additionally, we categorize this task into two sub-tasks for the vehicle (352 images and 352 QA pairs) and person attribute recognition tasks (108 images and 108 QA pairs), respectively.

In addition, there are three seasoning tasks described as follows.

1. **Calculate the Sum of Different Objects (Fig. 11(a)).** Counting various objects and calculating their total number accurately (300 images and 300 QA pairs).

2. **Intention Reasoning (Fig. 11(b)).** Reasoning the next route and turn of the specific object (98 images and 98 QA pairs).

3. **Attribute Reasoning (Fig. 11(c)).** Reasoning the specific materials and functions of the given objects, such as inferring the function of the dustbin via its appearance (100 images and 100 QA pairs).

## C  Discussion on Data Diversity, Scalability, and Limitations

The creation of a diverse and scalable dataset poses significant challenges, particularly in the context of high-resolution, real-world scenarios. This section discusses our rationale for domain selection, identifies current limitations in dataset diversity and scalability, and explores the potential for leveraging models to enhance scalability. By addressing these key aspects, we aim to provide a comprehensive overview of our approach and outline possible avenues for improvement in the future.

### C.1  Advancing Dataset Diversity and Scalability: Challenges and Model-Assisted Strategies

The creation of a high-resolution, diverse, and scalable dataset is a multifaceted challenge that involves balancing domain-specific requirements, practical constraints, and innovative approaches to enhance efficiency. In this section, we summarize the key aspects of our work, including the rationale behind our domain selection, existing limitations in dataset construction, and the potential for model-assisted strategies to improve scalability.

**1. Rationale for Domain Selection (Appendix. C.2):** We prioritized domains such as remote sensing, surveillance, and autonomous driving for their practical value and unique challenges, focusing on high-resolution imagery with complex scenarios. These domains are better suited for testing nuanced perception and reasoning compared to simpler datasets like COCO, which lack scene complexity and high resolution.

**2. Current Limitations and Plans for Extension (Appendix. C.3):** Our dataset faces challenges in task diversity and scalability. There is a lack of high-resolution natural scene data and underrepresentation of domains like indoor scenes, healthcare, and AR/VR. Additionally, the dataset construction

process, requiring significant human effort, limits scalability. Future plans include capturing natural images, expanding to more domains, and exploring strategies to reduce manual effort.

**3. Exploring Model-Assisted Approaches to Enhance Scalability (Appendix. C.4):** We trialed MLLMs for data filtering and question generation. While models like GPT-4o effectively filtered images, their performance in generating complex QA pairs was suboptimal, often producing lower task difficulty and higher error rates compared to manual annotation. This suggests that while model-assisted pipelines can reduce workload, further refinement is needed to match the quality of manual processes.

### C.2 Why We Select Domains Such As Remote Sensing And Autonomous Driving Instead of Others Like Embodied AI or Natural Images in COCO

**Practical Value:** Instead of opting for data sources like COCO, our primary objective is to enable the understanding of high-resolution, real-world scenarios that are closely tied to domains such as autonomous driving and surveillance. Recent studies have demonstrated a growing application of MLLMs in these realistic fields, which pose unique challenges to their perception and reasoning abilities. The selected domains specifically demand high image resolution, enabling MLLMs to showcase progress in these settings. As a result, we prioritize areas where MLLMs are currently poised to make a significant impact.

**Benchmark Objective:** Our benchmark is not limited to autonomous driving or surveillance applications. The core goal is to evaluate MLLMs' fundamental perception and reasoning capabilities within these domains. The tasks we design include various scenarios that researchers have identified as appropriate for assessment through multiple-choice QA formats. We deliberately avoid more specific or complex tasks, such as robotic arm trajectory prediction, to maintain a focus on fundamental capabilities.

**Task Difficulty:** We aim to emphasize high-resolution imagery and focus on challenging and subtle objects. In contrast, embodied AI scenarios often feature large, static objects, which pose relatively limited challenges to perception tasks. Similarly, while COCO indeed provides a large number of samples, its images are relatively small in resolution, with insufficient complexity in the scenes. For instance, many COCO images feature only a single object, making it difficult to create sufficiently challenging questions.

This limitation is one of the primary reasons we did not select COCO as a major data source. Covering all the scenes and categories in COCO would require capturing images ourselves using cameras in complex environments to generate sufficiently challenging, high-resolution perception tasks. This approach would undoubtedly be resource-intensive and difficult to scale, making it less feasible for our benchmark objectives.

### C.3 Existing Limitations and Plans for Extension

At present, the main limitations of our work are concentrated in the diversity of tasks and the scalability of the dataset.

1. **Perception of Natural Scenes:** Although we discussed in Section C.2 some data sources that were excluded during the dataset construction process, such as COCO, which were deemed too simple for high-resolution perception tasks, there remain challenges in finding sufficiently complex, high-resolution natural images containing small objects on the internet. This limitation has resulted in a lack of natural scene data in our dataset, which may hinder the evaluation of models' question-answering capabilities related to natural landscapes. To address this issue, we plan to collect natural scene data through self-captured images or generate complex high-resolution images using generative models in future work, thereby compensating for this shortcoming.

2. **Extension to Other Domains:** Beyond domains like remote sensing, surveillance, and autonomous driving, which have high-resolution requirements and demand understanding of complex scenes, there are other practical applications such as indoor scene understanding, commerce, health-care, robotics, and AR/VR that also share similar needs. Due to cost constraints, our current work does not encompass all these domains, leading to potential bias in the dataset. For example, surveillance and autonomous driving data are primarily human-centric, which might lack evaluations of

other critical factors. In future work, we aim to further explore these mentioned domains, identify suitable images, and annotate them to enhance the diversity of our dataset.

3. **Weak Scalability:** Our dataset construction involves two key stages requiring human involvement.

- **Stage 1: Image Selection and Filtering:** During this stage, we ensure: - The images contain challenging or valuable small objects. - The images are of high quality, free from glare or noise. - The scenes are clear enough to minimize ambiguity in QA tasks (e.g., if a question asks about a person in a blue shirt in the upper right corner, there should be only one such person).

By ensuring high-quality and challenging images, we provide annotators with images more likely to yield usable QA pairs, enhancing annotation efficiency.

- **Stage 2: Human Annotation:** This involves crafting questions and answers for the complex images. The human cost throughout this process is relatively high, which makes it challenging to scale to other languages or data scenarios. To mitigate this issue, we have discussed in Section C.4 some promising alternative strategies, such as leveraging MLLMs to reduce the reliance on human labor.

## C.4 EXPLORING MODEL-ASSISTED APPROACHES TO ENHANCE DATASET SCALABILITY

To explore whether models can assist in improving the scalability of our dataset construction process, we conducted a small-scale trial using the following approach:

**1. Data Filtering:** We implemented a model-based strategy to streamline the data filtering process. A minimum resolution threshold of $1024 \times 1024$ was set to ensure the images were sufficiently large. We employed a MLLM to remove images that were noisy or unclear. The MLLM was prompted with the following instruction:

> "We provide an image; please score the scene from two aspects on a scale of [1, 2, ..., 10] and provide Objects with Observational Challenges:
>
> 1. **Complexity:** Assess whether the image contains a large number of elements, such as various objects. If the scene features only one prominent object and the foreground/background is relatively simple, the score should be 0. Conversely, if the image includes a multitude of elements and the scene is complex, the score should be higher.
>
> 2. **Object Salience:** Evaluate whether the image contains small objects that are difficult to observe clearly, specifically objects occupying less than 1/20 of the total pixel area. If all objects in the image can be observed very clearly or are relatively large, the score should be 0.
>
> 3. **Objects with Observational Challenges:** If there are objects in the image that are difficult to observe or smaller targets, please list these objects and their locations."

Using this scoring strategy, annotators could easily reference the model's assessments of image complexity and the presence of small objects. Currently, we are utilizing GPT-4o as the scoring model. In the future, a voting mechanism involving multiple models could be employed for improved quality assurance. This approach has successfully reduced the workload for annotators, enabling us to quickly filter out simple images.

**2. Manual Annotation:** We explored using GPT-4o for question generation and answer construction. However, the results were suboptimal for high-resolution scenarios, especially in reasoning tasks. The generated samples were either too easy or contained errors, rendering them unusable.

To address this, we tested multiple models—Qwen2-vl-72b, Llava-ov-72b, Claude3.5-sonnect, and GPT-4o. Each model independently generated three questions and answers for a single image, following the construction standards outlined in our article. Predefined prompts were provided to ensure consistency. Human experts then reviewed and retained the most challenging and reasonable questions from the model outputs.

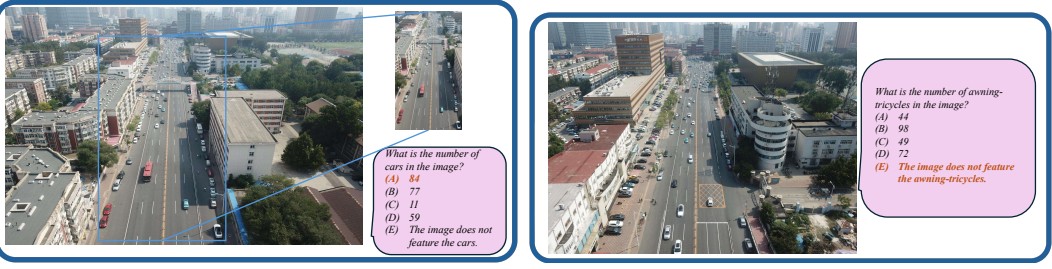

(a) Object Counting in Monitoring Images.   (b) Object Existence in Monitoring Images.

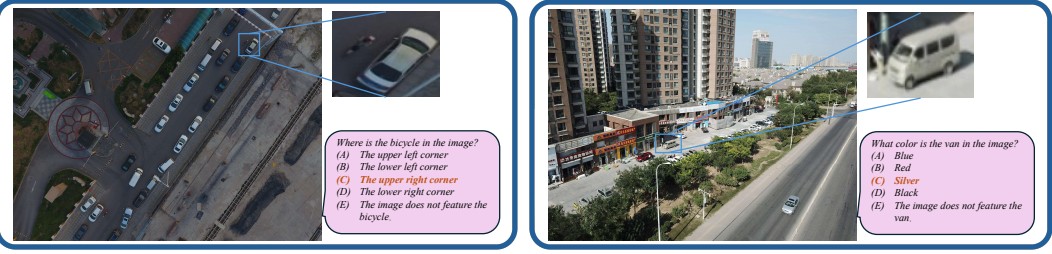

(c) Object Location in Monitoring Images.   (d) Color Recognition in Monitoring Images.

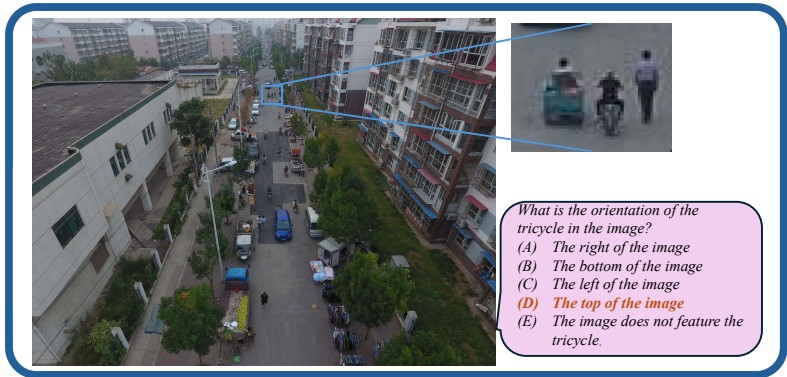

(e) Orientation Perception in Monitoring Images.

Figure 10: **Data Examples for Perception Tasks in Monitoring Images**

While this method allowed us to obtain some high-quality QA pairs, our small-scale experiments revealed that: - The task difficulty generated by models was generally lower than that of purely manual annotation. - The error rate in model-generated samples was significantly higher. - Manually reviewing multiple model outputs did not substantially reduce the time required compared to direct manual annotation.

**Observations and Future Directions:** Our findings suggest that, for ultra-high-resolution perception tasks, existing model-assisted pipelines are not yet optimal. The limitations include reduced task complexity and higher error rates compared to purely manual processes. Additionally, the time efficiency of manual review was not significantly improved. We believe this represents a valuable direction for future research.

## D EXPERIMENTAL RESULTS ON ALL TASK SPLITS

**OCR in The Wild Performance.** Tab. 6 displays the performance of various models on real-world OCR tasks. Generally speaking, when image resolution is high, the more advanced models still

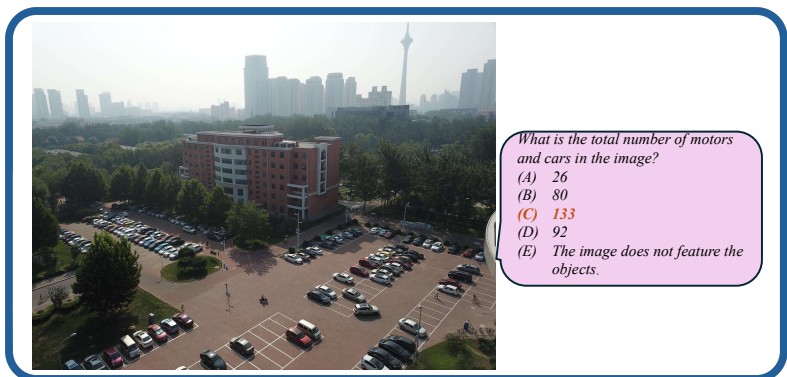

(a) Reasoning Task of Calculating the Sum of Different Objects in Monitoring Images.

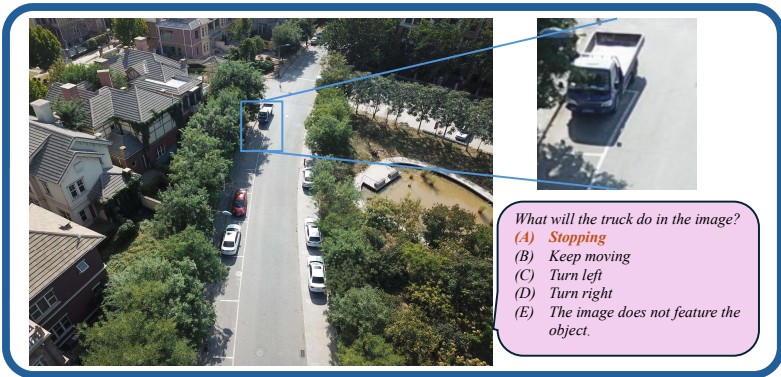

(b) Reasoning task of Intention of the Special Object in Monitoring Images.

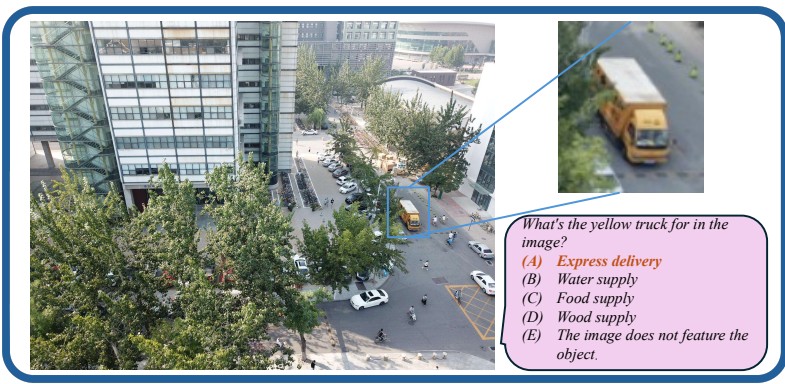

(c) Reasoning task of Attribute of the Special Object in Monitoring Images.

Figure 11: **Data Examples for Reasoning Tasks in Monitoring Images**

demonstrate commendable OCR capabilities. However, this does not imply that our task is of low difficulty. The average accuracy rates of Qwen-VL and the basic LLaVA model on perception tasks are only slightly better than random guessing. In this task, the gap between open-source and closed-source models is not significant. GPT-4o ranks first in overall performance, while Claude 3.5 Sonnet leads in reasoning tasks.

**Diagram and Table.** Tab. 7 presents the results for the diagram domain, where some of the more advanced models perform relatively well, with four models achieving an average accuracy of over

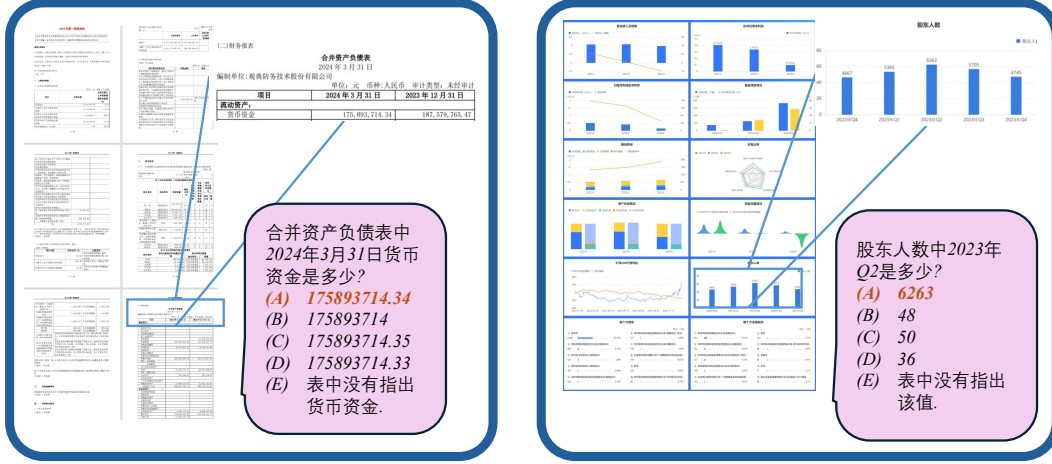

Figure 12: **Data Examples for Perception Tasks in MME-RealWorld-CN**

Table 6: **Experimental results on the `OCR in the Wild` tasks** are categorized as follows: "Product" represents products and advertisements; "B & M & P" represents books, maps, and posters; "Contact" denotes contact information and addresses; "Identity" pertains to identity information; and "Signage" refers to signage and other text. Models are ranked according to their average performance on perception tasks, from highest to lowest. Rows corresponding to proprietary models are highlighted in gray for distinction.

| Method | LLM | Perception | | | | | | | Reasoning | | | |
|---|---|---|---|---|---|---|---|---|---|---|---|---|
| | | Product | B & M & P | Contact | Identity | Signage | Avg | Avg-C | Scene | Character | Avg | Avg-C |
| Qwen2-VL | Qwen2-7B | 81.32 | 82.64 | 84.40 | 84.51 | 76.13 | 81.38 | 81.80 | 63.20 | 63.60 | 63.40 | 63.40 |
| GPT-4o | - | 79.65 | 79.23 | 74.88 | 73.66 | 77.38 | 77.69 | 76.96 | 64.80 | 58.00 | 61.40 | 61.40 |
| InternVL-2 | InternLM2.5-7b-Chat | 72.21 | 80.58 | 72.10 | 73.47 | 68.70 | 73.92 | 73.41 | 56.00 | 58.80 | 57.40 | 57.40 |
| Claude 3.5 Sonnet | - | 71.89 | 83.67 | 61.15 | 64.64 | 69.70 | 72.47 | 70.21 | 62.60 | 61.20 | 61.90 | 61.90 |
| InternVL-Chat-V1-5 | InternLM2-Chat-20B | 69.83 | 75.56 | 71.75 | 73.36 | 67.03 | 71.51 | 71.51 | 57.60 | 56.00 | 56.80 | 56.80 |
| CogVLm2-llama3-Chat | LLama3-8B | 70.35 | 66.82 | 74.00 | 76.41 | 67.03 | 69.97 | 70.92 | 58.80 | 49.20 | 54.00 | 54.00 |
| Mini-Gemini-34B-HD | Nous-Hermes-2-Yi-34B | 72.14 | 79.04 | 64.30 | 55.16 | 66.61 | 69.55 | 67.45 | 60.80 | 57.60 | 59.20 | 59.20 |
| InternLM-XComposer2.5 | InternLM2-7B | 65.34 | 71.83 | 66.90 | 78.29 | 65.69 | 69.25 | 69.61 | 50.40 | 56.40 | 53.40 | 53.40 |
| Gemini-1.5-pro | - | 65.92 | 66.37 | 74.41 | 70.19 | 66.36 | 67.62 | 68.65 | 56.60 | 48.80 | 52.70 | 52.70 |
| MiniCPM-V 2.5 | LLama3-8B | 64.69 | 69.52 | 71.58 | 68.90 | 62.19 | 66.79 | 67.38 | 48.80 | 39.20 | 44.00 | 44.00 |
| Cambrian-1-34B | Nous-Hermes-2-Yi-34B | 67.65 | 77.30 | 65.20 | 50.47 | 64.19 | 66.45 | 64.37 | 54.00 | 56.00 | 55.00 | 55.00 |
| GPT-4o-mini | - | 62.32 | 68.87 | 54.11 | 62.56 | 58.51 | 62.51 | 61.27 | 52.80 | 41.20 | 47.00 | 47.00 |
| Cambrian-1-8B | LLama3-8B-Instruct | 59.18 | 67.27 | 55.98 | 52.93 | 52.25 | 58.68 | 57.52 | 52.80 | 53.60 | 53.20 | 53.20 |
| Monkey | Qwen-7B | 55.58 | 54.47 | 55.38 | 59.98 | 50.42 | 54.63 | 54.77 | 32.40 | 22.00 | 27.20 | 27.20 |
| SliME-8B | LLama3-8B | 55.97 | 57.30 | 41.25 | 55.05 | 49.92 | 53.45 | 51.90 | 55.60 | 50.80 | 53.20 | 53.20 |
| mPLUG-DocOwl 1.5 | LLaMA-7B | 54.62 | 52.22 | 59.10 | 63.03 | 32.99 | 51.15 | 52.39 | 46.80 | 38.40 | 42.60 | 42.60 |
| SliME-13B | Vicuna-13B | 52.25 | 46.50 | 46.97 | 53.76 | 53.17 | 50.58 | 50.53 | 45.60 | 36.40 | 41.00 | 41.00 |
| DeepSeek-VL | DeepSeek-LLM-7b-base | 53.72 | 55.06 | 31.72 | 44.13 | 49.42 | 49.55 | 46.81 | 48.80 | 41.60 | 45.20 | 45.20 |
| LLaVA-Next | LLama3-8B | 50.77 | 49.45 | 38.30 | 38.73 | 53.51 | 47.94 | 46.15 | 58.00 | 52.40 | 55.20 | 55.20 |
| YI-VL-34B | Yi-34B-Chat | 48.33 | 50.55 | 33.97 | 37.91 | 43.57 | 44.95 | 42.87 | 46.80 | 38.00 | 42.40 | 42.40 |
| ShareGPT4V-13B | Vicuna-13B | 46.92 | 40.51 | 40.38 | 46.01 | 47.66 | 44.55 | 44.30 | 31.20 | 20.80 | 26.00 | 26.00 |
| LLaVA1.5-13B | Vicuna-13B | 45.51 | 40.39 | 39.69 | 47.54 | 46.74 | 44.10 | 43.97 | 36.80 | 23.60 | 30.20 | 30.20 |
| Mini-Gemini-7B-HD | Vicuna-7B-v1.5 | 40.05 | 44.44 | 42.98 | 44.37 | 39.32 | 42.02 | 42.23 | 38.00 | 32.80 | 35.40 | 35.40 |
| ShareGPT4V-7B | Vicuna-7B | 40.50 | 35.43 | 37.61 | 42.14 | 41.99 | 39.39 | 39.53 | 25.60 | 22.70 | 24.15 | 24.15 |
| MiniGPT-v2 | Llama 2-7B-Chat | 40.05 | 34.73 | 38.99 | 41.08 | 41.82 | 39.02 | 39.33 | 36.40 | 23.60 | 30.00 | 30.00 |
| LLaVA1.5-7B | Vicuna-7B | 39.67 | 34.92 | 37.26 | 40.26 | 41.90 | 38.69 | 38.80 | 30.80 | 21.20 | 26.00 | 26.00 |
| TextMonkey | Qwen-7B | 38.12 | 31.96 | 44.89 | 45.19 | 33.89 | 37.30 | 38.81 | 36.00 | 24.80 | 30.40 | 30.40 |
| LLaVA-Next | Qwen-72B | 43.58 | 45.72 | 20.28 | 14.91 | 41.24 | 37.07 | 33.15 | 17.60 | 16.80 | 17.20 | 17.20 |
| Qwen-VL-Chat | Qwen-7B | 32.73 | 37.62 | 27.38 | 33.22 | 26.88 | 32.37 | 31.57 | 36.40 | 20.80 | 28.60 | 28.60 |

Table 7: **Experimental results on the `Diagram and Table` tasks**. Models are ranked according to their average performance on perception tasks. Rows corresponding to proprietary models are highlighted in gray for distinction.

| Method | LLM | Perception | | | | Reasoning | | | |
|--------|-----|---------|-------|------|-------|---------|-------|------|-------|
| | | Diagram | Table | Avg | Avg-C | Diagram | Table | Avg | Avg-C |
| Qwen2-VL | Qwen2-7B | 74.70 | 68.59 | 70.18 | 71.65 | 47.13 | 49.39 | 48.60 | 48.37 |
| Claude 3.5 Sonnet | - | 71.31 | 66.08 | 67.44 | 68.70 | 60.92 | 61.35 | 61.20 | 61.14 |
| InternLM-XComposer2.5 | InternLM2-7B | 69.05 | 62.12 | 63.92 | 65.59 | 43.10 | 39.88 | 41.00 | 41.49 |
| InternVL-2 | InternLM2.5-7B-Chat | 68.83 | 60.68 | 62.80 | 64.76 | 42.53 | 37.12 | 39.00 | 39.83 |
| InternVL-Chat-V1-5 | InternLM2-Chat-20B | 61.55 | 53.81 | 55.83 | 57.68 | 37.36 | 34.36 | 35.40 | 35.86 |
| MiniCPM-V 2.5 | LLama3-8B | 57.81 | 51.05 | 52.81 | 54.43 | 26.44 | 34.66 | 31.80 | 30.55 |
| CogVLm2-llama3-Chat | LLama3-8B | 51.52 | 46.09 | 47.51 | 48.81 | 31.61 | 33.44 | 32.80 | 32.53 |
| GPT-4o | - | 47.35 | 46.44 | 46.68 | 46.90 | 44.25 | 45.09 | 44.80 | 44.67 |
| Mini-Gemini-34B-HD | Nous-Hermes-2-Yi-34B | 47.63 | 43.21 | 44.36 | 45.42 | 38.51 | 39.57 | 39.20 | 39.04 |
| GPT-4o-mini | - | 46.22 | 43.54 | 44.23 | 44.88 | 38.51 | 40.49 | 39.80 | 39.50 |
| Cambrian-1-34B | Nous-Hermes-2-Yi-34B | 43.67 | 39.30 | 40.44 | 41.49 | 37.93 | 34.97 | 36.00 | 36.45 |
| Gemini-1.5-pro | - | 41.41 | 39.37 | 39.90 | 40.39 | 35.63 | 31.90 | 33.20 | 33.77 |
| Cambrian-1-8B | LLama3-8B-Instruct | 36.25 | 31.48 | 32.73 | 33.87 | 27.59 | 27.30 | 27.40 | 27.45 |
| Monkey | Qwen-7B | 34.98 | 31.63 | 32.51 | 33.31 | 18.39 | 22.09 | 20.80 | 20.24 |
| mPLUG-DocOwl 1.5 | LLama-7B | 30.74 | 28.85 | 29.34 | 29.80 | 18.39 | 20.55 | 19.80 | 19.47 |
| SliME-8B | LLama3-8B | 29.75 | 29.19 | 29.34 | 29.47 | 29.89 | 29.14 | 29.40 | 29.52 |
| LLaVA-Next | Qwen-72B | 27.77 | 27.65 | 27.68 | 27.71 | 36.78 | 32.82 | 34.20 | 34.80 |
| LLaVA-Next | LLama3-8B | 26.64 | 26.63 | 26.63 | 26.64 | 22.99 | 23.62 | 23.40 | 23.31 |
| DeepSeek-VL | DeepSeek-LLM-7b-base | 23.67 | 23.27 | 23.38 | 23.47 | 22.99 | 24.23 | 23.80 | 23.61 |
| Mini-Gemini-7B-HD | Vicuna-7B-v1.5 | 21.20 | 22.70 | 22.31 | 21.95 | 27.01 | 23.31 | 24.60 | 25.16 |
| SliME-13B | Vicuna-13B | 19.28 | 21.40 | 20.93 | 20.34 | 38.51 | 39.26 | 39.00 | 38.89 |
| MiniGPT-v2 | Llama 2-7B-Chat | 18.59 | 21.06 | 20.41 | 19.83 | 20.69 | 20.25 | 20.40 | 20.47 |
| LLaVA1.5-13B | Vicuna-13B | 18.30 | 20.83 | 20.17 | 19.57 | 22.41 | 19.94 | 20.80 | 21.18 |
| ShareGPT4V-13B | Vicuna-13B | 18.37 | 20.81 | 20.17 | 19.59 | 21.84 | 20.25 | 20.80 | 21.05 |
| LLaVA1.5-7B | Vicuna-7B | 18.30 | 20.71 | 20.08 | 19.51 | 21.84 | 19.94 | 20.60 | 20.89 |
| ShareGPT4V-7B | Vicuna-7B | 18.30 | 20.71 | 20.08 | 19.51 | 21.84 | 19.94 | 20.60 | 20.89 |
| YI-VL-34B | Yi-34B-Chat | 15.90 | 16.03 | 15.99 | 15.97 | 23.56 | 27.30 | 26.00 | 25.43 |
| Qwen-VL-Chat | Qwen-7B | 18.42 | 14.56 | 15.59 | 16.49 | 14.94 | 12.88 | 13.60 | 13.91 |
| TextMonkey | Qwen-7B | 6.71 | 5.65 | 5.93 | 6.18 | 3.45 | 1.53 | 2.20 | 2.49 |

60%. Reasoning tasks, however, have proved to be more challenging. Only Claude 3.5 Sonnet manage to exceed 60% accuracy, standing out significantly, with the second-ranked Qwen2-VL trailing by 10%. Additionally, models like LLaVA-Next, which have performed well on existing benchmarks like chartQA, show noticeably weaker performance on our dataset, underscoring the higher difficulty of the our benchmark.

**Remote Sensing.** Tab. 8 presents the performance of various models on remote sensing tasks. It is evident that models performing well on remote sensing data typically either employ special handling for high-resolution images (e.g., Mini-Gemini-HD, SliME, Cambrian) or have vision encoders designed to support high-resolution inputs (e.g., InternVL). Among these, SliME achieves the highest performance due to its support for the largest resolution. However, even the top-performing model, SliME-8B, shows poor performance on counting tasks with extremely large images, with only 30% accuracy. Some closed-source models perform even worse, with GPT-4o-mini achieving only 2% accuracy. This highlights the high demands of remote sensing data on resolution and detail perception.

**Autonomous Driving.** Tab. 9 and Tab. 10 show the perception and reasoning performance of various models in autonomous driving scenarios. As a critical application area, autonomous driving remains a challenge for MLLMs, with no model currently capable of reliably addressing tasks such as intent prediction, traffic light recognition, and object counting solely through text. Only Claude 3.5 Sonnet achieve an average perception accuracy exceeding 40%. Reasoning tasks are even more difficult, with even the most advanced models achieving only around 30% accuracy. Autonomous driving is inherently a high-risk domain that demands very high accuracy for practical deployment. This indicates that more powerful multimodal models with 3D spatial prediction and understanding ability, or specialized fine-tuning on domain-specific datasets for driving expertise, are necessary before MLLMs can be effectively applied in this field.

**Monitoring Performance**. Tab. 11 presents the performance of various models under monitoring scenarios. As can be observed, the monitoring task poses a high degree of difficulty. Traditional models like Qwen-VL and LLaVA have an accuracy rate of around 20%, which is nearly equivalent to random guessing. Open-source models significantly outperform closed-source models. For

Table 8: **Experimental results on the `Remote Sensing` tasks.** Models are ranked according to their average performance. Rows corresponding to proprietary models are highlighted in gray for distinction.

| Method | LLM | Color | Count | Position | Avg | Avg-C |
|---|---|---|---|---|---|---|
| Qwen2-VL | Qwen2-7B | 50.92 | 21.21 | 61.73 | 44.81 | 44.62 |
| SliME-8B | LLama3-8B | 45.66 | 28.63 | 52.19 | 42.27 | 42.16 |
| Mini-Gemini-34B-HD | Nous-Hermes-2-Yi-34B | 41.12 | 21.29 | 58.31 | 40.40 | 40.24 |
| Cambrian-1-8B | LLama3-8B-Instruct | 38.01 | 20.55 | 61.10 | 40.05 | 39.89 |
| InternVL-2 | InternLM2.5-7B-Chat | 47.41 | 25.69 | 44.63 | 39.35 | 39.24 |
| Cambrian-1-34B | Nous-Hermes-2-Yi-34B | 37.05 | 22.76 | 55.69 | 38.63 | 38.50 |
| InternLM-XComposer2.5 | InternLM2-7B | 45.34 | 17.62 | 44.95 | 36.12 | 35.97 |
| InternVL-Chat-V1-5 | InternLM2-Chat-20B | 34.10 | 17.86 | 48.29 | 33.55 | 33.42 |
| YI-VL-34B | Yi-34B-Chat | 34.02 | 19.00 | 41.53 | 31.62 | 31.52 |
| Mini-Gemini-7B-HD | Vicuna-7B-v1.5 | 37.29 | 22.43 | 33.97 | 31.30 | 31.23 |
| LLaVA-Next | Qwen-72B | 33.86 | 23.08 | 30.31 | 29.13 | 29.08 |
| GPT-4o | - | 34.18 | 15.17 | 37.07 | 28.92 | 28.81 |
| CogVLm2-llama3-Chat | LLama3-8B | 37.69 | 18.35 | 29.99 | 28.76 | 28.68 |
| MiniCPM-V 2.5 | LLama3-8B | 37.69 | 11.50 | 33.49 | 27.69 | 27.56 |
| SliME-13B | Vicuna-13B | 30.04 | 18.43 | 28.80 | 25.82 | 25.76 |
| Claude 3.5 Sonnet | - | 31.87 | 18.11 | 27.95 | 25.74 | 25.98 |
| DeepSeek-VL | DeepSeek-LLM-7b-base | 29.40 | 7.34 | 39.30 | 25.49 | 25.35 |
| LLaVA-Next | LLama3-8B | 30.04 | 20.55 | 25.74 | 25.42 | 25.44 |
| Monkey | Qwen-7B | 22.07 | 16.97 | 35.72 | 24.99 | 24.92 |
| mPLUG-DocOwl 1.5 | LLaMA-7B | 27.81 | 16.39 | 26.81 | 23.71 | 23.67 |
| MiniGPT-v2 | Llama 2-7B-Chat | 23.35 | 20.15 | 26.41 | 23.33 | 23.30 |
| LLaVA1.5-13B | Vicuna-13B | 26.22 | 16.88 | 26.57 | 23.27 | 23.22 |
| ShareGPT4V-13B | Vicuna-13B | 25.58 | 16.97 | 26.49 | 23.06 | 23.01 |
| LLaVA1.5-7B | Vicuna-7B | 23.11 | 16.88 | 26.25 | 22.12 | 22.08 |
| ShareGPT4V-7B | Vicuna-7B | 23.03 | 16.88 | 26.25 | 22.10 | 22.05 |
| Qwen-VL-Chat | Qwen-7B | 16.97 | 11.50 | 16.87 | 15.14 | 15.11 |
| Gemini-1.5-pro | - | 13.39 | 8.32 | 20.13 | 13.99 | 13.95 |
| TextMonkey | Qwen-7B | 6.93 | 2.04 | 25.86 | 11.69 | 11.61 |
| GPT-4o-mini | - | 5.82 | 2.61 | 11.54 | 6.69 | 6.66 |

Table 9: **Experimental results on the `Autonomous Driving` perception tasks.** Models are ranked according to their average performance. Rows corresponding to proprietary models are highlighted in gray for distinction.

| Method | LLM | Identity | Motion | | | | Traffic Signal | Object Count | Avg | Avg-C |
|---|---|---|---|---|---|---|---|---|---|---|
| | | | Vehicle | Multi-vehicle | Pedestrain | Multi-pedestrain | | | | |
| Claude 3.5 Sonnet | - | 58.66 | 18.45 | 35.48 | 32.32 | 31.64 | 37.31 | 33.19 | 40.77 | 49.72 |
| Cambrian-1-8B | LLama3-8B-Instruct | 56.77 | 50.00 | 33.29 | 32.32 | 14.00 | 35.82 | 33.06 | 38.52 | 47.65 |
| InternVL-2 | InternLM2.5-7B-Chat | 46.68 | 39.24 | 30.98 | 34.76 | 17.24 | 37.81 | 34.58 | 35.46 | 41.07 |
| Qwen2-VL | Qwen2-7B | 43.69 | 39.87 | 31.96 | 28.05 | 14.40 | 53.73 | 32.64 | 34.62 | 39.16 |
| MiniCPM-V 2.5 | LLama3-8B | 44.96 | 41.77 | 30.86 | 31.71 | 19.88 | 37.31 | 29.17 | 34.15 | 39.56 |
| SliME-8B | LLama3-8B | 44.50 | 51.90 | 28.68 | 29.27 | 15.21 | 29.35 | 33.61 | 33.66 | 39.08 |
| InternLM-XComposer2.5 | InternLM2-7B | 46.23 | 48.10 | 26.61 | 32.93 | 11.76 | 40.30 | 32.50 | 33.63 | 39.93 |
| Cambrian-1-34B | Nous-Hermes-2-Yi-34B | 43.96 | 38.61 | 31.96 | 32.93 | 12.37 | 27.36 | 33.89 | 33.39 | 38.68 |
| DeepSeek-VL | DeepSeek-LLM-7b-base | 44.05 | 63.29 | 25.88 | 36.59 | 18.05 | 39.30 | 27.22 | 33.39 | 38.72 |
| Mini-Gemini-34B-HD | Nous-Hermes-2-Yi-34B | 39.78 | 42.41 | 36.09 | 31.10 | 16.63 | 17.41 | 31.53 | 32.70 | 36.24 |
| InternVL-Chat-V1-5 | InternLM2-Chat-20B | 40.42 | 39.87 | 26.97 | 27.44 | 18.66 | 32.34 | 30.14 | 31.42 | 35.92 |
| CogVLm2-llama3-Chat | LLama3-8B | 33.15 | 36.71 | 29.77 | 31.71 | 19.27 | 43.78 | 28.19 | 30.22 | 31.69 |
| Monkey | Qwen-7B | 35.97 | 60.76 | 24.30 | 37.20 | 18.26 | 32.34 | 24.72 | 29.67 | 32.82 |
| YI-VL-34B | Yi-34B-Chat | 36.24 | 41.77 | 29.60 | 31.71 | 20.89 | 16.92 | 19.17 | 28.31 | 32.28 |
| mPLUG-DocOwl 1.5 | LLama-7B | 26.74 | 60.76 | 24.79 | 31.10 | 22.72 | 43.28 | 26.53 | 28.28 | 27.51 |
| SliME-13B | Vicuna-13B | 26.61 | 46.84 | 24.54 | 32.32 | 17.65 | 43.28 | 27.50 | 27.16 | 26.89 |
| Gemini-1.5-pro | - | 32.61 | 10.13 | 30.23 | 8.54 | 16.02 | 10.45 | 31.31 | 26.64 | 29.63 |
| LLaVA1.5-13B | Vicuna-13B | 23.25 | 31.65 | 24.91 | 31.10 | 25.96 | 36.32 | 26.80 | 26.12 | 24.69 |
| ShareGPT4V-13B | Vicuna-13B | 23.25 | 31.01 | 24.91 | 31.10 | 25.96 | 36.82 | 26.81 | 26.12 | 24.69 |
| LLaVA1.5-7B | Vicuna-7B | 23.25 | 31.01 | 24.91 | 31.10 | 25.96 | 35.32 | 26.81 | 26.04 | 24.65 |
| ShareGPT4V-7B | Vicuna-7B | 23.25 | 31.01 | 24.91 | 31.10 | 25.96 | 35.32 | 26.81 | 26.04 | 24.65 |
| MiniGPT-v2 | Llama 2-7B-Chat | 23.71 | 53.16 | 23.16 | 28.66 | 20.49 | 35.32 | 28.06 | 25.96 | 24.84 |
| Mini-Gemini-7B-HD | Vicuna-7B-v1.5 | 27.25 | 60.76 | 23.57 | 26.83 | 14.81 | 36.32 | 17.78 | 24.81 | 26.03 |
| GPT-4o-mini | - | 19.07 | 45.57 | 24.67 | 23.78 | 11.36 | 40.30 | 31.11 | 24.18 | 21.63 |
| GPT-4o | - | 15.26 | 23.42 | 25.39 | 26.22 | 9.94 | 41.29 | 32.22 | 22.43 | 18.85 |
| LLaVA-Next | LLama3-8B | 21.44 | 41.77 | 22.36 | 29.88 | 9.23 | 22.39 | 8.06 | 18.66 | 20.05 |
| LLaVA-Next | Qwen-72B | 19.26 | 26.58 | 24.79 | 29.88 | 12.58 | 16.42 | 5.97 | 17.98 | 18.62 |
| Qwen-VL-Chat | Qwen | 9.26 | 35.44 | 15.43 | 23.17 | 8.32 | 34.83 | 16.39 | 15.08 | 12.17 |
| TextMonkey | Qwen-7B | 8.54 | 37.34 | 22.72 | 15.85 | 16.23 | 14.93 | 6.39 | 14.26 | 11.40 |

Table 10: **Experimental results on the `Autonomous Driving` reasoning tasks.** Models are ranked according to their average performance, from highest to lowest. Rows corresponding to proprietary models are highlighted in gray for distinction.

| Method | LLM | Intention | | | Relation | | | | Attention | Avg | Avg-C |
|---|---|---|---|---|---|---|---|---|---|---|---|
| | | Ego | Pedestrian | Verhicle | Ego2P | Ego2T | Ego2V | O2O | Signal | | |
| Monkey | Qwen-7B | 28.62 | 56.31 | 30.43 | 27.36 | 22.86 | 32.67 | 11.94 | 58.06 | 33.04 | 33.48 |
| Claude 3.5 Sonnet | - | 26.32 | 32.04 | 24.64 | 23.58 | 25.71 | 20.79 | 24.38 | 65.90 | 31.92 | 30.59 |
| YI-VL-34B | Yi-34B-Chat | 28.26 | 46.60 | 33.33 | 21.70 | 24.76 | 31.68 | 15.42 | 49.77 | 31.55 | 31.45 |
| SliME-8B | LLama3-8B | 28.29 | 39.81 | 33.33 | 24.53 | 19.05 | 22.77 | 10.45 | 63.59 | 31.55 | 30.37 |
| Qwen2-VL | Qwen2-7B | 19.08 | 43.69 | 35.75 | 13.21 | 29.52 | 22.77 | 18.41 | 64.98 | 31.47 | 30.99 |
| CogVLm2-llama3-Chat | LLama3-8B | 30.26 | 25.24 | 25.60 | 35.85 | 20.95 | 28.71 | 18.41 | 56.22 | 31.18 | 30.27 |
| MiniCPM-V 2.5 | LLama3-8B | 24.01 | 37.86 | 31.88 | 20.75 | 30.48 | 15.84 | 26.87 | 53.00 | 31.03 | 30.19 |
| SliME-13B | Vicuna-13B | 25.00 | 41.75 | 28.99 | 28.30 | 21.90 | 24.75 | 25.87 | 48.39 | 30.80 | 30.64 |
| LLaVA-Next | LLama3-8B | 32.89 | 49.51 | 33.82 | 28.30 | 25.71 | 24.75 | 7.96 | 43.32 | 30.73 | 30.78 |
| Cambrian-1-8B | LLama3-8B-Instruct | 25.00 | 41.75 | 35.27 | 23.58 | 23.81 | 16.83 | 11.44 | 60.37 | 30.73 | 29.86 |
| InternVL-2 | InternLM2.5-7B-Chat | 24.01 | 43.69 | 32.85 | 22.64 | 28.57 | 21.78 | 21.89 | 43.78 | 29.84 | 29.89 |
| LLaVA-Next | Qwen-72B | 30.59 | 52.43 | 35.27 | 23.58 | 27.62 | 29.70 | 8.96 | 35.48 | 29.69 | 30.37 |
| InternVL-Chat-V1-5 | InternLM2-Chat-20B | 25.99 | 32.04 | 31.88 | 16.98 | 22.86 | 25.74 | 9.45 | 57.14 | 28.94 | 27.89 |
| DeepSeek-VL | DeepSeek-LLM-7b-base | 30.26 | 17.48 | 27.05 | 22.64 | 25.71 | 24.75 | 6.97 | 51.15 | 27.31 | 25.92 |
| GPT-4o-mini | - | 11.51 | 19.42 | 24.64 | 22.64 | 28.57 | 17.82 | 31.34 | 54.84 | 26.79 | 26.40 |
| GPT-4o | - | 17.11 | 19.42 | 27.54 | 15.09 | 20.00 | 22.77 | 16.92 | 60.83 | 26.41 | 25.12 |
| mPLUG-DocOwl 1.5 | LLama-7B | 20.72 | 26.21 | 30.43 | 19.81 | 31.43 | 25.74 | 12.94 | 41.94 | 26.04 | 26.14 |
| LLaVA1.5-13B | Vicuna-13B | 23.36 | 18.45 | 24.15 | 26.42 | 23.81 | 22.77 | 25.37 | 30.41 | 24.78 | 24.39 |
| Qwen-VL-Chat | Qwen | 20.39 | 21.36 | 20.77 | 16.04 | 23.81 | 17.82 | 16.92 | 50.69 | 24.63 | 23.60 |
| ShareGPT4V-13B | Vicuna-13B | 23.36 | 17.48 | 26.09 | 25.47 | 27.63 | 22.77 | 25.37 | 26.27 | 24.55 | 24.33 |
| Cambrian-1-34B | Nous-Hermes-2-Yi-34B | 14.14 | 24.27 | 22.71 | 24.53 | 30.48 | 31.68 | 10.95 | 46.54 | 24.40 | 25.52 |
| LLaVA1.5-7B | Vicuna-7B | 23.36 | 17.48 | 26.09 | 25.47 | 27.62 | 22.77 | 25.37 | 23.96 | 24.18 | 24.03 |
| ShareGPT4V-7B | Vicuna-7B | 23.36 | 17.48 | 26.09 | 25.47 | 27.62 | 22.77 | 25.37 | 23.96 | 24.18 | 24.03 |
| InternLM-XComposer2.5 | InternLM2-7B | 25.33 | 36.89 | 34.30 | 17.92 | 26.67 | 25.74 | 23.88 | 44.24 | 24.03 | 28.78 |
| MiniGPT-v2 | Llama 2-7B-Chat | 23.68 | 25.24 | 28.02 | 28.30 | 22.86 | 21.78 | 2.49 | 37.33 | 23.66 | 23.71 |
| Mini-Gemini-7B-HD | Vicuna-7B-v1.5 | 21.05 | 6.80 | 14.49 | 19.81 | 15.24 | 25.74 | 15.42 | 54.38 | 23.29 | 21.80 |
| Mini-Gemini-34B-HD | Nous-Hermes-2-Yi-34B | 14.14 | 22.33 | 21.74 | 23.58 | 31.43 | 30.69 | 10.45 | 47.00 | 22.84 | 24.91 |
| TextMonkey | Qwen-7B | 9.54 | 24.27 | 23.67 | 24.53 | 17.14 | 20.79 | 11.44 | 35.94 | 20.01 | 20.81 |
| Gemini-1.5-pro | - | 13.49 | 18.45 | 28.02 | 6.60 | 6.67 | 6.93 | 23.88 | 32.72 | 19.20 | 17.33 |

instance, InternVL-2 has an average accuracy rate of 53.19 on perception tasks, greatly surpassing GPT-4o's 33.93. We notice that closed-source models such as GPT-4o have a high frequency of answering "E", with over 35% of responses choosing "E". This suggests that closed-source models may be more inclined to refrain from responding when the answer is uncertain. Furthermore, we find that while most models perform reasonably well on counting tasks, they struggle with tasks related to spatial relationship judgment and attribute recognition. Moreover, related reasoning tasks also pose a high level of difficulty, with no model achieving an accuracy rate over 40% to date. In combination with the results from autonomous driving tasks, we observe that MLLMs exhibit significant deficiencies in understanding, predicting, and reasoning about the dynamic information of objects in 2D or 3D space. Although the input to these models is a single frame image rather than a temporal sequence of video frames, there remains a considerable gap between their performance and that of humans. For humans, who possess rich experiential knowledge of dynamics, it is not difficult to infer the future states of objects from a single image in unambiguous situations. Therefore, modern MLLMs are still far from having the capability to function as world models.

# E EXPERIMENTAL RESULTS ON ALL TASK SPLITS OF MME-REALWORLD-CN

The detailed results on MME-RealWorld-CN can be found in Tab. 13, Tab. 15, Tab. 14, Tab. 16, Tab. 17, and Tab. 18.

# F MME-REALWORLD-LITE

To facilitate the evaluation of high-resolution perception abilities in multimodal large language models, we plan to sample 50 examples from each task within each domain, resulting in a mini-test set.

Tab. 19 summarizes our initial tests with several models. While it is challenging to fully preserve model performance on the original dataset due to the sampling process, the overall task difficulty has increased. We reduced the number of samples for simpler tasks, such as OCR and basic reasoning, making the test set more demanding. For instance, Qwen2-VL's performance decreased from 56.4 on the full dataset to 46.7 on the reduced test set, underscoring the increased difficulty.

Table 11: **Experimental results on the `Monitoring` tasks**. Models are ranked according to their average performance on perception tasks. Rows corresponding to proprietary models are highlighted in gray for distinction.

| Method | LLM | Perception | | | | | | | Reasoning | | | | |
|---|---|---|---|---|---|---|---|---|---|---|---|---|---|
| | | Vehicle | | | Pedestrain | | | | | | | | |
| | | Counting | Location | Attribute | Counting | Attribute | Avg | Avg-C | Calculate | Intention | Property | Avg | Avg-C |
| InternVL-2 | InternLM2.5-7B-Chat | 70.07 | 25.74 | 28.98 | 59.68 | 12.04 | 53.19 | 41.62 | 51.67 | 21.43 | 41.00 | 43.57 | 38.03 |
| InternVL-Chat-V1-5 | InternLM2-Chat-20B | 72.53 | 23.53 | 27.27 | 55.24 | 7.41 | 51.16 | 39.52 | 39.33 | 26.53 | 42.00 | 37.35 | 35.95 |
| Cambrian-1-8B | LLama3-8B-Instruct | 62.01 | 29.41 | 20.45 | 55.44 | 7.41 | 47.68 | 37.07 | 46.00 | 29.59 | 44.00 | 42.37 | 39.86 |
| Cambrian-1-34B | Nous-Hermes-2-Yi-34B | 51.32 | 33.09 | 26.14 | 55.14 | 12.96 | 45.98 | 37.44 | 11.33 | 18.37 | 45.00 | 19.48 | 24.90 |
| SliME-8B | LLama3-8B | 60.53 | 33.82 | 28.98 | 34.48 | 31.48 | 40.62 | 38.32 | 32.33 | 40.82 | 43.00 | 36.14 | 38.72 |
| Mini-Gemini-34B-HD | Nous-Hermes-2-Yi-34B | 53.95 | 17.65 | 22.73 | 43.45 | 7.41 | 39.61 | 30.80 | 11.67 | 17.35 | 50.00 | 20.48 | 26.34 |
| InternLM-XComposer2.5 | InternLM2-7B | 52.63 | 13.24 | 17.61 | 46.98 | 0.93 | 39.48 | 28.48 | 13.67 | 13.27 | 34.00 | 17.67 | 20.31 |
| MiniCPM-V 2.5 | LLama3-8B | 62.66 | 16.91 | 22.73 | 36.49 | 4.63 | 38.70 | 30.35 | 36.00 | 35.71 | 41.00 | 36.95 | 37.57 |
| Qwen2-VL | Qwen2-7B | 55.59 | 19.85 | 25.00 | 35.69 | 11.11 | 37.30 | 29.45 | 30.33 | 19.39 | 55.00 | 33.13 | 34.91 |
| YI-VL-34B | Yi-34B-Chat | 56.25 | 8.09 | 23.86 | 32.47 | 5.56 | 34.85 | 26.85 | 28.00 | 28.57 | 44.00 | 31.33 | 33.52 |
| Mini-Gemini-7B-HD | Vicuna-7B-v1.5 | 47.86 | 13.97 | 25.00 | 34.07 | 13.89 | 34.15 | 28.16 | 21.00 | 24.49 | 42.00 | 25.90 | 29.16 |
| GPT-4o | - | 50.66 | 15.44 | 19.89 | 34.17 | 6.48 | 33.93 | 26.76 | 4.00 | 13.27 | 41.00 | 19.42 | 19.42 |
| CogVLm2-llama3-Chat | LLama3-8B | 48.19 | 26.47 | 22.16 | 32.36 | 12.04 | 33.74 | 29.16 | 40.00 | 40.82 | 45.00 | 41.16 | 41.94 |
| Claude 3.5 Sonnet | - | 50.99 | 33.77 | 11.93 | 33.37 | 8.33 | 32.19 | 28.43 | 34.37 | 18.37 | 44.00 | 32.25 | 32.25 |
| Gemini-1.5-pro | - | 52.63 | 9.56 | 10.80 | 31.05 | 10.08 | 31.11 | 24.21 | 11.67 | 13.27 | 27.00 | 17.31 | 17.31 |
| LLaVA-Next | Qwen-72B | 57.89 | 10.39 | 27.27 | 15.32 | 28.70 | 29.37 | 28.16 | 23.33 | 38.78 | 28.00 | 27.31 | 30.04 |
| Monkey | Qwen-7B | 42.76 | 40.44 | 21.02 | 21.77 | 9.26 | 28.01 | 27.21 | 25.33 | 21.43 | 39.00 | 27.31 | 28.59 |
| DeepSeek-VL | DeepSeek-LLM-7b-base | 43.91 | 7.35 | 17.33 | 24.70 | 9.26 | 26.97 | 21.59 | 5.33 | 19.39 | 48.00 | 16.67 | 24.24 |
| GPT-4o-mini | - | 44.57 | 8.82 | 8.52 | 26.71 | 3.70 | 26.50 | 19.80 | 7.33 | 8.16 | 19.00 | 11.50 | 11.50 |
| mPLUG-DocOwl 1.5 | LLaMA-7B | 34.87 | 19.12 | 26.42 | 21.27 | 6.48 | 24.97 | 22.19 | 10.00 | 33.67 | 39.00 | 20.48 | 27.56 |
| SliME-13B | Vicuna-13B | 20.56 | 22.79 | 23.30 | 29.33 | 12.96 | 24.73 | 22.28 | 33.00 | 26.53 | 40.00 | 33.13 | 33.18 |
| Qwen-VL-Chat | Qwen-7B | 37.66 | 16.18 | 21.88 | 14.62 | 12.04 | 22.13 | 20.75 | 14.67 | 17.35 | 21.00 | 16.47 | 17.67 |
| LLaVA1.5-13B | Vicuna-13B | 14.47 | 21.59 | 20.45 | 24.29 | 12.96 | 20.45 | 19.30 | 27.67 | 23.47 | 31.00 | 27.51 | 27.38 |
| LLaVA-Next | LLama3-8B | 46.71 | 0.00 | 22.16 | 4.13 | 23.15 | 19.46 | 19.27 | 13.67 | 46.94 | 18.00 | 21.08 | 26.20 |
| ShareGPT4V-13B | Vicuna-13B | 14.31 | 22.06 | 15.62 | 23.99 | 12.04 | 19.26 | 17.88 | 27.33 | 24.49 | 30.00 | 27.31 | 27.27 |
| MiniGPT-v2 | Llama 2-7B-Chat | 13.98 | 22.06 | 15.34 | 24.40 | 11.11 | 19.26 | 17.69 | 13.67 | 19.39 | 24.00 | 16.87 | 19.02 |
| LLaVA1.5-7B | Vicuna-7B | 14.31 | 22.06 | 15.62 | 23.79 | 11.11 | 19.13 | 17.67 | 27.33 | 23.47 | 24.00 | 25.90 | 24.93 |
| ShareGPT4V-7B | Vicuna-7B | 14.31 | 22.06 | 15.62 | 23.79 | 11.11 | 19.13 | 17.67 | 27.33 | 23.47 | 25.00 | 26.10 | 25.27 |
| TextMonkey | Qwen-7B | 39.47 | 6.62 | 7.10 | 8.17 | 0.00 | 16.14 | 12.92 | 0.67 | 4.08 | 16.00 | 4.42 | 6.92 |

This compact test set offers researchers a practical yet robust tool for small-scale, focused evaluations and model comparisons. It maintains a balance across diverse tasks, ensuring that evaluations remain insightful while being accessible to the broader research community.

# G  OTHER ANALYSIS

**Extended Metrics** In addition to the multiple-choice option, we also plan to release a performance evaluation version of the dataset (EM), which retains only the question without providing choices. The evaluation will be conducted using exact match or GPT-match[8] to assess the final result. This approach prevents models from relying on information from the choices. We selected 50 samples from each task for testing, using the prompt "Please respond to the question with a single word or phrase," prompting the model to generate a direct response. The generated response is then compared to the correct answer to check for consistency. The input-output format of EM compared to naive MCQ is shown in Fig. 16.

In our experiments (Tab. 23), all models showed a noticeable performance drop when choices were removed (EM vs. MCQ). Additionally, using GPT-4 to align the model's response with the intended meaning of both the response and the correct option (line 10 vs. line 9) showed some performance improvement; however, the results still lagged behind those with choices provided. Consequently, Machine Match is a key evaluation strategy we plan to prioritize in future assessments. Under this evaluation strategy, GPT-4 achieves around 30% accuracy, highlighting the increased difficulty level of our tasks. **This suggests that the challenge level of tasks based on our benchmark data could be further enhanced**.

**Simple Tricks Cannot Solve the Problem of Large Image Perception Effectively.** A straightforward strategy was attempted to alleviate the visual perception challenges posed by large images. Intuitively, when humans observe high-resolution remote sensing images, they often zoom in for a closer look. Additionally, questions and answers often include directional cues, such as "the upper right corner of the image," "above the image," or "below the image." Hence, it would be more reasonable to segment the corresponding image before processing.

To implement this, each image is uniformly divided into 9 sub-images, and questions are posed for each sub-image, obtaining a corresponding answer. Each image has a multiple-choice result; if any

---

[8]We prompt GPT-4O with the following to determine consistency between the model output and the correct answer: "Please determine whether the following two responses are consistent. If they are, output 1; otherwise, output 0. Answer A: {answer1}; Answer B: {answerB}. The output number (0 or 1) is:"

Table 12: **Experimental results on the reasoning tasks of MME-RealWorld-CN.** Models are ranked according to their average performance. Rows corresponding to proprietary models are highlighted in gray for distinction. "OCR", "DT", "MO", and "AD" each indicate a specific task domain: Optical Character Recognition in the Wild, Diagram and Table, Monitoring and Autonomous Driving, respectively. "Avg" and "Avg-C" indicate the weighted average accuracy and the unweighted average accuracy across subtasks in each domain.

| Method | LLM | Reasoning | | | | | |
|---|---|---|---|---|---|---|---|
| **Task Split** | | OCR | DT | MO | AD | Avg | Avg-C |
| # QA pairs | | 207 | 602 | 298 | 800 | 1907 | 1907 |
| InternVL-2.5 | InternLM2.5-7B-Chat | 53.14 | 73.59 | 30.20 | 31.25 | 46.83 | 47.05 |
| InternVL-2 | InternLM2.5-7B-Chat | 44.93 | 74.92 | 38.14 | 29.00 | 46.65 | 46.75 |
| Qwen2-VL | Qwen2-7B | 38.16 | 72.92 | 57.00 | 33.37 | 46.46 | 50.36 |
| Claude 3.5 Sonnet | - | 74.44 | 65.79 | 31.54 | 25.12 | 44.31 | 49.22 |
| SliME-8B | LLama3-8B | 44.44 | 70.93 | 30.54 | 29.13 | 44.21 | 43.76 |
| InternVL-Chat-V1-5 | InternLM2-Chat-20B | 48.79 | 67.11 | 30.20 | 29.88 | 43.74 | 44.00 |
| CogVLM2-llama3-Chat | LLama3-8B | 33.81 | 65.24 | 37.25 | 29.00 | 42.25 | 41.33 |
| YI-VL-34B | Yi-34B-Chat | 37.68 | 61.46 | 33.22 | 29.75 | 41.16 | 40.53 |
| Monkey | Qwen-7B | 43.96 | 56.81 | 32.55 | 28.38 | 39.70 | 40.43 |
| Mini-Gemini-7B-HD | Vicuna-7B-v1.5 | 28.50 | 68.61 | 25.50 | 24.00 | 38.80 | 36.65 |
| Mini-Gemini-34B-HD | Nous-Hermes-2-Yi-34B | 29.95 | 60.47 | 25.50 | 29.63 | 38.75 | 36.39 |
| LLaVA-Next | LLama3-8B | 14.93 | 58.14 | 29.53 | 28.25 | 36.44 | 32.71 |
| Cambrian-1-8B | LLama3-8B-Instruct | 27.54 | 47.51 | 31.21 | 31.25 | 35.97 | 34.38 |
| SliME-13B | Vicuna-13B | 44.93 | 49.17 | 30.54 | 23.87 | 35.18 | 37.13 |
| LLaVA-Next | Qwen-72B | 24.64 | 40.87 | 27.52 | 28.12 | 31.67 | 30.29 |
| InternLM-XComposer2.5 | InternLM2-7B | 18.36 | 40.70 | 16.78 | 30.00 | 30.05 | 26.46 |
| GPT-4o | - | 33.81 | 39.87 | 20.81 | 22.75 | 29.05 | 29.31 |
| DeepSeek-VL | DeepSeek-LLM-7b-base | 36.23 | 23.59 | 25.50 | 29.25 | 27.63 | 28.64 |
| Cambrian-1-34B | Hermes2-Yi-34B | 21.74 | 31.40 | 23.49 | 25.12 | 26.48 | 25.44 |
| LLaVA1.5-13B | Vicuna-13B | 36.23 | 27.08 | 25.84 | 23.25 | 26.27 | 28.10 |
| ShareGPT4V-13B | Vicuna-13B | 35.75 | 27.91 | 24.83 | 22.88 | 26.17 | 27.84 |
| MiniCPM-V 2.5 | LLama3-8B | 36.23 | 29.90 | 16.44 | 23.87 | 25.95 | 26.61 |
| LLaVA1.5-7B | Vicuna-7B | 33.33 | 25.91 | 25.17 | 23.25 | 25.48 | 26.92 |
| ShareGPT4V-7B | Vicuna-7B | 33.33 | 25.91 | 24.83 | 23.25 | 25.43 | 26.83 |
| Qwen-VL-Chat | Qwen-7B | 30.92 | 36.41 | 13.42 | 19.88 | 25.29 | 25.16 |
| GPT-4o-mini | - | 27.88 | 27.08 | 14.77 | 26.87 | 25.16 | 24.15 |
| MiniGPT-v2 | Llama 2-7B-Chat | 34.30 | 28.57 | 19.80 | 22.13 | 25.12 | 26.20 |
| mPLUG-DocOwl 1.5 | LLama-7B | 37.68 | 24.42 | 19.80 | 22.38 | 24.28 | 26.07 |
| TextMonkey | Qwen-7B | 27.53 | 31.07 | 12.08 | 22.50 | 24.12 | 23.29 |
| Gemini-1.5-pro | - | 5.30 | 5.32 | 14.77 | 15.67 | 11.14 | 10.26 |

choice result is non-E, the image is assigned a result based on the majority vote. Otherwise, the result is E. The results are shown in Tab. 20, indicating a noticeable decline in perception performance across various datasets. There could be several reasons for this decline:

1. The images are very large, and the queried objects are very fine-grained. After segmenting these images, the task remains highly challenging.

2. Positional words in the questions lose their original meaning when the image is segmented without further processing, leading to misleading interpretations.

3. Image segmentation cannot guarantee that the queried objects are not split into different sub-images, resulting in the loss of correct answers.

Thus, simple image segmentation tricks are unlikely to significantly improve the perception of complex images. Instead, the inherent capabilities of the models themselves should be relied upon and improved.

**Analyzing Incorrect Choices.** We investigate the distribution of incorrect choices across a range of models, as shown in Fig. 13. We can see that MLLMs show different response strategies when dealing with questions imbued with uncertainty. Larger models generally adopt a more conservative approach, often opting for the safer response "E", as illustrated from Fig. 13(a) to 13(c). In contrast, smaller MLLMs often lean towards the first option—usually option "A"—in similar situations, as shown in Fig. 13(d) and 13(e). Notably, InternVL-2 presents a unique distribution of incorrect choices that is remarkably uniform, which may account for its exceptional performance in our evaluation.

Table 13: **Experimental results on the `OCR in the Wild` tasks of MME-RealWorld-CN** are categorized as follows: "Product" represents products and advertisements; "B & M & P" represents books, maps, and posters; "Contact" denotes contact information and addresses; "Identity" pertains to identity information; and "Signage" refers to signage and other text. Models are ranked according to their average performance on perception tasks, from highest to lowest. Rows corresponding to proprietary models are highlighted in gray for distinction.

| Method | LLM | Perception | | | | | | | Reasoning | | | |
|---|---|---|---|---|---|---|---|---|---|---|---|---|
| | | Product | B & M & P | Contact | Identity | Signage | Avg | Avg-C | Scene | Character | Avg | Avg-C |
| Qwen2-VL | Qwen2-7B | 70.68 | 66.43 | 78.43 | 59.04 | 68.26 | 70.28 | 68.57 | 48.00 | 28.97 | 38.16 | 38.49 |
| InternVL-2 | InternLM2.5-7B-Chat | 69.51 | 69.76 | 74.51 | 69.92 | 50.60 | 69.92 | 66.86 | 43.93 | 46.00 | 44.93 | 44.97 |
| InternVL-Chat-V1-5 | InternLM2-Chat-20B | 62.91 | 63.81 | 59.56 | 57.68 | 51.81 | 60.59 | 59.15 | 47.66 | 50.00 | 48.79 | 48.83 |
| GPT-4o | - | 60.00 | 61.90 | 58.58 | 47.63 | 34.94 | 55.90 | 52.61 | 35.51 | 32.00 | 33.81 | 33.76 |
| Claude 3.5 Sonnet | - | 49.25 | 65.48 | 53.43 | 53.73 | 39.76 | 54.44 | 52.33 | 84.39 | 63.79 | 74.44 | 74.09 |
| SliME-8B | LLama3-8B | 57.09 | 43.57 | 62.25 | 55.19 | 38.55 | 53.93 | 51.33 | 37.38 | 52.00 | 44.44 | 44.69 |
| YI-VL-34B | Yi-34B-Chat | 45.63 | 39.29 | 59.80 | 63.90 | 34.94 | 51.41 | 48.71 | 29.91 | 46.00 | 37.68 | 37.96 |
| SliME-13B | Vicuna-13B | 57.48 | 45.24 | 48.77 | 50.00 | 48.19 | 50.63 | 49.94 | 47.66 | 42.00 | 44.93 | 44.83 |
| Gemini-1.5-pro | - | 47.57 | 30.24 | 65.69 | 53.32 | 30.12 | 48.32 | 45.39 | 4.98 | 5.65 | 5.30 | 5.32 |
| Cambrian-1-34B | Hermes2-Yi-34B | 40.78 | 46.19 | 52.21 | 60.37 | 12.05 | 48.11 | 42.32 | 20.56 | 23.00 | 21.74 | 21.78 |
| CogVLm2-llama3-Chat | LLama3-8B | 44.47 | 41.43 | 53.92 | 48.34 | 28.92 | 46.12 | 43.42 | 35.51 | 32.00 | 33.81 | 33.76 |
| Mini-Gemini-34B-HD | Nous-Hermes-2-Yi-34B | 39.03 | 38.81 | 47.79 | 44.19 | 31.33 | 41.82 | 40.23 | 17.76 | 43.00 | 29.95 | 30.38 |
| LLaVA-Next | LLama3-8B | 36.50 | 32.86 | 52.45 | 45.02 | 21.69 | 40.62 | 37.70 | 17.66 | 12.00 | 14.93 | 14.83 |
| Monkey | Qwen-7B | 44.85 | 32.86 | 43.14 | 40.46 | 38.55 | 40.46 | 39.97 | 43.93 | 44.00 | 43.96 | 43.97 |
| Mini-Gemini-7B-HD | Vicuna-7B-v1.5 | 37.91 | 33.97 | 43.57 | 43.11 | 40.00 | 39.66 | 39.71 | 19.63 | 38.00 | 28.50 | 28.82 |
| InternLM-XComposer2.5 | InternLM2-7B | 33.98 | 54.76 | 33.82 | 38.80 | 22.89 | 39.26 | 36.85 | 9.35 | 28.00 | 18.36 | 18.68 |
| mPLUG-DocOwl 1.5 | LLama-7B | 29.32 | 31.19 | 39.71 | 32.99 | 39.76 | 33.33 | 34.59 | 42.06 | 33.00 | 37.68 | 37.53 |
| MiniCPM-V 2.5 | LLama3-8B | 28.54 | 31.90 | 33.58 | 39.83 | 28.92 | 33.23 | 32.55 | 38.32 | 34.00 | 36.23 | 36.16 |
| LLaVA-Next | Qwen-72B | 25.24 | 27.86 | 33.82 | 49.38 | 2.41 | 32.76 | 27.74 | 29.91 | 19.00 | 24.64 | 24.46 |
| Cambrian-1-8B | LLama3-8B-Instruct | 27.77 | 40.95 | 33.58 | 33.82 | 10.84 | 32.71 | 29.39 | 28.97 | 26.00 | 27.54 | 27.49 |
| TextMonkey | Qwen-7B | 32.82 | 27.86 | 34.31 | 32.37 | 16.87 | 31.24 | 28.85 | 33.64 | 21.00 | 27.53 | 27.32 |
| GPT-4o-mini | - | 30.29 | 41.90 | 22.55 | 26.56 | 14.46 | 29.56 | 27.15 | 22.27 | 33.89 | 27.88 | 28.08 |
| ShareGPT4V-13B | Vicuna-13B | 28.16 | 27.62 | 27.94 | 28.63 | 24.10 | 27.94 | 27.29 | 37.38 | 34.00 | 35.75 | 35.69 |
| LLaVA1.5-13B | Vicuna-13B | 27.77 | 27.62 | 27.94 | 27.39 | 24.10 | 27.36 | 26.96 | 38.32 | 34.00 | 36.23 | 36.16 |
| Qwen-VL-Chat | Qwen | 32.04 | 26.19 | 28.92 | 24.90 | 10.84 | 27.36 | 24.58 | 27.10 | 35.00 | 30.92 | 31.05 |
| DeepSeek-VL | DeepSeek-LLM-7b-base | 24.08 | 28.10 | 28.92 | 28.63 | 22.89 | 27.10 | 26.52 | 37.38 | 35.00 | 36.23 | 36.19 |
| MiniGPT-v2 | Llama 2-7B-Chat | 26.21 | 27.86 | 27.94 | 26.14 | 22.89 | 26.78 | 26.21 | 36.45 | 32.00 | 34.30 | 34.23 |
| ShareGPT4V-7B | Vicuna-7B | 25.44 | 26.90 | 27.94 | 27.39 | 24.10 | 26.73 | 26.35 | 37.38 | 29.00 | 33.33 | 33.19 |
| LLaVA1.5-7B | Vicuna-7B | 25.44 | 26.90 | 27.94 | 26.14 | 22.89 | 26.36 | 25.86 | 37.38 | 29.00 | 33.33 | 33.19 |

Table 14: **Experimental results on the `Diagram and Table` tasks of MME-RealWorld-CN**. Models are ranked according to their average performance on perception tasks. Rows corresponding to proprietary models are highlighted in gray for distinction.

| Method | LLM | Perception | | | | Reasoning | | | |
|---|---|---|---|---|---|---|---|---|---|
| | | Diagram | Table | Avg | Avg-C | Diagram | Table | Avg | Avg-C |
| Qwen2-VL | Qwen2-7B | 86.38 | 92.03 | 89.20 | 89.21 | 74.42 | 71.43 | 72.92 | 72.93 |
| Claude 3.5 Sonnet | - | 84.39 | 63.79 | 74.09 | 74.09 | 68.11 | 63.46 | 65.79 | 65.79 |
| InternVL-2 | InternLM2.5-7B-Chat | 92.36 | 61.79 | 71.63 | 77.08 | 77.08 | 72.76 | 74.92 | 74.92 |
| InternVL-Chat-V1-5 | InternLM2-Chat-20B | 68.11 | 49.17 | 60.12 | 58.64 | 65.45 | 68.77 | 67.11 | 67.11 |
| SliME-8B | LLama3-8B | 88.70 | 55.15 | 58.25 | 71.93 | 64.45 | 77.41 | 70.93 | 70.93 |
| GPT-4o | - | 41.86 | 29.24 | 54.86 | 35.55 | 56.48 | 23.26 | 39.87 | 39.87 |
| YI-VL-34B | Yi-34B-Chat | 47.51 | 39.52 | 49.52 | 43.52 | 59.80 | 63.12 | 61.46 | 61.46 |
| SliME-13B | Vicuna-13B | 47.84 | 35.55 | 48.49 | 41.70 | 54.49 | 49.17 | 49.17 | 49.17 |
| Cambrian-1-34B | Hermes2-Yi-34B | 29.57 | 35.22 | 44.34 | 32.40 | 9.30 | 53.49 | 31.40 | 31.40 |
| Monkey | Qwen-7B | 44.85 | 41.53 | 41.12 | 43.19 | 54.15 | 59.47 | 56.81 | 56.81 |
| Mini-Gemini-34B-HD | Nous-Hermes-2-Yi-34B | 30.90 | 42.52 | 40.60 | 36.71 | 50.50 | 70.43 | 60.47 | 60.47 |
| Gemini-1.5-pro | - | 13.29 | 7.97 | 39.78 | 10.63 | 6.98 | 5.65 | 6.32 | 6.32 |
| CogVLm2-llama3-Chat | LLama3-8B | 8.64 | 28.24 | 39.48 | 18.44 | 67.11 | 63.37 | 65.24 | 65.24 |
| Mini-Gemini-7B-HD | Vicuna-7B-v1.5 | 37.54 | 30.56 | 39.29 | 34.05 | 60.13 | 77.08 | 68.61 | 68.61 |
| InternLM-XComposer2.5 | InternLM2-7B | 45.18 | 30.23 | 38.88 | 37.71 | 33.89 | 47.51 | 40.70 | 40.70 |
| LLaVA-Next | LLama3-8B | 12.61 | 39.53 | 37.49 | 26.07 | 51.50 | 64.78 | 58.14 | 58.14 |
| mPLUG-DocOwl 1.5 | LLama-7B | 22.26 | 31.89 | 31.83 | 27.08 | 14.62 | 34.22 | 24.42 | 24.42 |
| GPT-4o-mini | - | 45.41 | 32.23 | 31.79 | 38.82 | 20.27 | 33.89 | 27.08 | 27.08 |
| MiniCPM-V 2.5 | LLama3-8B | 25.25 | 28.24 | 31.67 | 26.75 | 29.57 | 30.23 | 29.90 | 29.90 |
| TextMonkey | Qwen-7B | 41.86 | 31.61 | 30.76 | 36.74 | 36.88 | 25.25 | 31.07 | 31.07 |
| Cambrian-1-8B | LLama3-8B-Instruct | 24.58 | 20.60 | 30.28 | 22.59 | 40.53 | 54.49 | 47.51 | 47.51 |
| LLaVA-Next | Qwen-72B | 2.66 | 28.90 | 28.69 | 15.78 | 19.27 | 62.46 | 40.87 | 40.87 |
| Qwen-VL-Chat | Qwen-7B | 21.26 | 37.87 | 27.89 | 29.57 | 30.90 | 41.91 | 36.41 | 36.41 |
| ShareGPT4V-13B | Vicuna-13B | 25.58 | 27.24 | 27.57 | 26.41 | 24.92 | 30.90 | 27.91 | 27.91 |
| MiniGPT-v2 | Llama 2-7B-Chat | 29.24 | 26.58 | 27.05 | 27.91 | 26.91 | 30.23 | 28.57 | 28.57 |
| LLaVA1.5-13B | Vicuna-13B | 25.25 | 27.24 | 26.25 | 26.25 | 25.25 | 28.90 | 27.08 | 27.08 |
| DeepSeek-VL | DeepSeek-LLM-7b-base | 20.93 | 24.25 | 26.02 | 22.59 | 28.24 | 18.94 | 23.59 | 23.59 |
| LLaVA1.5-7B | Vicuna-7B | 24.25 | 27.24 | 25.75 | 25.75 | 24.58 | 27.24 | 25.91 | 25.91 |
| ShareGPT4V-7B | Vicuna-7B | 24.25 | 27.24 | 25.75 | 25.75 | 24.58 | 27.24 | 25.91 | 25.91 |

Table 15: **Experimental results on the `Remote Sensing` tasks of MME-RealWorld-CN.** Models are ranked according to their average performance. Rows corresponding to proprietary models are highlighted in gray for distinction.

| Method | LLM | Color | Count | Position | Avg | Avg-C |
|---|---|---|---|---|---|---|
| SliME-8B | LLama3-8B | 45.00 | 21.00 | 58.00 | 41.33 | 41.33 |
| InternVL-2 | InternLM2.5-7B-Chat | 52.00 | 23.00 | 49.00 | 41.33 | 41.33 |
| Qwen2-VL | Qwen2-7B | 44.00 | 18.00 | 53.00 | 38.33 | 41.33 |
| InternLM-XComposer2.5 | InternLM2-7B | 48.00 | 17.00 | 50.00 | 38.33 | 38.33 |
| Mini-Gemini-34B-HD | Nous-Hermes-2-Yi-34B | 50.00 | 7.00 | 60.00 | 38.28 | 38.82 |
| Cambrian-1-8B | LLama-3-8B-Instruct | 40.00 | 7.00 | 63.33 | 35.86 | 36.55 |
| YI-VL-34B | Yi-34B-Chat | 35.00 | 17.00 | 51.00 | 34.33 | 34.33 |
| Cambrian-1-34B | Hermes2-Yi-34B | 31.00 | 11.00 | 62.22 | 33.79 | 34.50 |
| Claude 3.5 Sonnet | - | 38.00 | 19.00 | 41.00 | 32.67 | 32.67 |
| InternVL-Chat-V1-5 | InternLM2-Chat-20B | 33.00 | 10.00 | 53.00 | 32.00 | 32.00 |
| LLaVA-Next | LLama-8B | 45.00 | 16.00 | 34.00 | 31.67 | 31.67 |
| Monkey | Qwen-7B | 33.00 | 21.00 | 25.56 | 26.55 | 26.53 |
| DeepSeek-VL | DeepSeek-LLM-7b-base | 34.00 | 8.00 | 34.00 | 25.44 | 25.36 |
| LLaVA-Next | Qwen-72B | 21.00 | 11.00 | 39.00 | 23.67 | 23.67 |
| GPT-4o | - | 19.00 | 11.00 | 41.00 | 23.67 | 23.67 |
| CogVLm2-llama3-Chat | LLama3-8B | 34.00 | 15.00 | 17.00 | 22.00 | 22.00 |
| MiniGPT-v2 | Llama 2-7B-Chat | 25.00 | 20.00 | 12.22 | 19.31 | 19.13 |
| mPLUG-DocOwl 1.5 | LLama-7B | 21.00 | 16.00 | 18.89 | 18.62 | 18.63 |
| ShareGPT4V-13B | Vicuna-13B | 25.00 | 15.00 | 12.22 | 17.59 | 17.45 |
| LLaVA1.5-13B | Vicuna-13B | 26.00 | 15.00 | 11.00 | 17.33 | 17.33 |
| SliME-13B | Vicuna-13B | 27.00 | 11.00 | 14.00 | 17.33 | 17.33 |
| ShareGPT4V-7B | Vicuna-7B | 24.00 | 15.00 | 12.22 | 17.24 | 17.12 |
| Mini-Gemini-7B-HD | Vicuna-7B-v1.5 | 24.00 | 15.00 | 12.22 | 17.24 | 17.12 |
| LLaVA1.5-7B | Vicuna-7B | 24.00 | 15.00 | 11.00 | 16.67 | 16.67 |
| MiniCPM-V 2.5 | LLama3-8B | 11.00 | 6.00 | 33.00 | 16.67 | 16.67 |
| Qwen-VL-Chat | Qwen-7B | 21.00 | 9.00 | 15.00 | 15.00 | 15.00 |
| Gemini-1.5-pro | - | 9.00 | 8.00 | 20.00 | 12.33 | 12.33 |
| TextMonkey | Qwen-7B | 11.00 | 1.00 | 23.33 | 11.38 | 11.68 |
| GPT-4o-mini | - | 3.00 | 5.00 | 14.00 | 7.33 | 7.33 |

Table 16: **Experimental results on the `Autonomous Driving` perception tasks of MME-RealWorld-CN.** Models are ranked according to their average performance. Rows corresponding to proprietary models are highlighted in gray for distinction.

| Method | LLM | Identity | Motion | | | | Traffic Signal | Object Count | Avg | Avg-C |
|---|---|---|---|---|---|---|---|---|---|---|
| | | | Vehicle | Multi-vehicle | Pedestrain | Multi-pedestrain | | | | |
| Qwen2-VL | Qwen2-7B | 40.00 | 47.00 | 38.00 | 27.00 | 17.00 | 60.00 | 29.00 | 36.86 | 36.86 |
| Monkey | Qwen-7B | 35.00 | 59.00 | 22.00 | 26.00 | 41.00 | 50.00 | 18.00 | 35.86 | 35.86 |
| DeepSeek-VL | DeepSeek-LLM-7b-base | 30.00 | 61.00 | 32.00 | 39.00 | 16.00 | 46.00 | 26.00 | 35.71 | 35.71 |
| Cambrian-1-8B | LLama3-8B-Instruct | 51.00 | 54.00 | 44.00 | 25.00 | 10.00 | 37.00 | 28.00 | 35.57 | 35.57 |
| Mini-Gemini-34B-HD | Nous-Hermes-2-Yi-34B | 49.00 | 42.00 | 43.00 | 32.00 | 21.00 | 28.00 | 25.00 | 34.29 | 34.29 |
| CogVLm2-llama3-Chat | LLama3-8B | 29.00 | 47.00 | 37.00 | 39.00 | 24.00 | 43.00 | 21.00 | 34.14 | 34.29 |
| InternVL-2 | InternLM2.5-7B-Chat | 39.00 | 43.00 | 29.00 | 37.00 | 13.00 | 48.00 | 30.00 | 34.14 | 34.14 |
| InternLM-XComposer2.5 | InternLM2-7B | 35.00 | 43.00 | 34.00 | 36.00 | 15.00 | 43.00 | 29.00 | 33.57 | 33.57 |
| SliME-13B | Vicuna-13B | 26.00 | 45.00 | 20.00 | 33.00 | 27.00 | 38.00 | 24.00 | 33.23 | 30.43 |
| Claude 3.5 Sonnet | | 50.00 | 16.00 | 38.00 | 31.00 | 31.00 | 28.00 | 31.00 | 32.43 | 32.14 |
| InternVL-Chat-V1-5 | InternLM2-Chat-20B | 41.00 | 52.00 | 32.00 | 34.00 | 15.00 | 21.00 | 30.00 | 32.14 | 32.14 |
| SliME-8B | LLama3-8B | 44.00 | 28.00 | 34.00 | 31.00 | 15.00 | 30.00 | 37.00 | 31.29 | 31.29 |
| MiniGPT-v2 | Llama 2-7B-Chat | 23.00 | 42.00 | 20.00 | 30.00 | 34.00 | 36.00 | 21.00 | 29.43 | 29.43 |
| LLaVA1.5-13B | Vicuna-13B | 23.00 | 38.00 | 20.00 | 33.00 | 27.00 | 39.00 | 22.00 | 28.66 | 28.86 |
| mPLUG-DocOwl 1.5 | LLama-7B | 22.00 | 60.00 | 27.00 | 33.00 | 28.00 | 17.00 | 12.00 | 28.43 | 28.43 |
| Mini-Gemini-7B-HD | Vicuna-7B-v1.5 | 23.00 | 37.00 | 20.00 | 33.00 | 27.00 | 35.00 | 23.00 | 28.29 | 28.29 |
| LLaVA1.5-7B | Vicuna-7B | 23.00 | 26.00 | 20.00 | 33.00 | 27.00 | 36.00 | 22.00 | 28.14 | 26.71 |
| ShareGPT4V-7B | Vicuna-7B | 23.00 | 36.00 | 20.00 | 33.00 | 27.00 | 36.00 | 22.00 | 28.14 | 28.14 |
| ShareGPT4V-13B | Vicuna-13B | 23.00 | 36.00 | 20.00 | 33.00 | 27.00 | 36.00 | 22.00 | 28.14 | 28.14 |
| YI-VL-34B | Yi-34B-Chat | 35.00 | 42.00 | 36.00 | 22.00 | 25.00 | 15.00 | 19.00 | 27.71 | 27.71 |
| Qwen-VL-Chat | Qwen | 28.00 | 40.00 | 29.00 | 16.00 | 19.00 | 28.00 | 10.00 | 27.36 | 24.29 |
| LLaVA-Next | LLama3-8B | 27.00 | 49.00 | 30.00 | 27.00 | 21.00 | 29.00 | 8.00 | 27.29 | 27.29 |
| TextMonkey | Qwen-7B | 32.00 | 36.00 | 18.00 | 31.00 | 29.00 | 32.00 | 9.00 | 26.71 | 26.71 |
| Cambrian-1-34B | Hermes2-Yi-34B | 30.00 | 39.00 | 35.00 | 31.00 | 5.00 | 11.00 | 34.00 | 26.43 | 26.43 |
| MiniCPM-V 2.5 | LLama3-8B | 40.00 | 29.00 | 39.00 | 25.00 | 17.00 | 39.00 | 18.00 | 26.00 | 29.57 |
| GPT-4o-mini | - | 15.00 | 44.00 | 21.00 | 14.00 | 9.00 | 33.00 | 32.00 | 24.00 | 24.00 |
| LLaVA-Next | Qwen-72B | 20.00 | 41.00 | 35.00 | 27.00 | 15.00 | 15.00 | 9.00 | 23.14 | 23.14 |
| GPT-4o | - | 16.00 | 20.00 | 28.00 | 24.00 | 12.00 | 20.00 | 28.00 | 21.14 | 21.14 |
| Gemini-1.5-pro | - | 32.00 | 12.00 | 29.00 | 8.00 | 11.00 | 14.00 | 19.00 | 17.57 | 17.86 |

Table 17: **Experimental results on the `Autonomous Driving` reasoning tasks of MME-RealWorld-CN.** Models are ranked according to their average performance, from highest to lowest. Rows corresponding to proprietary models are highlighted in gray for distinction.

| Method | LLM | Intention | | | Relation | | | | Attention | Avg | Avg-C |
|---|---|---|---|---|---|---|---|---|---|---|---|
| | | Ego | Pedestrian | Verhicle | Ego2P | Ego2T | Ego2V | O2O | Signal | | |
| Qwen2-VL | Qwen2-7B | 25.00 | 37.00 | 36.00 | 23.00 | 33.00 | 24.00 | 16.00 | 73.00 | 33.37 | 33.38 |
| Cambrian-1-8B | LLama3-8B-Instruct | 21.00 | 54.00 | 31.00 | 17.00 | 33.00 | 21.00 | 17.00 | 56.00 | 31.25 | 31.25 |
| InternLM-XComposer2.5 | InternLM2-7B | 19.00 | 28.00 | 33.00 | 22.00 | 37.00 | 27.00 | 18.00 | 56.00 | 30.00 | 30.00 |
| InternVL-Chat-V1-5 | InternLM2-Chat-20B | 30.00 | 25.00 | 35.00 | 26.00 | 26.00 | 24.00 | 9.00 | 64.00 | 29.88 | 29.88 |
| YI-VL-34B | Yi-34B-Chat | 30.00 | 37.00 | 32.00 | 24.00 | 28.00 | 30.00 | 15.00 | 42.00 | 29.75 | 29.75 |
| Mini-Gemini-34B-HD | Nous-Hermes-2-Yi-34B | 22.00 | 13.00 | 26.00 | 26.00 | 35.00 | 36.00 | 16.00 | 63.00 | 29.63 | 29.63 |
| DeepSeek-VL | DeepSeek-LLM-7b-base | 30.00 | 37.00 | 34.00 | 23.00 | 29.00 | 20.00 | 10.00 | 51.00 | 29.25 | 29.25 |
| SliME-8B | LLama3-8B | 33.00 | 43.00 | 31.00 | 23.00 | 19.00 | 23.00 | 9.00 | 52.00 | 29.13 | 29.13 |
| CogVLm2-llama3-Chat | LLama3-8B | 24.00 | 32.00 | 29.00 | 29.00 | 21.00 | 27.00 | 19.00 | 51.00 | 29.00 | 29.00 |
| InternVL-2 | InternLM2.5-7B-Chat | 25.00 | 33.00 | 30.00 | 26.00 | 33.00 | 26.00 | 11.00 | 48.00 | 29.00 | 29.00 |
| Monkey | Qwen-7B | 22.00 | 46.00 | 31.00 | 21.00 | 23.00 | 21.00 | 9.00 | 54.00 | 28.38 | 28.38 |
| LLaVA-Next | LLama3-8B | 27.00 | 43.00 | 34.00 | 26.00 | 24.00 | 23.00 | 11.00 | 38.00 | 28.25 | 28.25 |
| LLaVA-Next | Qwen-72B | 30.00 | 49.00 | 37.00 | 20.00 | 29.00 | 23.00 | 8.00 | 29.00 | 28.12 | 28.13 |
| GPT-4o-mini | | 11.00 | 11.00 | 24.00 | 26.00 | 39.00 | 27.00 | 20.00 | 57.00 | 26.87 | 26.88 |
| Cambrian-1-34B | Hermes2-Yi-34B | 11.00 | 10.00 | 24.00 | 22.00 | 38.00 | 29.00 | 10.00 | 57.00 | 25.12 | 25.13 |
| Claude 3.5 Sonnet | | 34.00 | 20.00 | 25.00 | 16.00 | 14.00 | 12.00 | 15.00 | 65.00 | 25.12 | 25.13 |
| Mini-Gemini-7B-HD | Vicuna-7B-v1.5 | 19.00 | 21.00 | 22.00 | 27.00 | 27.00 | 22.00 | 25.00 | 29.00 | 24.00 | 24.00 |
| MiniCPM-V 2.5 | LLama3-8B | 22.00 | 28.00 | 28.00 | 20.00 | 28.00 | 17.00 | 12.00 | 36.00 | 23.87 | 23.88 |
| SliME-13B | Vicuna-13B | 21.00 | 21.00 | 24.00 | 31.00 | 20.00 | 15.00 | 25.00 | 34.00 | 23.87 | 23.88 |
| LLaVA1.5-7B | Vicuna-7B | 19.00 | 21.00 | 22.00 | 26.00 | 27.00 | 22.00 | 25.00 | 24.00 | 23.25 | 23.25 |
| LLaVA1.5-13B | Vicuna-13B | 19.00 | 21.00 | 22.00 | 28.00 | 26.00 | 21.00 | 25.00 | 24.00 | 23.25 | 23.25 |
| ShareGPT4V-7B | Vicuna-7B | 19.00 | 21.00 | 22.00 | 26.00 | 27.00 | 22.00 | 25.00 | 24.00 | 23.25 | 23.25 |
| ShareGPT4V-13B | Vicuna-13B | 19.00 | 21.00 | 22.00 | 25.00 | 27.00 | 20.00 | 25.00 | 24.00 | 22.88 | 22.88 |
| GPT-4o | | 26.00 | 16.00 | 23.00 | 19.00 | 20.00 | 14.00 | 10.00 | 54.00 | 22.75 | 22.75 |
| TextMonkey | Qwen-7B | 21.00 | 24.00 | 26.00 | 20.00 | 20.00 | 21.00 | 9.00 | 39.00 | 22.50 | 22.50 |
| mPLUG-DocOwl 1.5 | LLama-7B | 17.00 | 22.00 | 20.00 | 20.00 | 33.00 | 21.00 | 10.00 | 36.00 | 22.38 | 22.38 |
| MiniGPT-v2 | Llama 2-7B-Chat | 21.00 | 23.00 | 22.00 | 22.00 | 31.00 | 16.00 | 10.00 | 32.00 | 22.13 | 22.13 |
| Qwen-VL-Chat | Qwen | 18.00 | 13.00 | 8.00 | 22.00 | 27.00 | 22.00 | 14.00 | 35.00 | 19.88 | 19.88 |
| Gemini-1.5-pro | | 17.00 | 12.00 | 27.00 | 5.00 | 8.00 | 10.00 | 14.00 | 32.00 | 15.67 | 15.63 |

Table 18: **Experimental results on the `Monitoring` tasks of MME-RealWorld-CN.** Models are ranked according to their average performance on perception tasks. Rows corresponding to proprietary models are highlighted in gray for distinction.

| Method | LLM | Perception | | | | | | | Reasoning | | | | |
|---|---|---|---|---|---|---|---|---|---|---|---|---|---|
| | | Vehicle | | | Pedestrain | | Avg | Avg-C | Calculate | Intention | Property | Avg | Avg-C |
| | | Counting | Location | Attribute | Counting | Attribute | | | | | | | |
| InternVL-2 | InternLM2.5-7B-Chat | 70.07 | 25.74 | 28.98 | 59.68 | 12.04 | 39.30 | 39.30 | 51.67 | 21.43 | 41.00 | 38.14 | 38.03 |
| LLaVA-Next | LLama3-8B | 62.00 | 14.00 | 33.00 | 38.00 | 30.00 | 35.40 | 35.40 | 13.00 | 45.92 | 30.00 | 29.53 | 29.64 |
| LLaVA-Next | Qwen-72B | 72.00 | 11.00 | 26.00 | 35.00 | 29.00 | 34.60 | 34.60 | 24.00 | 13.27 | 45.00 | 27.52 | 27.42 |
| InternVL-Chat-V1-5 | InternLM2-Chat-20B | 62.00 | 22.00 | 20.00 | 47.00 | 11.00 | 32.40 | 32.40 | 25.00 | 11.00 | 54.00 | 30.13 | 30.00 |
| Qwen2-VL | Qwen2-7B | 63.00 | 18.00 | 15.00 | 43.00 | 8.00 | 29.40 | 29.40 | 27.00 | 17.35 | 57.00 | 33.89 | 33.78 |
| SliME-8B | LLama3-8B | 55.00 | 18.00 | 17.00 | 40.00 | 16.00 | 29.20 | 29.20 | 24.00 | 31.63 | 36.00 | 30.54 | 30.54 |
| Mini-Gemini-34B-HD | Nous-Hermes-2-Yi-34B | 62.00 | 18.00 | 12.00 | 37.00 | 10.00 | 27.80 | 27.80 | 12.00 | 14.29 | 50.00 | 25.50 | 25.43 |
| Cambrian-1-8B | LLama3-8B-Instruct | 61.00 | 22.00 | 13.00 | 36.00 | 6.00 | 27.60 | 27.60 | 30.00 | 21.43 | 42.00 | 31.21 | 31.14 |
| Cambrian-1-34B | Hermes2-Yi-34B | 62.00 | 23.00 | 7.00 | 39.00 | 7.00 | 27.60 | 27.60 | 11.00 | 15.31 | 33.00 | 19.80 | 19.77 |
| mPLUG-DocOwl 1.5 | LLama-7B | 56.00 | 19.00 | 29.00 | 24.00 | 10.00 | 25.60 | 25.60 | 4.00 | 17.35 | 38.00 | 19.80 | 19.78 |
| YI-VL-34B | Yi-34B-Chat | 57.00 | 7.00 | 18.00 | 31.00 | 11.00 | 25.20 | 24.80 | 24.00 | 26.50 | 49.00 | 33.21 | 33.17 |
| GPT-4o | - | 61.00 | 13.00 | 8.00 | 37.00 | 7.00 | 25.20 | 25.20 | 10.00 | 8.16 | 44.00 | 20.80 | 20.72 |
| Gemini-1.5-pro | - | 63.00 | 13.00 | 9.00 | 38.00 | 3.00 | 25.20 | 25.20 | 9.00 | 10.20 | 25.00 | 14.76 | 14.73 |
| Claude 3.5 Sonnet | | 62.00 | 10.00 | 10.00 | 34.00 | 9.00 | 25.00 | 25.00 | 23.00 | 22.45 | 49.00 | 31.54 | 31.48 |
| CogVLm2-llama3-Chat | LLama3-8B | 32.00 | 26.00 | 22.00 | 33.00 | 11.00 | 24.80 | 24.80 | 27.00 | 37.76 | 47.00 | 37.25 | 37.25 |
| GPT-4o-mini | - | 60.00 | 6.00 | 8.00 | 35.00 | 1.00 | 22.00 | 22.00 | 8.00 | 14.29 | 22.00 | 14.77 | 14.76 |
| DeepSeek-VL | DeepSeek-LLM-7b-base | 62.00 | 6.00 | 11.00 | 24.00 | 5.00 | 21.60 | 21.60 | 9.00 | 15.31 | 52.00 | 25.50 | 25.44 |
| MiniCPM-V 2.5 | LLama3-8B | 55.00 | 15.00 | 11.00 | 17.00 | 4.00 | 20.40 | 20.40 | 11.00 | 11.22 | 27.00 | 16.44 | 16.41 |
| Qwen-VL-Chat | Qwen-7B | 58.00 | 11.00 | 8.00 | 0.00 | 24.00 | 20.20 | 20.20 | 7.00 | 12.00 | 21.00 | 13.34 | 13.33 |
| TextMonkey | Qwen-7B | 57.00 | 4.00 | 13.00 | 24.00 | 0.00 | 19.60 | 19.60 | 1.00 | 12.24 | 23.00 | 12.08 | 12.08 |
| InternLM-XComposer2.5 | InternLM2-7B | 60.00 | 5.00 | 8.00 | 24.00 | 0.00 | 19.40 | 19.40 | 0.00 | 9.18 | 41.00 | 16.78 | 16.73 |
| Monkey | Qwen-7B | 12.00 | 40.00 | 22.00 | 20.00 | 2.00 | 19.20 | 19.20 | 23.00 | 24.49 | 50.00 | 32.55 | 32.50 |
| SliME-13B | Vicuna-13B | 17.00 | 21.00 | 16.00 | 23.00 | 12.00 | 17.80 | 17.80 | 26.00 | 27.55 | 38.00 | 30.54 | 30.52 |
| LLaVA1.5-13B | Vicuna-13B | 16.00 | 21.00 | 17.00 | 20.00 | 11.00 | 17.00 | 17.00 | 26.00 | 23.47 | 28.00 | 25.84 | 25.82 |
| ShareGPT4V-13B | Vicuna-13B | 15.00 | 21.00 | 17.00 | 20.00 | 11.00 | 16.80 | 16.80 | 26.00 | 23.47 | 25.00 | 24.83 | 24.82 |
| Mini-Gemini-7B-HD | Vicuna-7B-v1.5 | 15.00 | 21.00 | 17.00 | 20.00 | 11.00 | 16.80 | 16.80 | 26.00 | 23.47 | 27.00 | 25.50 | 25.49 |
| LLaVA1.5-7B | Vicuna-7B | 15.00 | 21.00 | 16.00 | 20.00 | 11.00 | 16.60 | 16.60 | 26.00 | 23.47 | 26.00 | 25.17 | 25.16 |
| ShareGPT4V-7B | Vicuna-7B | 15.00 | 21.00 | 16.00 | 20.00 | 11.00 | 16.60 | 16.60 | 26.00 | 23.47 | 25.00 | 24.83 | 24.82 |
| MiniGPT-v2 | Llama 2-7B-Chat | 9.00 | 13.00 | 16.00 | 23.00 | 11.00 | 14.40 | 14.40 | 16.00 | 18.37 | 25.00 | 19.80 | 19.79 |

Table 19: **Performance comparison of various MLLMs across perception and reasoning tasks on MME-RealWorld-lite.** Metrics include task-specific scores (e.g., OCR, RS, DT, MO, AD) and their averages.

| Method | LLM | Overall | Perception | | | | | | Reasoning | | | | |
|---|---|---|---|---|---|---|---|---|---|---|---|---|---|
| | | | OCR | RS | DT | MO | AD | Avg | OCR | DT | MO | AD | Avg |
| GPT-4o | - | 46.4 | 80.8 | 45.3 | 65.0 | 33.9 | 37.4 | 49.1 | 72.0 | 50.0 | 42.0 | 32.8 | 42.1 |
| GPT-4o-mini | - | 37.4 | 69.9 | 23.3 | 62.0 | 19.4 | 34.3 | 38.8 | 57.0 | 39.0 | 18.7 | 35.0 | 35.2 |
| Intern-VL-2.5 | InternLM2.5-7B-Chat | 47.1 | 81.2 | 46.0 | 79.0 | 29.5 | 38.9 | 49.7 | 68.0 | 53.0 | 40.7 | 35.0 | 42.9 |
| Qwen2-VL | Qwen2-7B | 46.7 | 85.6 | 40.0 | 74.0 | 27.9 | 36.0 | 48.2 | 73.0 | 46.0 | 47.3 | 35.8 | 44.4 |
| LLaVA-OV | Qwen2-7B | 48.5 | 82.0 | 50.7 | 64.0 | 34.2 | 44.9 | 52.8 | 71.0 | 43.0 | 44.7 | 34.8 | 42.7 |
| Slime | Llama3-8B | 37.1 | 57.6 | 36.0 | 51.0 | 29.2 | 32.9 | 37.7 | 51.0 | 27.0 | 40.7 | 33.5 | 36.4 |

Table 20: **Exploring the Limitations of Simple Image Segmentation for Large Image Perception.** The results are evaluated on the perception tasks of MME-RealWorld.

| Method | LLM | Perception | | | | | | |
|--------|-----|------|------|------|------|------|------|-------|
| | | OCR | RS | DT | MO | AT | Avg | Avg-C |
| SliME-8B | LLama3-8B | 53.45 | 42.27 | 29.34 | 40.62 | 33.66 | 40.29 | 39.87 |
| Split Img | LLama3-8B | 53.66 | 35.26 | 31.11 | 39.07 | 30.57 | 38.84 | 37.93 |
| Qwen2-VL | Qwen2-7B | 81.38 | 44.81 | 70.18 | 37.3 | 34.62 | 58.96 | 53.66 |
| Split Img | Qwen2-7B | 67.8 | 36.3 | 49.33 | 34.97 | 34.21 | 47.91 | 44.52 |
| InternVL-2 | InternLM2.5-7B-Chat | 73.92 | 39.35 | 62.80 | 53.19 | 35.46 | 55.82 | 52.94 |
| Split Img | InternLM2.5-7B-Chat | 61.43 | 33.09 | 44.19 | 34.88 | 31.58 | 43.75 | 41.03 |

Table 21: **Performance of different models on intention prediction tasks**.

| Method | Eng | CN | AD-Intention-Eng | | | AD-Intention-CN | | | Avg |
|--------|-----|-----|-----|-----|-----|-----|-----|-----|-----|
| | Monitoring | Monitoring | Ego | Pedestrian | Verhicle | Ego | Pedestrian | Verhicle | |
| GPT-4o | 13.27 | 8.16 | 17.11 | 19.42 | 27.54 | 26.00 | 16.00 | 23.00 | 18.81 |
| Claude 3.5 Sonnet | 18.37 | 22.45 | 26.32 | 32.04 | 24.64 | 34.00 | 20.00 | 25.00 | 25.35 |
| InternVL-2 | 21.43 | 21.43 | 24.01 | 43.69 | 32.85 | 25.00 | 33.00 | 30.00 | 28.93 |
| Qwen2-VL | 19.39 | 17.35 | 19.08 | 43.69 | 35.75 | 25.00 | 37.00 | 36.00 | 29.16 |

**Limitations of MLLMs in Understanding Dynamic Information.** In combination with the intention prediction results from autonomous driving and monitoring tasks (Tab. 21), we observe that MLLMs exhibit significant deficiencies in understanding, predicting, and reasoning about the dynamic information of objects, such as predicting the steering of a car. Although the input to these models is a single frame image rather than a video, there remains a considerable gap between their performance and that of humans. Therefore, it seems that these MLLMs are still far from having the capability to be world models.

**Chain-of-Thought Reasoning does not enhance the model's high-resolution perception capabilities.** As shown in Fig. 16, we further evaluate the impact of CoT reasoning on high-resolution tasks (for both MCQ and the EM mentioned above). Tab. 23 demonstrates that while CoT reasoning can assist the model in reasoning tasks, especially in chart-related reasoning tasks, it provides minimal benefit for perception tasks (line 6 vs. line 7 and line 8 vs. line 11). In fact, for models like GPT-4o-mini, perception performance even declines. This finding suggests that high-resolution perception tasks are already highly challenging on their own. Recent work MVoT (Li et al., 2025) investigates multimodal CoT, opening new avenues for future research. However, if the model cannot effectively receive and understand visual inputs, then even the strongest reasoning capabilities of an LLM will only lead to limited improvements. Therefore, enhancing MLLM's comprehension of visual inputs remains a critical focus.

**What matters for ultra-resolution image perception: LLM capability or visual cognition capability?** It is indeed challenging to completely separate a model's capability to process input images from its inherent perceptual ability, as these two aspects are closely coupled. However, in the case of high-resolution images, the model's capacity to process inputs seems especially crucial. For example, as shown in the Tab. 24, Mini-Gemini-7B-HD and LLaVA1.5-7B use similar LLM architectures and have comparable training data, yet Mini-Gemini-7B-HD exhibits far superior high-resolution perception capabilities. This demonstrates the critical importance of handling higher-resolution data effectively. As a result, most modern MLLMs have incorporated various image-splitting strategies to accommodate larger maximum resolutions. Nevertheless, simply supporting higher resolution is not a complete solution to high-resolution perception challenges. For instance, while Intern-VL2 has a higher input resolution limit than Qwen2-VL, its overall performance is slightly lower. This implies that the ability to handle larger image resolutions alone is insufficient for robust high-resolution perception. The model's inherent capabilities (such as information extraction and comprehension) also play a vital role.

**Instruction Following Abilities.** As described in Sec. 2.1, our prompts specify that the model should directly select and output a single answer. In this regard, closed-source models generally perform better, with outputs being more concise and directly aligned with the instructions. However,

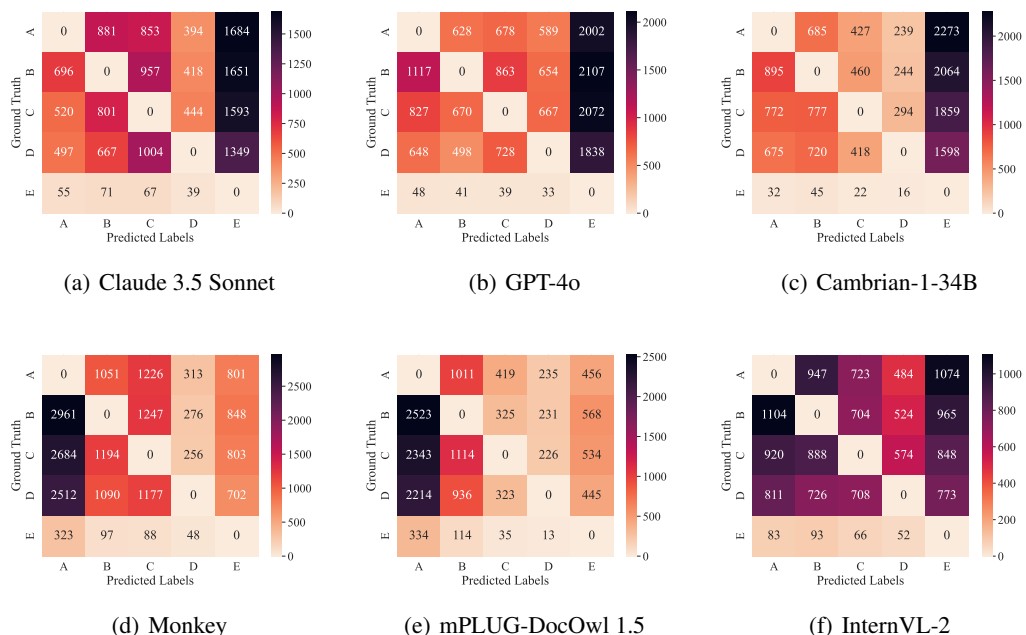

Figure 13: **Distribution of incorrec choices.** The matrix reveals distinct response behaviors among different MLLMs. Larger models tend to select the safer option "E", while smaller models exhibit a bias toward the first option "A". InternVL-2, however, shows a unique uniform error distribution.

we have observed that some open-source models do not strictly adhere to our queries and generate a significant amount of additional analysis. Sometimes, they even produce outputs that are excessively verbose, continuing until the token count reaches the predefined maximum limit. This indicates that the open-source models have a lot of room for optimization in the ability of instruction following.

**Preparatory Models have better AI security.** We designed three metrics to assess the frequency and performance of models when predicting "E" (indicating refusal to answer):

- **Correct E (%):** This metric measures the percentage of times a model correctly predicts "E" when the ground truth is "E," helping evaluate whether the model can recognize questions that genuinely require a refusal to respond.
- **E Ratio in Wrong Predictions (%):** This indicates the proportion of wrong answers labeled as "E" among all incorrect predictions, offering insights into the model's tendency to choose "E" when uncertain.
- **# Predicted E:** This counts the total number of times a model predicts "E" within the current split, providing a straightforward view of "E" predictions overall.

Finally, **Error Ratio (%)** reflects the model's perceptual accuracy within the domain. Below are the summarized results from Table. 22, from which we draw several key insights:

(a) Less capable models, such as SliME-8B, rarely opt for "E" and instead tend to select answers they consider correct, leading to low rates of both correct "E" predictions and "E" frequency across domains.

(b) More advanced open-source models, like InternVL, show "E" frequencies comparable to closed-source models on simpler tasks, such as OCR or Diagram and Table interpretation. In these easier tasks, most models are confident in their answers, so the frequency of selecting "E" remains low.

(c) On more challenging tasks, such as Remote Sensing and Monitoring, all models have higher error rates. Notably, proprietary models like GPT-4o and Claude35 exhibit both higher accuracy in predicting "E" and a greater frequency of selecting "E" compared to

Table 22: In-depth comparison of metrics associated with different models' tendencies to select response 'E'.

| Remote sensing | | | | |
|---|---|---|---|---|
| **Model** | **Correct E (%)** | **E Ratio in Wrong Prediction (%)** | **# Predicted E** | **Error Ratio (%)** |
| GPT-4o | 50.00% | 56.19% | 1500 | 71.08% |
| Claude35 | 71.43% | 63.00% | 1759 | 74.26% |
| SliME | 7.14% | 9.13% | 198 | 57.73% |
| InternVL | 21.43% | 18.31% | 418 | 60.65% |
| Monitoring | | | | |
| **Model** | **Correct E (%)** | **E Ratio in Wrong Prediction (%)** | **# Predicted E** | **Error Ratio (%)** |
| GPT-4o | 94.18% | 47.39% | 1138 | 67.50% |
| Claude35 | 97.26% | 48.96% | 1206 | 69.89% |
| SliME | 85.96% | 18.23% | 571 | 65.11% |
| InternVL | 91.44% | 31.71% | 800 | 62.39% |
| OCR | | | | |
| **Model** | **Correct E (%)** | **E Ratio in Wrong Prediction (%)** | **# Predicted E** | **Error Ratio (%)** |
| GPT-4o | 15.07% | 18.57% | 284 | 23.56% |
| Claude35 | 13.70% | 38.02% | 686 | 28.49% |
| SliME | 0.00% | 3.86% | 51 | 50.32% |
| InternVL | 21.92% | 28.19% | 498 | 27.40% |
| Diagram and Table | | | | |
| **Model** | **Correct E (%)** | **E Ratio in Wrong Prediction (%)** | **# Predicted E** | **Error Ratio (%)** |
| GPT-4o | 11.76% | 24.84% | 763 | 53.65% |
| Claude35 | 5.88% | 4.10% | 78 | 32.92% |
| SliME | 0.00% | 9.30% | 39 | 70.65% |
| InternVL | 23.53% | 29.58% | 692 | 39.15% |

Table 23: **Performance of different models under various evaluation metrics**, with input-output formats for each metric shown in Figure 16. Removing choices (EM) significantly reduces model performance; while using GPT-4o for matching model responses helps somewhat, overall accuracy remains low. CoT reasoning benefits reasoning tasks but has minimal impact on perception tasks.

| Line | Metric | Method | Perception | | | | | | Reasoning | | | | |
|---|---|---|---|---|---|---|---|---|---|---|---|---|---|
| | | | OCR | RS | DT | MO | AD | Avg | OCR | DT | MO | AD | Avg |
| 1 | MCQ | SliME | 58 | 36 | 51 | 29 | 33 | 37.7 | 51 | 27 | 41 | 34 | 36.4 |
| 2 | EM | Exact Match | 10 | 13 | 2 | 15 | 11 | 11.3 | 2 | 3 | 8 | 0 | 2.3 |
| 3 | MCQ | LLaVA-OV | 82 | 51 | 64 | 34 | 45 | 52.8 | 71 | 43 | 45 | 35 | 42.7 |
| 4 | EM | Exact Match | 34 | 15 | 25 | 13 | 13 | 18.8 | 4 | 20 | 13 | 0 | 5.8 |
| 5 | MCQ | GPT-4o-mini | 70 | 23 | 62 | 19 | 34 | 38.8 | 57 | 39 | 19 | 35 | 35.2 |
| 6 | EM | Exact Match | 39 | 11 | 30 | 6 | 14 | 11.3 | 21 | 33 | 4 | 7 | 11.9 |
| 7 | CoT | CoT-MCQ | 67 | 22 | 57 | 21 | 31 | 36.7 | 60 | 51 | 34 | 33 | 39.1 |
| 8 | MCQ | GPT-4o | 81 | 45 | 65 | 34 | 37 | 49.1 | 72 | 50 | 42 | 33 | 42.1 |
| 9 | EM | Exact Match | 43 | 20 | 29 | 9 | 20 | 22.8 | 28 | 40 | 5 | 9 | 15.1 |
| 10 | EM | Machine match | 53 | 31 | 46 | 16 | 32 | 33.1 | 61 | 45 | 22 | 27 | 33.0 |
| 11 | CoT | CoT-MCQ | 83 | 53 | 63 | 39 | 40 | 49.3 | 75 | 66 | 41 | 41 | 49.3 |
| 12 | CoT-EM | Machine match | 55 | 30 | 45 | 25 | 29 | 35.0 | 67 | 52 | 28 | 29 | 36.8 |

InternVL, which maintains a relatively low "E" frequency (10%-35%) even on difficult tasks.

In summary, for simpler multimodal tasks, the safety profiles of advanced MLLMs and proprietary models are comparable. However, on more complex tasks, proprietary models demonstrate a significantly higher level of safety by opting for "E" when uncertain, aligning better with human values by avoiding misleading answers. Given that open-source models currently undergo limited alignment with human preferences, this presents an important direction for future research and development.

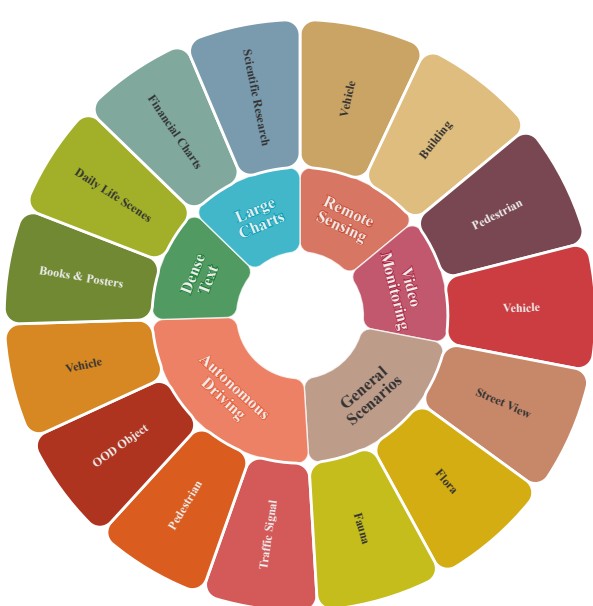

Figure 14: **Diverse data sources encompassing over 15 basic image categories**, including text-dense scenes, autonomous driving, video surveillance, natural environments, and more, highlighting a wide range of scene diversity.

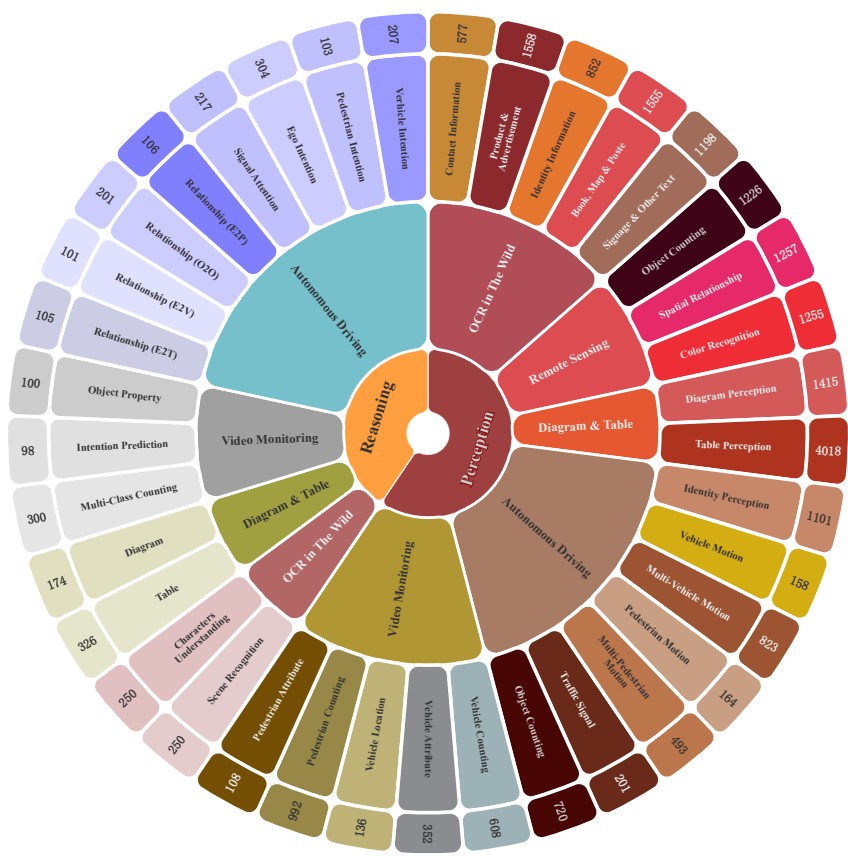

Figure 15: **Detailed information on task distributions across different domains**.

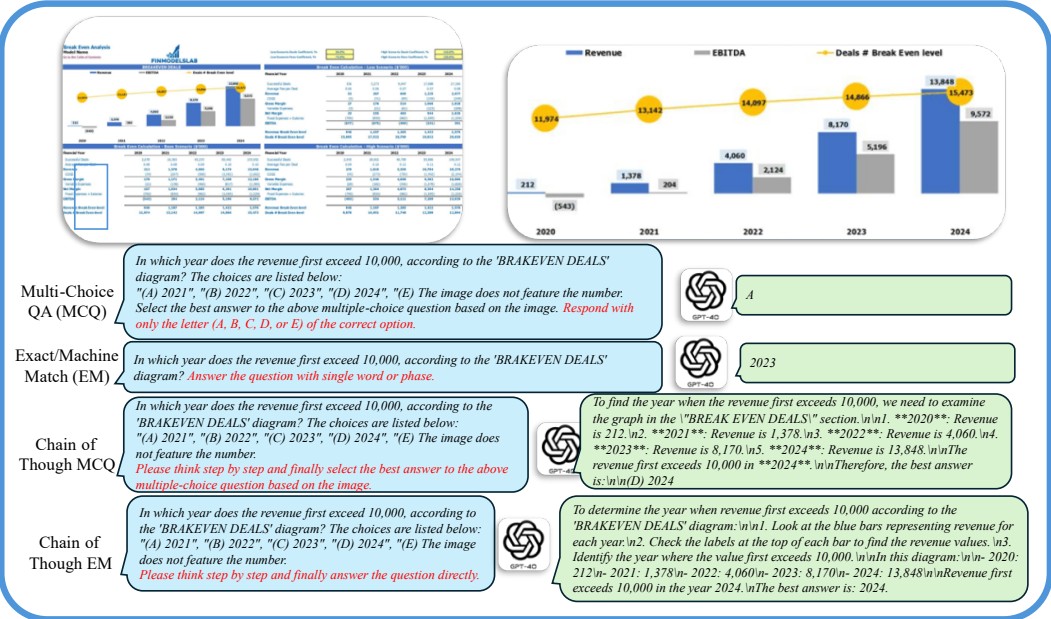

Figure 16: Illustration of input-output formats for various experimental setups. MCQ represents the default multiple-choice question format in the dataset. EM removes the choices to prevent the model from relying on choice information, instead directly matching the final answer. CoT (Chain of Thought) prompts the model to first perform reasoning before providing the final answer.

Table 24: Maximum Acceptable Resolution of Different MLLMs.

| Model | LLM | Max Resolution |
|---|---|---|
| Qwen-VL-Chat | Qwen | 448 |
| LLaVA1.5-7B | Vicuna-7B | 336 |
| LLaVA1.5-13B | Vicuna-13B | 336 |
| LLaVA-Next | LLama3-8B | 672 |
| LLaVA-Next | Qwen-72B | 672 |
| mPLUG-DocOwl 1.5 | LLama-7B | 448 |
| ShareGPT4V-7B | Vicuna-7B | 336 |
| ShareGPT4V-13B | Vicuna-13B | 336 |
| MiniGPT-v2 | Llama 2-7B-Chat | 448 |
| Monkey | Qwen-7B | 896*1334 |
| Cambrian-1-8B | LLama3-8B-Instruct | 1024 |
| Cambrian-1-34B | Hermes2-Yi-34B | 1024 |
| DeepSeek-VL | DeepSeek-LLM-7b-base | 1024 |
| YI-VL-34B | Yi-34B-Chat | 448 |
| MiniCPM-V 2.5 | LLama3-8B | 1344 |
| InternLM-XComposer2.5 | InternLM2-7B | 4096 |
| CogVLm2-llama3-Chat | LLama3-8B | 1344 |
| Mini-Gemini-7B-HD | Vicuna-7B-v1.5 | 672 |
| Mini-Gemini-34B-HD | Nous-Hermes-2-Yi-34B | 672 |
| SliME-13B | Vicuna-13B | 2016 |
| SliME-8B | LLama3-8B | 2016 |
| InternVL-Chat-V1-5 | InternLM2-Chat-20B | 4096 |
| InternVL-2 | InternLM2.5-7B-Chat | 4096 |
| Qwen2-VL | Qwen2-7B | 3584 |
| GPT-4o | - | Private |
| GPT-4o-mini | - | Private |
| Gemini-1.5-pro | - | Private |
| Claude 3.5 Sonnet | - | Private |

