# OpenReview forum: "MME-RealWorld: Could Your Multimodal LLM Challenge High-Resolution Real-World Scenarios that are Difficult for Humans?"
_ICLR.cc/2025/Conference — ICLR 2025 Poster_

### Official Review · Reviewer_FpJY · 2024-11-01

**Soundness:** 3
**Presentation:** 3
**Contribution:** 3
**Rating:** 6
**Confidence:** 4

**Summary:**

This paper introduces a new benchmark dataset, MME-RealWorld, designed to evaluate the capabilities of Multimodal Large Language Models (MLLMs) in handling high-resolution, real-world scenarios. The dataset addresses limitations in existing benchmarks, such as small data scale, reliance on model-based annotations, and insufficient task difficulty. MME-RealWorld contains over 13,000 high-quality, high-resolution images annotated with 29,429 question-answer pairs across 43 subtasks and 5 real-world scenarios, making it the largest manually annotated benchmark to date. The paper reports the performance of 29 prominent MLLMs on this benchmark, revealing that even the most advanced models struggle to achieve 60% accuracy, indicating significant challenges in perceiving high-resolution images and understanding complex real-world scenarios.

**Strengths:**

- 1) The MME RealWorld proposed in this paper has significant advantages in terms of data scale, annotation quality, visual content resolution, language types, task types, and task domain diversity, filling the gaps in existing work. It emphasizes the relevance of benchmarks and the real world, providing a strong and persuasive benchmark for evaluating the visual-language abilities of multimodal agents in real-world application-related scenarios.

- 2) The experimental section of this paper is solid. The authors present rich test results and analysis, providing various statistical indicators to reveal the limitations of existing VLMs in fine-grained image perception and dynamic information understanding, as well as biases of models from different sources in visual-language tasks. This offers valuable insights for improving the performance of VLMs in various application scenarios.

**Weaknesses:**

- 1) The main challenges of MME RealWorld stem from high-resolution images and complex content. However, the corresponding questions are only focused on image content recognition and simple single-step reasoning, showing limitations in task difficulty and the requirement for understanding capabilities of large models.

- 2) Some methods displayed on the leaderboard are restricted by fixed input resolutions. In high-resolution scenarios, directly resizing input images may result in the loss of information needed to answer questions. Therefore, is the model's error due to the inability to find the correct information from complex image content or because the necessary information was not provided at the input stage? Supplementing such discussions can further enhance the persuasiveness of the paper.

**Questions:**

- 1) In line 315, the paper says "InternVL-2 demonstrates the strongest perception abilities, outperforming other closed-source models.", but Tab.3 shows that Qwen2-VL is the best performing model. So is there a typo here? Please point out if I've misunderstood.

- 2) It is recommended to add the max resolution supported by the baseline model to the table to enhance readability.

---

> ### Author Response · Authors · 2024-11-14
> **Response to the reviewer [Concern 1]**
>
> **Concern 1  However, the corresponding questions are only focused on image content recognition and simple single-step reasoning, showing limitations in task difficulty and the requirement for understanding capabilities of large models.**
>
> Thank you for your valuable feedback. You are correct that the current questions are relatively simple, primarily focused on image content recognition and basic single-step reasoning. However, **the poor performance of current MLLMs on these tasks highlights that these models still have a long way to go** in terms of high-resolution perception and general reasoning capabilities. Once these simpler problems are better addressed, it will be more valuable to explore more complex question forms.
>
>
> We also appreciate your suggestion, and **we are indeed working on designing more challenging tasks and evaluation metrics**. Over the past period, we have also been considering other question formats. For example, we are preparing to release an alternative evaluation version of the dataset, where only the questions are provided without answer choices. In this setup, we will assess the models’ performance using exact match or GPT-match methods, which will prevent models from relying on the provided choices.
>
> In our initial tests, we selected 50 samples from each task and used the prompt “Please respond to the question with a single word or phrase,” encouraging models to generate direct answers. The table below shows the results of these tests, where all models showed significant performance degradation without the choices:
>
> |      Method     | Perception |    |    |    |    |      |
> |:---------------:|:----------:|:--:|:--:|:--:|:--:|:----:|
> |                 |     OCR    | RS | DT | MO | AD |  Avg |
> |      Slime      |     58     | 36 | 51 | 29 | 33 | 37.7 |
> |    wo choice    |     10     | 13 |  2 | 15 | 11 | 11.3 |
> |     LLaVA-OV    |     82     | 51 | 64 | 34 | 45 | 52.8 |
> |    wo choice    |     34     | 15 | 25 | 13 | 13 | 18.8 |
> |      GPT-4o     |     81     | 45 | 65 | 34 | 37 | 49.1 |
> |    wo choice    |     43     | 20 | 29 |  9 | 20 | 22.8 |
> |+Machine match |     53     | 31 | 46 | 16 | 32 | 33.1 |
>
> As you can see, when no choices are provided, the performance significantly drops. However, we also observed that for tasks like OCR, where the answer is fairly unique, exact match methods are likely to yield correct results. For tasks with more open-ended responses (e.g., "Where is the person in yellow clothes located in the image?"), it is much more difficult to assess accuracy using exact matches, as responses like "top-right of the image" and "in front of the store" could both be correct.
>
> To address this challenge, we have incorporated GPT-4o to align the model's responses with the correct answer in terms of meaning (shown in the last row). This alignment approach has led to a performance improvement, but it still lags behind results with the choice-based setup. Nonetheless, we believe that the "Machine match" strategy, which involves aligning model outputs with the intended meaning, will be an important evaluation approach moving forward. Under this strategy, GPT-4o’s performance only reaches about 30%, further increasing the difficulty of our questions.
>
> We will continue to explore additional evaluation strategies in the future. Once again, thank you for your constructive suggestions!

---

> > ### Author Response · Authors · 2024-11-14
> > **Response to the reviewer [Concern 2-4]**
> >
> > **Concern 2 Therefore, is the model's error due to the inability to find the correct information from complex image content or because the necessary information was not provided at the input stage?**
> >
> > Thank you for this valuable suggestion! We plan to add the following discussion to the paper.
> >
> > It is indeed challenging to fully separate the model’s ability to accept input images from its intrinsic perception capability, as these two aspects are closely intertwined. However, based on current findings, the ability to effectively handle input appears to be particularly crucial for high-resolution images. For example, in the table below, both Mini-Gemini-7B-HD and LLaVA1.5-7B use similar LLM architectures and have similar training data, yet Mini-Gemini-7B-HD demonstrates significantly stronger high-resolution perception capabilities. This indicates that handling higher-resolution data is essential. To address this, many MLLMs now employ various image-splitting strategies to increase the maximum resolution they can process.
> >
> > Nevertheless, simply improving resolution perception does not fully solve the problem. For example, while Intern-VL2 has a higher maximum input resolution than Qwen2-VL, its overall performance is slightly inferior. This suggests that the ability to process higher-resolution images alone does not completely address the high-resolution perception problem. The model's ability to extract and understand information is equally crucial.
> >
> > Additionally, as mentioned in the paper, the performance of almost all MLLMs on MME-RealWorld is still not good enough, and we also discussed the computational efficiency issues. Efficiently processing ultra-high-resolution images remains an open question in the field.
> >
> > To aid in comparison, we have included the maximum resolution for several models in the table below:
> >
> > |         Model         |          LLM         | Max Resolution |
> > |:---------------------:|:--------------------:|:--------------:|
> > |      Qwen-VL-Chat     |         Qwen         |       448      |
> > |      LLaVA1.5-7B      |       Vicuna-7B      |       336      |
> > |      LLaVA1.5-13B     |      Vicuna-13B      |       336      |
> > |       LLaVA-Next      |       LLama3-8B      |       672      |
> > |       LLaVA-Next      |       Qwen-72B       |       672      |
> > |    mPLUG-DocOwl 1.5   |       LLama-7B       |       448      |
> > |     ShareGPT4V-7B     |       Vicuna-7B      |       336      |
> > |    ShareGPT4V-13B     |      Vicuna-13B      |       336      |
> > |       MiniGPT-v2      |   Llama 2-7B-Chat    |       448      |
> > |         Monkey        |        Qwen-7B       |    896*1334    |
> > |     Cambrian-1-8B     |  LLama3-8B-Instruct  |      1024      |
> > |     Cambrian-1-34B    |    Hermes2-Yi-34B    |      1024      |
> > |      DeepSeek-VL      | DeepSeek-LLM-7b-base |      1024      |
> > |       YI-VL-34B       |      Yi-34B-Chat     |       448      |
> > |     MiniCPM-V 2.5     |       LLama3-8B      |      1344      |
> > | InternLM-XComposer2.5 |     InternLM2-7B     |      4096      |
> > |  CogVLm2-llama3-Chat  |       LLama3-8B      |      1344      |
> > |   Mini-Gemini-7B-HD   |    Vicuna-7B-v1.5    |       672      |
> > |   Mini-Gemini-34B-HD  | Nous-Hermes-2-Yi-34B |       672      |
> > |       SliME-13B       |      Vicuna-13B      |      2016      |
> > |        SliME-8B       |       LLama3-8B      |      2016      |
> > |   InternVL-Chat-V1-5  |  InternLM2-Chat-20B  |      4096      |
> > |       InternVL-2      |  InternLM2.5-7B-Chat |      4096      |
> > |        Qwen2-VL       |        Qwen-7B       |      3584      |
> >
> > ----
> >
> > **Concern 3 InternVL-2 demonstrates the strongest perception abilities, outperforming other closed-source models**
> >
> > Thank you for this suggestion. We have revised the manuscript accordingly to reflect these details.
> >
> > ----
> >
> > **Concern 4: "It is recommended to add the max resolution supported by the baseline model to the table to enhance readability."**
> >
> > We have addressed this recommendation in our response to Concern 2 by adding the maximum resolution supported by various models, providing a clear reference for readers.

---

> > > ### Comment · Reviewer_FpJY · 2024-11-19
> > > **Thanks for your reply**
> > >
> > > Thank you for your response. Q2 has been resolved. Indeed, as you mentioned, both input resolution perception and content comprehension need to be improved simultaneously, and it is necessary to try to find the trade-off in this process.

---

> > > ### Comment · Reviewer_FpJY · 2024-11-19
> > > **Improve my scoring**
> > >
> > > Overall, the authors' responses have addressed my concerns, and I agree that this submission presents a valuable dataset in terms of data scale, annotation quality, visual content resolution, language types, task types, and task domain diversity. Thus, I  decided to improve my rating.

---

### Official Review · Reviewer_qNEP · 2024-11-03

**Soundness:** 3
**Presentation:** 3
**Contribution:** 2
**Rating:** 6
**Confidence:** 3

**Summary:**

In this paper, the authors present the MME-RealWorld dataset which stands out for a few key reasons. First, it’s the largest human-annotated benchmark for real-world scenarios with nearly 30,000 QA pairs created by 32 volunteers. The data is high quality, featuring high-resolution images that capture important details and every annotation was double-checked by a professional team. The tasks in this dataset are challenging, reflecting real-world needs that even top models struggle to handle accurately. The authors also include a Chinese section in their dataset with 5,917 QA pairs to avoid translation and cultural issues.

In addition, the authors evaluate a total of 24 open-source MLLMs (some of which are public APIs) on QA pairs that emphasize perception capabilities, reasoning, and a focus on Chinese. They share insights into the strengths and limitations of current models, showing that even the most advanced ones struggle with these benchmarks, with the top ones achieving high 50s%.

**Strengths:**

The dataset is a comprehensive and high-quality collection of realworld QA pairs manually annotated to capture complex details and ensure accuracy. It features high resolution images essential for interpreting information in certain domains like MO, with annotations rigorously cross-checked by small group of professionals. The dataset includes particularly challenging, real-world tasks, where top models struggle to handle, with performance generally falling below 60% accuracy. Additionally, a dedicated Chinese section addresses translation and cultural gaps often seen in other datasets. Finally, an evaluation of numerous open-source models highlights current limitations in handling such complex, real-world scenarios.

**Weaknesses:**

The dataset is certainly valuable and a step forward compared to existing QA-focused benchmarks. However, the contribution may be somewhat limited for publication in this conference due to a few areas. For instance, the diversity of data sources appears limited, with an overemphasis on specific tasks like autonomous driving (If the goal is to capture embodied understanding, related areas in robotics could also be included to broaden the dataset’s scope) Similarly, the monitoring section could benefit from a more varied range of examples and so on. There is not a discussion on why these areas are chosen over others. While tackling everything can be challenging, my worry is that this dataset will introduce bias toward certain domains while trying to address limitations of other benchmarks.

The use of multiple-choice challenges, while popular, remains a somewhat limited way to assess model capabilities. A more open-ended evaluation method would provide a richer assessment of model understanding, especially on this complex benchmark. The addition of a fifth option in multiple-choice questions is a positive step, but more could be done to move beyond predefined answers.

Additionally, the reliance on full human annotation restricts scalability. It would be beneficial if the authors leveraged this benchmark as a foundation to extend to larger datasets and broader domains.

Finally, while English and Chinese are the primary languages in current benchmarks, including more languages would strengthen the dataset’s multilingual utility.

I clearly see the values of the work, however, I think it might be better suited for a dataset track or even a relevant workshop focused on dataset, evaluation.

**Questions:**

Do authors have plans or see it feasible to increase diversity of data sources and tasks? Is the pipeline on data collection, labeling and evaluation scalable? can you easily extend to other languages and data sources? from the current version it looks very manual and not scalable. It would be good to share more insight as extending the dataset to more image sources and languages, even in future, would be great. (please refer to my comments in the previous section)

Can we add more open ended evaluation methods? Moving beyond fixed options, or providing more open-ended tasks without predefined answers, would allow for a deeper evaluation of model understanding.

---

> ### Author Response · Authors · 2024-11-14
> **Response to the reviewer [Concern 1,2]**
>
> **Concern 1: why these areas are chosen over others.**
>
> ----
>
> Thank you for your insightful feedback. We appreciate the recognition of our dataset's value in advancing QA-focused benchmarks.
>
> **Practical Value:** Instead of choosing data sources like COCO, our primary goal was to facilitate understanding of high-resolution, real-world scenarios, which are closely related to fields like autonomous driving and surveillance. In recent studies, MLLMs have increasingly been applied to these realistic domains, which pose unique challenges for multimodal large language models (MLLMs). The domains we selected specifically demand high image resolution, allowing MLLMs to demonstrate progress in these high-resolution settings. We have thus prioritized areas where MLLMs currently have the potential for meaningful impact.
>
> **Benchmark Objective:** Our benchmark is not intended solely for autonomous driving or surveillance applications. Our core aim is to evaluate MLLMs’ fundamental perception and reasoning abilities within these domains. Therefore, our tasks encompass various scenarios that our researchers deemed suitable for assessment through multiple-choice QA formats. We intentionally excluded more specific or complex tasks, such as robotic arm trajectory prediction, to maintain the focus on fundamental capabilities.
>
> Additionally, as suggested, including a broader range of embodied understanding tasks—such as those in robotics—would expand the dataset’s scope. In future iterations, we plan to extend the dataset to additional domains, including human-robot interaction and real-time monitoring applications, to promote a more balanced representation and reduce domain-specific bias.
>
> ----
>
> **Concern 2: "While tackling everything can be challenging, my worry is that this dataset will introduce bias toward certain domains while trying to address limitations of other benchmarks."**
>
> As mentioned in Concern 1, the tasks we included are fundamental perception and reasoning tasks designed by our expert researchers. They do not cover complex tasks such as dynamic tracking in surveillance images or trajectory prediction in autonomous driving. Thus, they remain within the realm of basic perception/reasoning capability testing for MLLMs and we try our best to make them not interfere with specific downstream tasks strongly.
>
> Furthermore, researchers have flexibility in selecting splits for evaluation. For instance, we also included standard MLLM tasks such as OCR and chart recognition, which involve higher-resolution images and improved annotation quality compared to traditional MLLM benchmarks. We believe these tasks are also valuable for advancing foundational research in their respective fields and will have a meaningful impact on MLLM development in these areas.

---

> > ### Comment · Reviewer_qNEP · 2024-11-20
> > **Response to feedback - Concern 1 on diversity and task coverage**
> >
> > I appreciate the authors' response and comments. However, I still find the response unconvincing. Perception is a broad field, and focusing solely on video monitoring and autonomous driving represents only a small fraction of it. Furthermore, there are numerous chart and OCR VQA benchmarks available. Datasets like COCO provide much greater diversity in terms of object and scene coverage. Additionally, I disagree with the claim that 'we try our best to make them not interfere with specific downstream tasks strongly,' as the data choices are somewhat limited, inherently introducing a particular focus. For instance, the benchmark places emphasis on vehicles and pedestrians.

---

> > > ### Author Response · Authors · 2024-11-21
> > > **Thanks for your suggesstion!**
> > >
> > > Thank you for your response. We are very open to expanding the dataset, but we would appreciate it if the reviewer could clarify what constitutes a "diverse" perception dataset in terms of size or the number of categories it should include. Alternatively, if the reviewer believes that a convincing benchmark must include specific classes of images, we would like to know which ones.  Actually, video monitoring and autonomous driving make up only a small portion of the dataset (about 1/4 of the perception tasks). We have over 10,000 samples in the OCR chart perception task, featuring a wide variety of street scenes, natural images, and chart data, which is already significantly larger than many specialized OCR datasets.
> > >
> > > Furthermore, we would like to clarify the reviewer's comment that "the benchmark places emphasis on vehicles and pedestrians." First, vehicles and pedestrians, as dynamic objects, are major components of outdoor scenes and hold significant value. Additionally, we have numerous items related to background objects (such as traffic lights) and logical reasoning about the interactions between vehicles and pedestrians, which are not limited to tasks focused solely on vehicle and pedestrian perception.
> > >
> > > Moreover, while COCO does have a large number of samples, its images are relatively small, have insufficient resolution, and the scenes are not very complex. For example, many images contain only a single object, making it difficult to formulate challenging questions. This is why we did not choose COCO as a primary data source. To cover all the scenes and categories in COCO, we might have to capture images ourselves with cameras in complex scenarios to create sufficiently challenging high-resolution perception problems. This approach would undoubtedly be costly and more difficult to scale. If the reviewer has any suggestions in this regard, we would be more than happy to hear them.

---

> > > > ### Comment · Reviewer_qNEP · 2024-11-21
> > > > **Scene and image diversity**
> > > >
> > > > In addition to autonomous driving and video monitoring, many other areas and domains can be considered, such as indoor scene understanding, commerce, healthcare, robotics, AR/VR, and environmental monitoring. Including these domains will ensure that the dataset's scene and data coverage is not overly focused on specific environments or objects.
> > > >
> > > > If the authors include a detailed discussion on the data section, covering plans for extension, existing limitations, additional experiments on evaluation, and strategies to scale while reducing costs, I will be inclined to improve my rating.

---

> > > > > ### Author Response · Authors · 2024-11-22
> > > > > **Thanks for your suggesstion!**
> > > > >
> > > > > Thank you for your insightful feedback. In the revised version of our paper, we have addressed the concerns you raised in Section 2.5 and provided detailed discussions in the appendix to cover the key aspects comprehensively. The updates include:
> > > > >
> > > > > 1. **Rationale for Domain Selection (Appendix C.1):**
> > > > >    We prioritized domains such as remote sensing, surveillance, and autonomous driving for their practical value and unique challenges, focusing on high-resolution imagery with complex scenarios. These domains are better suited for testing nuanced perception and reasoning compared to simpler datasets like COCO, which lack scene complexity and high resolution.
> > > > >
> > > > > 2. **Current Limitations and Plans for Extension (Appendix C.2):**
> > > > >    Our dataset faces challenges in task diversity and scalability. For instance, there is a lack of high-resolution natural scene data and underrepresentation of domains like indoor scenes, healthcare, and AR/VR. Additionally, the dataset construction process, which requires significant human effort, limits scalability. To address these limitations, we propose capturing natural images, expanding to more domains, and exploring strategies to reduce manual effort in future iterations.
> > > > >
> > > > > 3. **Exploring Model-Assisted Approaches to Enhance Scalability (Appendix C.3):**
> > > > >    We trialed MLLMs for data filtering and question generation. While models like GPT-4o effectively filtered images, their performance in generating complex QA pairs was suboptimal, with lower task difficulty and higher error rates compared to manual annotation. This suggests that while model-assisted pipelines can reduce annotators' workload, further refinement is required to match the quality of manual processes.
> > > > >
> > > > > In addition to these discussions, we have also conducted new experiments in Section 3.3 (**Extended Metrics**), where we analyzed the impact of different metrics and studied the effect of removing choice-based options on model performance. Furthermore, we investigated the influence of different prompting techniques, such as chain-of-thought, on high-resolution perception tasks.
> > > > >
> > > > > We hope these additions address your concerns and provide clarity on our future plans and strategies to enhance the dataset. Your feedback has been invaluable in refining our work, and we greatly appreciate your consideration for an improved rating.

---

> > > > > > ### Author Response · Authors · 2024-11-25
> > > > > > **Kindly Request for Reviewer's Feedback**
> > > > > >
> > > > > > Dear Reviewer qNEP,
> > > > > >
> > > > > > Since the End of author/reviewer discussions is coming soon, may we know if our previous response addresses your main concerns? If so, we kindly ask for your reconsideration of the score. Should you have any further advice on the paper and/or our rebuttal, please let us know and we will be more than happy to engage in more discussion and paper improvements.
> > > > > >
> > > > > > Thank you so much for devoting time to improving our paper!

---

> ### Author Response · Authors · 2024-11-14
> **Response to the reviewer [Concern 3-5]**
>
> **Concern 3 The use of multiple-choice challenges, while popular, remains a somewhat limited way to assess model capabilities.**
>
> ----
>
> Thank you for your insightful suggestion. We greatly appreciate your input, as it is crucial for improving the quality of our dataset. Over the past period, we have indeed been considering alternative question formats. In response, we are preparing to release an additional evaluation version of the dataset, where only the questions are provided, and no answer choices are given. This will allow us to assess model performance using methods such as exact match or GPT-match, which can help prevent models from relying on the provided choices.
>
> In this approach, we selected 50 samples from each task and used the prompt, “Please respond to the question with a single word or phrase,” to encourage the models to generate direct answers. We then compare the model’s output to the correct answer to evaluate its performance. The table below shows the results from our initial tests, where all models experienced a significant drop in performance when no choices were provided:
>
> |      Method     | Perception |    |    |    |    |      |
> |:---------------:|:----------:|:--:|:--:|:--:|:--:|:----:|
> |                 |     OCR    | RS | DT | MO | AD |  Avg |
> |      Slime      |     58     | 36 | 51 | 29 | 33 | 37.7 |
> |    wo choice    |     10     | 13 |  2 | 15 | 11 | 11.3 |
> |     LLaVA-OV    |     82     | 51 | 64 | 34 | 45 | 52.8 |
> |    wo choice    |     34     | 15 | 25 | 13 | 13 | 18.8 |
> |      GPT-4o     |     81     | 45 | 65 | 34 | 37 | 49.1 |
> |    wo choice    |     43     | 20 | 29 |  9 | 20 | 22.8 |
> |+Machine match |     53     | 31 | 46 | 16 | 32 | 33.1 |
>
> As we can see, the performance drops notably without answer choices. However, we also observe that for tasks like OCR, where the answer is more unique, an exact match method is likely to yield the correct answer. For other tasks, especially those involving more open-ended responses (e.g., "Where is the person in yellow clothes located in the image?"), it becomes more difficult to measure accuracy using exact match, since answers like "top-right of the image" and "in front of the store" might both be valid. To address this challenge, we also experimented with using GPT-4o to align the model’s response with the intended meaning of the correct answer (as shown in the last row). This alignment process did lead to a performance improvement, but it still didn’t match the results obtained when answer choices were provided.
>
> We believe that the "Machine match" strategy will be one of the evaluation approaches we prioritize moving forward. We will also continue exploring other evaluation strategies to further improve the assessment process.
>
> Once again, thank you for your valuable suggestion!
>
> ----
>
> **Concern 4  the reliance on full human annotation restricts scalability.**
>
> This is a very valid observation, and indeed, the cost of annotating this dataset has been significant. However, we believe that human annotation provides a level of control and reliability that automated methods cannot currently match. High-quality, human-annotated data is essential for accurately evaluating MLLMs, and we plan to expand it in the future to cover additional domains and task types while maintaining this standard of quality.
>
> ----
>
> **Concern 5: "Including more languages would strengthen the dataset’s multilingual utility."**
>
> Expanding to additional languages presents a far greater challenge than adding new tasks or domains. Notably, when creating the MME-RealWorld CN version, we made a concerted effort to control annotation quality and avoid overlap between Chinese and English evaluations. This involved collecting scenes specifically in Chinese and enlisting Chinese language experts for annotation. Expanding to other languages, especially less commonly spoken ones, would require collaboration with language-specific researchers and the collection of unique image sources for each language. While this is a formidable task, we are actively exploring the possibility of extending the dataset to include widely spoken languages, such as Korean, Japanese, and Spanish. This process will require time to ensure both data quality and proper annotation.

---

> > ### Author Response · Authors · 2024-11-14
> > **Response to the reviewer [Concern 6-7]**
> >
> > **Concern 6: "However, I think it might be better suited for a dataset track or a workshop focused on datasets and evaluation."**
> >
> > We respectfully disagree with the suggestion that benchmarks should necessarily be published in workshops or dataset-specific tracks. In recent years, a considerable number of MLLM benchmarks have been published in top-tier computer vision and machine learning conferences, underscoring the value of high-quality benchmarks in advancing the field. For instance, ICLR has previously featured prominent benchmark papers, such as MathVista (ICLR 24 Oral), MathVerse (ECCV 24), BLINK (ECCV 24), and MM-Vet (ICML 24). These examples highlight the role that comprehensive benchmarks play in pushing the field forward, aligning with the inclusive scope of ICLR.
> >
> > ----
> >
> > **Concern 7: "Do the authors have plans or see it feasible to increase the diversity of data sources and tasks?"**
> >
> > In previous responses, we briefly addressed the scalability of our approach. For image sources, we begin by identifying widely-used datasets in our target domains. This initial step typically yields a substantial amount of data, after which we filter for high-resolution images to meet our standards. This process is relatively low-cost if we have access to domain-specific or language-specific experts when extending to additional languages.
> >
> > The scalability of human annotation poses a greater challenge, yet we maintain that using MLLMs for self-annotation is not a reasonable approach for their evaluation. This is a known limitation of many existing benchmarks. While there appears to be a trade-off between annotation quality and scalability, we are committed to expanding and refining the dataset. We believe that a high-quality, well-curated dataset holds more value than a large but inconsistently annotated one. Additionally, MME-RealWorld already ranks among the largest benchmarks in its category, demonstrating both size and quality.

---

> > > ### Comment · Reviewer_qNEP · 2024-11-20
> > > **Response to the authors feedback [Concern 6-7]**
> > >
> > > Regarding the best venue for publishing this work, I want to clarify that I was mainly referring to the paper in its current form and contributions. I’m well aware of the successful examples the authors shared.

---

> > > ### Comment · Reviewer_qNEP · 2024-11-20
> > > **Overall feedback to authors response**
> > >
> > > I want to thank the authors for sharing their feedback. After reviewing the responses and considering feedback from other reviewers, I still have concerns about the task selection and data diversity. Additionally, the plan for scaling remains unclear. While the benchmarks on open-ended Q&A are helpful, they seem somewhat preliminary and likely require further refinement. I would like to maintain my current rating. I believe the paper has potential but needs additional work to better prepare for publication in venues of this caliber.

---

> > ### Comment · Reviewer_qNEP · 2024-11-20
> > **Response to the authors feedback [Concern 3-5]**
> >
> > I like the table shared by the authors on benchmarking a more open-ended Q&A setup. Even in its preliminary form, I think it provides a better way to assess model capabilities on this benchmark.
> >
> > Regarding human annotation and scalability, I agree that human annotation, if done properly with the right expert crowd, will be important. However, there could be ways to improve efficiency by incorporating models in the loop or leveraging other techniques to address scalability. This might be outside the scope of the paper, but it would have been nice to see some discussion around this.

---

> ### Author Response · Authors · 2024-11-21
> **Preliminary Exploration of Using Multimodal Models to Assist Annotation**
>
> Regarding the reviewer's suggestion on incorporating models to improve annotation efficiency and scalability, we have considered several strategies to potentially reduce annotation costs without compromising quality.
>
> Our dataset construction involves two key stages requiring human involvement. The first stage is image selection and filtering, where we ensure:
>
> 1. The images contain challenging or valuable small objects.
> 2. The images are of high quality, free from glare or noise.
> 3. The scenes are clear enough to minimize ambiguity in QA tasks (e.g., if a question asks about a person in a blue shirt in the upper right corner, there should be only one such person).
>
> After ensuring high-quality and challenging images, we can provide annotators with images that are more likely to yield usable QA pairs, thus enhancing annotation efficiency.
>
> For the second stage, human annotation involves crafting questions and answers for these complex images.
>
> ----
>
> To explore whether models can help us enhance the scalability of our dataset, we have conducted a small-scale trial with the following approach:
>
> For data filtering, we adopted a model-based strategy. We first set a minimum resolution of 1024x1024 to ensure that the images are sufficiently large. We used a Multimodal Large Language Model (MLLM) to remove images that were noisy or unclear. The MLLM was then employed with the prompt: "We provide an image; please score the scene from two aspects on a scale of [1, 2, ..., 10]:
>
> 1. Complexity: Assess whether the image contains a large number of elements, such as various objects. If the scene features only one prominent object and the foreground/background is relatively simple, the score should be 0. Conversely, if the image includes a multitude of elements and the scene is complex, the score should be higher.
>
> 2. Object Salience: Evaluate whether the image contains small objects that are difficult to observe clearly. Specifically, objects occupying less than 1/20 of the total pixel area. If all objects in the image can be observed very clearly or are relatively large, the score should be 0.
>
> 3. Objects with Observational Challenges: If there are objects in the image that are difficult to observe or smaller targets, please list these objects and their locations. “
>
> With this strategy, our annotators can intuitively see the model's scoring of the images and the small objects for reference. Currently, we are using GPT-4o as the scoring model. In the future, we could use multiple models to vote for better quality assurance. This method has indeed reduced the workload for annotators, and we found that it allows us to quickly filter out simple images.
>
> In the second part, manual annotation, considering the reviewer’s comments on the scalability of manual annotation, we attempted to use GPT-4o for question generation and answer construction. However, GPT-4o did not perform well in high-resolution scenarios, especially in reasoning tasks, resulting in samples that were either too easy or incorrect, making them unusable. After some consideration, we tried using Qwen2-vl-72b, Llava-ov-72b, Claude3.5-sonnect, and GPT-4o to individually generate three questions and answers for a single image. The construction standards for questions and responses were consistent with those in our article, and prompts were input to the model as a prerequisite. Finally, humans experts retained the most challenging and reasonable questions.
>
> Through this method, we were able to obtain some good QA pairs. However, in small-scale experiments, we still observed that the task difficulty is easier and the error rate was much higher than with purely manual annotation, and the options designed manually were more reasonable. Additionally, the time spent manually reviewing multiple large model outputs was not significantly better than manual annotation.
>
> This suggests that for our ultra-high-resolution perception tasks, there may not yet be an optimal model annotation pipeline, which is a topic for future research. Any constructive feedback from the reviewers would be highly valuable.

---

> ### Author Response · Authors · 2024-12-01
> **Kindly Request for Reviewer's Feedback**
>
> Dear Reviewer qNEP,
>
> As the rebuttal ends soon, may we know if our responses have addressed your further comments?
>
> If so, we kindly ask for your reconsideration of the score. If any aspects require additional elaboration or refinement, we will be more than happy to engage in further improvements and discussion.
>
> Thanks again for your time.

---

### Official Review · Reviewer_4Fv1 · 2024-11-04

**Soundness:** 3
**Presentation:** 4
**Contribution:** 3
**Rating:** 8
**Confidence:** 3

**Summary:**

This paper presents a new evaluation benchmark for Multimodal Large Language Models (MLLMs), dubbed MME-RealWorld, which focuses on challenges that models face in the real world. Specifically, MME-RealWorld covers 29,429 question-answer pairs across 5 real-world scenarios. Experimental results on MME-RealWorld show that even the most advanced models still struggled in real-life scenarios. Besides, the authors have also conducted detailed analyses to explain the unsatisfying performance of MLLMs.

**Strengths:**

- The perspective of evaluating MLLMs in real-life scenarios, such as OCR in the Wild, Video Monitoring and Autonomous Driving is new and of significant value for practical deployment of MLLMs.
- The authors have conducted detailed comparisons with existing benchmark in Tab.2, which helps better capture the unique characteristics of MME-RealWorld.
- The authors have evaluated 24 open-sourced MLLMs and 4 closed-sourced MLLMs, which provide a comprehensive evaluation of current MLLMs.

**Weaknesses:**

- The evaluation on MME-RealWorld seems to require lots of computation resources, which may limit the accessibility for researchers with fewer resources.

**Questions:**

Do the authors have plans to expand or adapt MME-RealWorld to include new tasks or modalities as MLLMs capabilities evolve?

---

> ### Author Response · Authors · 2024-11-14
> **Response to the reviewer**
>
> **Concern 1 The evaluation on MME-RealWorld seems to require lots of computation resources, which may limit the accessibility for researchers with fewer resources.**
>
> ----
>
> This feedback is incredibly valuable! To facilitate the evaluation of high-resolution perception abilities in multimodal large language models, we plan to sample 50 examples from each task within each domain, resulting in a test set of 2,150 QA pairs. We believe, as per your suggestion, that this smaller-scale test set will have a greater impact.
>
> |    Method   |    LLM    | Overall | Perception |    |    |    |    |      | Reasoning |    |    |    |      |
> |:-----------:|:---------:|:-------:|:----------:|:--:|:--:|:--:|:--:|:----:|:---------:|:--:|:--:|:--:|:----:|
> |             |           |         |     OCR    | RS | DT | MO | AD |  Avg |    OCR    | DT | MO | AD |  Avg |
> | GPT-4o-mini |     -     |   37.4  |     70     | 23 | 62 | 19 | 34 | 38.8 |     57    | 39 | 19 | 35 | 35.2 |
> |   Qwen2-VL  |  Qwen2-7B |   46.7  |     86     | 40 | 74 | 28 | 36 | 48.2 |     73    | 46 | 47 | 36 | 44.4 |
> |   LLaVA-OV  |  Qwen2-7B |   48.5  |     82     | 51 | 64 | 34 | 45 | 52.8 |     71    | 43 | 45 | 35 | 42.7 |
> |    Slime    | Llama3-8B |   37.1  |     58     | 36 | 51 | 29 | 33 | 37.7 |     51    | 27 | 41 | 34 | 36.4 |
>
> Above are our initial tests with some models. While it is difficult to perfectly retain model performance on the original dataset due to the sampling process, the overall task difficulty has actually increased, as we’ve reduced the number of samples for simpler tasks such as OCR, charts, and basic reasoning tasks. For instance, the performance of Qwen2-VL dropped from 56.4 on the full dataset to 46.7 in this reduced test set.
>
> We believe this compact test set is an excellent tool for researchers to perform smaller-scale, focused evaluations and comparisons of their models. It strikes a good balance between practicality and robustness, making it more accessible for the community while still providing meaningful insights into model performance.

---

### Official Review · Reviewer_oyYW · 2024-11-04

**Soundness:** 3
**Presentation:** 3
**Contribution:** 3
**Rating:** 6
**Confidence:** 4

**Summary:**

This paper analyzes the limitations of existing MLLM benchmarks and observes three significant challenges (i.e., small data scale, restricted data quality of model-based annotations, and insufficient task difficulty especially caused by the limited image resolution). To address these issues, this paper constructs a new benchmark named MME-RealWorld that is the largest manually annotated benchmark to date and features the highest resolution and a targeted focus on real-world applications. The authors benchmark 29 advanced MLLMs on the introduced MME-RealWorld, revealing the limitations of existing MLLMs (none of them reach 60% accuracy).

**Strengths:**

+ The overall paper is well organized and easy to follow. The motivation of constructing the dataset is clear.
+ This paper contributes a large-scale, large-resolution and manually annotated VQA dataset, which contains 13,366 high-quality images and 29,429 question-answer pairs covering 43 subtasks across 5 real-world scenarios.
+ Large number of MLLMs are benchmarked on the introduced benchmark, i.e., 4 close-source and 25 open-source MLLMs in total.

**Weaknesses:**

- Although the dataset contribution is great, it is hard to find some insightful analysis from this paper. The authors only summarize the benchmark results. It would be better to provide more in-depth analysis and highlight the future direction.
- In Lines 479-480, the authors claim that ‘This indicates that most models’ visual perception modules fail to identify the objects in the images corresponding to our questions.’ This is not clear and convincing. I am wondering how do the authors attribute the higher frequency of ‘E’ outputs to the limited image detail perception of MLLMs?
- In Lines 481-485, the authors analyze the ‘limitations of MLLMs in understanding dynamic information’ without providing any qualitative or quantitative result. It would be more convincing to provide some evidences to support such statement.
- Figure 2a shows the different domains and tasks. It would be nice to add a figure to show the distributions of different domains and tasks for better readability.
- In Lines 93- 94, the authors claim ‘we collect a total of 13, 366 high-resolution images from more than 300K public and internet sources.’ I have doubts about the number of the data sources. 300K is a huge number, and how long does it take to collect the images?
- For better readability, it would be better to add a paragraph title for the last paragraph in Section 3.3.

**Questions:**

Please refer the Weaknesses.

---

> ### Author Response · Authors · 2024-11-14
> **Response to the reviewer [concern 1]**
>
> **Concern 1: Although the dataset contribution is great, it is hard to find some insightful analysis from this paper. The authors only summarize the benchmark results. It would be better to provide more in-depth analysis and highlight future directions.**
>
> ----
>
> Thank you for this highly relevant suggestion for enhancing the paper. As one of the few ultra-high-resolution benchmarks, we have conducted preliminary analyses on the trade-offs between computational cost and model performance to inspire future research directions. Specifically, we have highlighted the following insights:
>
> 1. The perceptual abilities of current models are limited, resulting in subpar performance on our benchmark.
>
> 2. Proficiency in Chinese is a significant limitation for existing models, though Llama 3, as a base model, demonstrates comparatively strong performance in this area.
>
> 3. Simple image-splitting techniques are insufficient to fully address high-resolution perception challenges.
>
> 4. Models still struggle with tasks involving the understanding, prediction, and inference of objects’ next actions, such as anticipating the intentions of vehicles or pedestrians in images, reflecting a gap relative to world models.
>
> 5. High-resolution models face computational limitations; excessive image patching demands considerable resources. Striking a balance between high-resolution perception and computational efficiency remains a critical issue.
>
> Recognizing that our current analysis may be insufficient, we have extended our evaluation with additional insights into model behavior, particularly focusing on instances where models choose to refuse answering. This is directly related to model safety and utility.
>
> **1. Correct E (%):** This metric measures the percentage of times a model correctly predicts "E" when the ground truth is "E," helping evaluate whether the model can recognize questions that genuinely require a refusal to respond.
>
> **2. E Ratio in Wrong Predictions (%):** This indicates the proportion of wrong answers labeled as "E" among all incorrect predictions, offering insights into the model's tendency to choose "E" when uncertain.
>
> **3. # Predicted E:** This counts the total number of times a model predicts "E" within the current split, providing a straightforward view of "E" predictions overall.
>
> The results are shown in Table 21 of the revised pdf or [Response to the reviewer a7Si [concern 2]]. Below are the summarized results, from which we draw several key insights:
>
> a) Less capable models, such as SliME-8B, rarely opt for "E" and instead tend to select answers they consider correct, leading to low rates of both correct "E" predictions and "E" frequency across domains.
>
> b) More advanced open-source models, like InternVL, show "E" frequencies comparable to closed-source models on simpler tasks, such as OCR or Diagram and Table interpretation. In these easier tasks, most models are confident in their answers, so the frequency of selecting "E" remains low.
>
> c) On more challenging tasks, such as Remote Sensing and Monitoring, all models have higher error rates. Notably, proprietary models like GPT-4o and Claude35 exhibit both higher accuracy in predicting "E" and a greater frequency of selecting "E" compared to InternVL, which maintains a relatively low "E" frequency (10%-35%) even on difficult tasks.
>
> In summary, for simpler multimodal tasks, the safety profiles of advanced MLLMs and proprietary models are comparable. However, on more complex tasks, proprietary models demonstrate a significantly higher level of safety by opting for "E" when uncertain, aligning better with human values by avoiding misleading answers. Given that open-source models currently undergo limited alignment with human preferences, this presents an important direction for future research and development.
>
> ----
> Overall, we have made every effort to provide a comprehensive analysis of model performance in high-resolution perception, Chinese-language scenarios, computational overhead, dynamic information recognition (especially critical for Embodied AI and autonomous driving), and model safety. In doing so, we have identified several key challenges in these areas and highlighted promising avenues for future research.
>
> **If the reviewers can point out specific areas where our experiments may not have been sufficiently thorough**, or where a more in-depth analysis is needed, we would be happy to provide additional details or further explore those aspects.

---

> ### Author Response · Authors · 2024-11-14
> **Response to the reviewer [concern 2-6]**
>
> **Concern 2 In Lines 479-480, the authors claim that ‘This indicates that most models’ visual perception modules fail to identify the objects in the images corresponding to our questions.’ This is not clear and convincing. I am wondering how do the authors attribute the higher frequency of ‘E’ outputs to the limited image detail perception of MLLMs?**
>
> ----
>
> To clarify, the statement regarding the high frequency of 'E' outputs is meant to highlight that the correct answers in our dataset have a relatively low proportion of 'E' as the answer choice. Therefore, when models predominantly select 'E', it suggests that they are unable to correctly identify the relevant objects in the images corresponding to the questions. This tendency indicates a weakness in the model’s visual perception ability, as it resorts to a default or non-informative answer (such as 'E') when unable to perceive the correct object or answer. We see this as an indication of limited visual perception capacity in the model, which is unable to effectively map the visual input to the correct answer.
>
> ----
>
> **Concern 3 In Lines 481-485, the authors analyze the ‘limitations of MLLMs in understanding dynamic information’ without providing any qualitative or quantitative result. It would be more convincing to provide some evidences to support such statement.**
>
> |                   |     Eng    |     CN     | AD-Intention-Eng |            |          | AD-Intention-CN |            |          |  Avg  |
> |-------------------|:----------:|:----------:|:----------------:|:----------:|:--------:|:---------------:|:----------:|:--------:|:-----:|
> |                   | Monitoring | Monitoring |        Ego       | Pedestrian | Verhicle |       Ego       | Pedestrian | Verhicle |       |
> |       GPT-4o      |    13.27   |    8.16    |       17.11      |    19.42   |   27.54  |      26.00      |    16.00   |   23.00  | 18.81 |
> | Claude 3.5 Sonnet |    18.37   |    22.45   |       26.32      |    32.04   |   24.64  |      34.00      |    20.00   |   25.00  | 25.35 |
> |     InternVL-2    |    21.43   |    21.43   |       24.01      |    43.69   |   32.85  |      25.00      |    33.00   |   30.00  | 28.93 |
> |      Qwen2-VL     |    19.39   |    17.35   |       19.08      |    43.69   |   35.75  |      25.00      |    37.00   |   36.00  | 29.16 |
>
> To address the reviewer’s suggestion for additional evidence supporting the limitations of MLLMs in understanding dynamic information, we have included detailed results (refer to Table 6 in the revised PDF) derived from autonomous driving and monitoring tasks focused on intention prediction. These results reveal that even the strongest models perform poorly on intention prediction tasks, with average accuracies not exceeding 30%. Intention prediction is a fundamental capability in fields like autonomous driving, and it is essential for MLLMs to evolve into robust world models capable of understanding dynamic contexts. This evidence further underscores the limitations of current models in this critical area.
>
> ----
>
> **Concern 4  Figure 2a shows the different domains and tasks. It would be nice to add a figure to show the distributions of different domains and tasks for better readability.**
>
> We appreciate the reviewer’s suggestion to improve the readability of the distribution of domains and tasks. To address this, we have included a comprehensive figure (Figure 15 in the revised PDF) that provides a detailed breakdown of the perception and reasoning tasks, along with the sample counts for each subtask. We believe this enhancement significantly improves the clarity and readability of our paper.
>
>
> ----
>
> **Concern 5 In Lines 93- 94, the authors claim ‘we collect a total of 13, 366 high-resolution images from more than 300K public and internet sources.’ I have doubts about the number of the data sources. 300K is a huge number, and how long does it take to collect the images?**
>
> Thank you for your question. The data collection process involved five multimodal research experts working for one month. In some areas, such as autonomous driving, remote sensing, and street view data, high-resolution images were readily available, so the main task was content filtering rather than image collection, which helped reduce the time needed. The majority of the effort went into filtering images that met the resolution and content requirements.
>
> For tasks with fewer public data sources, such as Chinese scene data, researchers had to manually search for and download the necessary images. Although this portion of the data was smaller, it still accounted for about 1/3 of the total time spent on data collection and organization.
>
> ----
>
> **Concern 6 For better readability, it would be better to add a paragraph title for the last paragraph in Section 3.3.**
>
> Thank you very much for your suggestion. We have added the title "Appendix Overview: Supplementary Analyses and Detailed Results" to improve readability in the last paragraph of Section 3.3.

---

> > ### Comment · Reviewer_oyYW · 2024-11-19
> >
> > Thanks for the detailed response. Most of my concerns have been addressed, and I would like to raise my rating.
> >
> > Additionally, I have some questions (might be slightly beyond the scope of this work).
> > - The authors mentioned ‘the perceptual abilities of current models are limited’. I am wondering if the current bottleneck is the LLM or the visual encoder and whether the different vision encoder largely effect the final performance? In addition to LLM information, it would be nice to provide the vision encoder information in the tables.
> > - Is it possible that the errors (high frequency of ‘E’ outputs) are caused by some other limitations of MLLMs (e.g., poor reasoning capabilities)? Is there any solution to probe the source of the errors?

---

> ### Author Response · Authors · 2024-11-19
> **Thanks for your suggestion!**
>
> **Concern 1: The authors mentioned, "the perceptual abilities of current models are limited."**
>
> It is indeed challenging to completely separate a model’s capability to process input images from its inherent perceptual ability, as these two aspects are closely coupled. However, in the case of high-resolution images, the model's capacity to process inputs seems especially crucial. For example, as shown in the table below, Mini-Gemini-7B-HD and LLaVA1.5-7B use similar LLM architectures and have comparable training data, yet Mini-Gemini-7B-HD exhibits far superior high-resolution perception capabilities. This demonstrates the critical importance of handling higher-resolution data effectively. As a result, most modern MLLMs have incorporated various image-splitting strategies to accommodate larger maximum resolutions.
>
> Nevertheless, simply supporting higher resolution is not a complete solution to high-resolution perception challenges. For instance, while Intern-VL2 has a higher input resolution limit than Qwen2-VL, its overall performance is slightly lower. This implies that the ability to handle larger image resolutions alone is insufficient for robust high-resolution perception. The model’s inherent capabilities (such as information extraction and comprehension) also play a vital role.
>
> Additionally, current MLLMs do not perform well on MME-RealWorld, and as we noted in the paper, computational efficiency is an issue. Efficiently handling ultra-high-resolution images remains an open question.
>
> Given the diversity of vision encoders (many of which are not open-sourced) and the fact that many models utilize multiple vision encoders, it is challenging to assess the quality of each encoder independently. Instead, we list the maximum resolution each method can handle in the table below, as this feature better characterizes a model’s ability to process high-resolution image inputs.
>
> |         Model         |          LLM         | Max Resolution |
> |:---------------------:|:--------------------:|:--------------:|
> |      Qwen-VL-Chat     |         Qwen         |       448      |
> |      LLaVA1.5-7B      |       Vicuna-7B      |       336      |
> |       LLaVA-Next      |       LLama3-8B      |       672      |
> |    mPLUG-DocOwl 1.5   |       LLama-7B       |       448      |
> |    ShareGPT4V-13B     |      Vicuna-13B      |       336      |
> |       MiniGPT-v2      |   Llama 2-7B-Chat    |       448      |
> |         Monkey        |        Qwen-7B       |    896*1334    |
> |     Cambrian-1-34B    |    Hermes2-Yi-34B    |      1024      |
> |      DeepSeek-VL      | DeepSeek-LLM-7b-base |      1024      |
> |       YI-VL-34B       |      Yi-34B-Chat     |       448      |
> |     MiniCPM-V 2.5     |       LLama3-8B      |      1344      |
> | InternLM-XComposer2.5 |     InternLM2-7B     |      4096      |
> |  CogVLm2-llama3-Chat  |       LLama3-8B      |      1344      |
> |   Mini-Gemini-34B-HD  | Nous-Hermes-2-Yi-34B |       672      |
> |       SliME-13B       |      Vicuna-13B      |      2016      |
> |   InternVL-Chat-V1-5  |  InternLM2-Chat-20B  |      4096      |
> |       InternVL-2      |  InternLM2.5-7B-Chat |      4096      |
> |        Qwen2-VL       |        Qwen-7B       |      3584      |
> ----
>
> **Concern 2 Is it possible that the errors (high frequency of ‘E’ outputs) are caused by some other limitations of MLLMs**
>
> This is an excellent question. In fact, as seen in Table 21, proprietary models tend to output “E” (for “none of the above”) at a slightly higher frequency, while weaker models, like SliME, are less inclined to choose “E.” We primarily attribute the “E” output to situations where the model either refuses to respond due to uncertainty or chooses “E” when it cannot select a correct answer. Importantly, this behavior is not necessarily a defect; refusing to respond when uncertain is much better than providing a potentially incorrect answer, where the latter could more confuse users. Currently, open-source models often lack robust alignment processes, making them more likely to select a specific answer instead of “E,” even if the question exceeds their capacity. Therefore, we believe that the frequency of choosing “E” is closely tied to the model's safety features.
>
> Of course, your suggestion that “E” outputs could result from poor reasoning capabilities is also valid. If a model consistently chooses “E,” it may indeed lack relevant reasoning or perception abilities. However, we have not observed a weak model that frequently outputs “E” in our experiments. Thus, our conclusions remain focused on the points made in the first paragraph and the experiments described in Concern 1. If the reviewer has encountered cases where models with poor reasoning capabilities frequently opt for “E” or exhibit similar behavior, we would greatly appreciate any shared insights to enable a more detailed discussion.

---

> > ### Comment · Reviewer_oyYW · 2024-11-19
> >
> > Thanks for your prompt response. I will keep my positive rating.

---

### Official Review · Reviewer_a7Si · 2024-11-04

**Soundness:** 3
**Presentation:** 3
**Contribution:** 3
**Rating:** 8
**Confidence:** 5

**Summary:**

This paper introduces MME-RealWorld, a large-scale, fully manually annotated benchmark for multimodal large language models across diverse real-world tasks. The MME-RealWorld has 13366 images with 29429 question-answer pairs annotated by humans. It covers 43 subtasks across 5 real-world scenarios. MME-RealWorld is the largest manually annotated benchmark to date, featuring the highest resolution and a targeted focus on real-world applications. It reports comprehensive results with various leading LVLMs.

**Strengths:**

1. MME-RealWorld is the largest manually annotated benchmark for LVLMs across 43 subtasks across 5 real-world scenarios.

2. MME-RealWorld has the highest resolution among manually annotated benchmarks.

3. It reveals several insights for current LVLMs. For example, existing models still lack abilities for image detail perception and dynamic information understanding.

**Weaknesses:**

Overall, the proposed MME-RealWorld is a solid paper. There are some additional comments:
1. The holistic MME-RealWorld is too large for developing LVLMs. The author should formulate a mini set of MME-RealWorld for community convenience. I suggest maintaining a balance of task types and difficulty levels while reducing the overall size
2. There exists a clear preference for preparatory APIs for selecting E or refusing to answer the question. Does this mean the preparatory APIs are better aligned with human values and have better AI security? The author may discuss if it is possible to compare preparatory APIs with open-sourced models in a better way. Moreover, I suggest that the authors conduct a detailed analysis of when and why models choose option E or refuse to answer.
3. MME-RealWorld has a large scale of images and question-answer pairs. The author should discuss if MME-RealWorld is sufficiently diverse or just collecting several similar images and QAs. More comparisons on the distribution of image types, question types, and answer types between MME-RealWorld and other benchmarks are welcomed.
4. It is better to have a discussion of knowledge leakage LVLMs on MME-RealWorld benchmark, as in [1].

[1] Are We on the Right Way for Evaluating Large Vision-Language Models?

**Questions:**

Overall, the proposed MME-RealWorld is a solid paper. There are some additional comments in Weakness section.

---

> ### Author Response · Authors · 2024-11-14
> **Response to the reviewer [concern 1]**
>
> **Concern 1** The holistic MME-RealWorld is too large for developing LVLMs. The author should formulate a mini set of MME-RealWorld for community convenience. I suggest maintaining a balance of task types and difficulty levels while reducing the overall size
>
> ----
>
> This feedback is incredibly valuable! To facilitate the evaluation of high-resolution perception abilities in multimodal large language models, we plan to sample 50 examples from each task within each domain, resulting in a test set of 2,150 QA pairs. We believe, as per your suggestion, that this smaller-scale test set will have a greater impact.
> |    Method   |    LLM    | Overall | Perception |    |    |    |    |      | Reasoning |    |    |    |      |
> |:-----------:|:---------:|:-------:|:----------:|:--:|:--:|:--:|:--:|:----:|:---------:|:--:|:--:|:--:|:----:|
> |             |           |         |     OCR    | RS | DT | MO | AD |  Avg |    OCR    | DT | MO | AD |  Avg |
> | GPT-4o-mini |     -     |   37.4  |     70     | 23 | 62 | 19 | 34 | 38.8 |     57    | 39 | 19 | 35 | 35.2 |
> |   Qwen2-VL  |  Qwen2-7B |   46.7  |     86     | 40 | 74 | 28 | 36 | 48.2 |     73    | 46 | 47 | 36 | 44.4 |
> |   LLaVA-OV  |  Qwen2-7B |   48.5  |     82     | 51 | 64 | 34 | 45 | 52.8 |     71    | 43 | 45 | 35 | 42.7 |
> |    Slime    | Llama3-8B |   37.1  |     58     | 36 | 51 | 29 | 33 | 37.7 |     51    | 27 | 41 | 34 | 36.4 |
>
> Above are our initial tests with some models. While it is difficult to perfectly retain model performance on the original dataset due to the sampling process, the overall task difficulty has actually increased, as we’ve reduced the number of samples for simpler tasks such as OCR, charts, and basic reasoning tasks. For instance, the performance of Qwen2-VL dropped from 56.4 on the full dataset to 46.7 in this reduced test set. This demonstrates how the smaller test set not only maintains a balance of different tasks but also keeps the tests challenging, aligning with your suggestion to "maintain difficulty levels."
> We believe this compact test set is an excellent tool for researchers to perform smaller-scale, focused evaluations and comparisons of their models. It strikes a good balance between practicality and robustness, making it more accessible for the community while still providing meaningful insights into model performance.

---

> > ### Author Response · Authors · 2024-11-14
> > **Response to the reviewer [concern 2]**
> >
> > **Concern 2: There exists a clear preference for preparatory APIs for selecting "E" or refusing to answer the question**
> > ----
> > To address your concern, we designed three metrics to assess the frequency and performance of models when predicting "E" (indicating refusal to answer):
> >
> > 1. **Correct E (%)**: This metric measures the percentage of times a model correctly predicts "E" when the ground truth is "E," helping evaluate whether the model can recognize questions that genuinely require a refusal to respond.
> >
> > 2. **E Ratio in Wrong Predictions (%)**: This indicates the proportion of wrong answers labeled as "E" among all incorrect predictions, offering insights into the model's tendency to choose "E" when uncertain.
> >
> > 3. **# Predicted E**: This counts the total number of times a model predicts "E" within the current split, providing a straightforward view of "E" predictions overall.
> >
> > | Remote sensing |               |                                 |               |                 |     Monitoring    |               |                                 |               |                 |
> > |:--------------:|:-------------:|:-------------------------------:|:-------------:|:---------------:|:-----------------:|:-------------:|:-------------------------------:|:-------------:|:---------------:|
> > |      Model     | Correct E (%) | E Ratio in Wrong Prediction (%) | # Predicted E | Error Ratio (%) |       Model       | Correct E (%) | E Ratio in Wrong Prediction (%) | # Predicted E | Error Ratio (%) |
> > |     GPT-4o     |     50.00%    |              56.19%             |      1500     |      71.08%     |       GPT-4o      |     94.18%    |              47.39%             |      1138     |      67.50%     |
> > |    Claude35    |     71.43%    |              63.00%             |      1759     |      74.26%     |      Claude35     |     97.26%    |              48.96%             |      1206     |      69.89%     |
> > |      SliME     |     7.14%     |              9.13%              |      198      |      57.73%     |       SliME       |     85.96%    |              18.23%             |      571      |      65.11%     |
> > |    InternVL    |     21.43%    |              18.31%             |      418      |      60.65%     |      InternVL     |     91.44%    |              31.71%             |      800      |      62.39%     |
> > |       **OCR**      |               |                                 |               |                 | **Diagram and Table** |               |                                 |               |                 |
> > |     GPT-4o     |     15.07%    |              18.57%             |      284      |      23.56%     |       GPT-4o      |     11.76%    |              24.84%             |      763      |      53.65%     |
> > |    Claude35    |     13.70%    |              38.02%             |      686      |      28.49%     |      Claude35     |     5.88%     |              4.10%              |       78      |      32.92%     |
> > |      SliME     |     0.00%     |              3.86%              |       51      |      50.32%     |       SliME       |     0.00%     |              9.30%              |       39      |      70.65%     |
> > |    InternVL    |     21.92%    |              28.19%             |      498      |      27.40%     |      InternVL     |     23.53%    |              29.58%             |      692      |      39.15%     |
> >
> > Finally, **Error Ratio (%)** reflects the model's perceptual accuracy within the domain. Above are the summarized results, from which we draw several key insights:
> >
> > a) Less capable models, such as SliME-8B, rarely opt for "E" and instead tend to select answers they consider correct, leading to low rates of both correct "E" predictions and "E" frequency across domains.
> >
> > b) More advanced open-source models, like InternVL, show "E" frequencies comparable to closed-source models on simpler tasks, such as OCR or Diagram and Table interpretation. In these easier tasks, most models are confident in their answers, so the frequency of selecting "E" remains low.
> >
> > c) On more challenging tasks, such as Remote Sensing and Monitoring, all models have higher error rates. Notably, proprietary models like GPT-4o and Claude35 exhibit both higher accuracy in predicting "E" and a greater frequency of selecting "E" compared to InternVL, which maintains a relatively low "E" frequency (10%-35%) even on difficult tasks.
> >
> > In summary, for simpler multimodal tasks, the safety profiles of advanced MLLMs and proprietary models are comparable. However, on more complex tasks, proprietary models demonstrate a significantly higher level of safety by opting for "E" when uncertain, aligning better with human values by avoiding misleading answers. Given that open-source models currently undergo limited alignment with human preferences, this presents an important direction for future research and development.
> > All the results are added into the main paper.

---

> ### Author Response · Authors · 2024-11-14
> **Other Concerns**
>
> **Concern 3: The author should discuss if MME-RealWorld is sufficiently diverse or simply collecting several similar images and QAs. More comparisons on the distribution of image types, question types, and answer types between MME-RealWorld and other benchmarks are welcomed.**
>
> ----
>
> Thank you for this excellent suggestion.
>
> To address your first question:
>
> **1. Diverse Data Sources:** As outlined in Section B.1.1, our benchmark draws from a significantly broader range of publicly available datasets. Specifically, our data originates from over 17 distinct sources, covering a variety of scenarios, including remote sensing, surveillance, autonomous driving, charts, and natural images. This approach differs from existing benchmarks, which primarily focus on natural scenes or sample data from a limited pool. For instance, MMBench primarily samples from existing VQA benchmarks like CoCo and TextVQA, leading to substantial overlap with prior datasets. In contrast, our data sources are more varied and avoid substantial overlap with known MLLM benchmarks.
>
> **2. Uniquely Curated Content:** Additionally, we include data manually sourced from the internet, incorporating unique scenarios, including purely Chinese-language settings. Such data is scarce in existing datasets and provides an element absent from current benchmarks. In Figure 14 of the revised PDF, we present the various categories of images in our dataset.
>
>
> Regarding question and answer formats, we largely follow the mainstream approach by adopting multiple-choice questions, so there are no major differences in this respect. Recognizing that image diversity may not have been adequately discussed in the initial draft, we have added a related discussion to further clarify this aspect.
>
> ----
>
>
> **Concern 4: It is better to have a discussion of knowledge leakage in LVLMs on the MME-RealWorld benchmark, as in [1].**
>
>
> Thank you for this suggestion. We appreciate the reference to [1], which we have cited in our paper as it provides valuable insights. We are mindful of potential knowledge leakage; however, MME-RealWorld benefits from a distinct advantage due to its entirely manual annotation process, which greatly minimizes this issue. Our strict annotation guidelines require annotators to focus on subtle details within each image, as shown in Figure 1. As a result, even when presented with the image, models struggle with these questions, and without the image, their responses are essentially random (closed-source models default to selecting “E” or rejecting to respond).
>
> We have added a related discussion to the paper to enhance readers’ understanding of our approach.

---

> ### Comment · Reviewer_a7Si · 2024-11-15
>
> Thanks for your solid response. My concerns are well solved. I will keep my rating.

---

### Meta-Review · Area_Chair_QGhj · 2024-12-20

**Metareview:**

This paper introduces MME-RealWorld, a benchmark designed to address limitations in existing multimodal large language model (MLLM) benchmarks, such as small data scale, restricted data quality, and insufficient task difficulty. The dataset comprises 13,366 high-quality images and 29,429 human-annotated question-answer pairs across 43 subtasks within five real-world scenarios. The authors benchmarked 29 MLLMs on MME-RealWorld, revealing limitations in current models. While the paper is praised for its clear motivation, well-organized presentation, and comprehensive dataset contribution, reviewers note the lack of in-depth analysis of results, insufficient evidence for claims regarding MLLM limitations, and the need for a smaller, balanced subset of the dataset for community use. Additional discussions on dataset diversity, model biases, and knowledge leakage are also suggested. Overall, it is a solid contribution with room for further improvement in analysis and usability. The meta-reviewer suggests accepatcne given the positive feedbacks from reviewers.

**Additional Comments On Reviewer Discussion:**

The authors' rebuttal generally convinces the reviewers.

---

### Decision · Program_Chairs · 2025-01-22

Accept (Poster)